# Nonlinear transformers can perform inference-time feature learning

Naoki Nishikawa [* 1 2]  Yujin Song [* 1 2]  Kazusato Oko [2 3]  Denny Wu [4 5]  Taiji Suzuki [1 2]

## Abstract

Pretrained transformers have demonstrated the ability to implement various algorithms at inference time without parameter updates. While theoretical works have established this capability through constructions and approximation guarantees, the optimization and statistical efficiency aspects remain understudied. In this work, we investigate how transformers learn features in-context – a key mechanism underlying their inference-time adaptivity. We focus on the in-context learning of single-index models $y = \sigma_*(\langle \boldsymbol{x}, \boldsymbol{\beta} \rangle)$, which are low-dimensional nonlinear functions parameterized by feature vector $\boldsymbol{\beta}$. We prove that transformers pretrained by gradient-based optimization can perform *inference-time feature learning*, i.e., extract information of the target features $\boldsymbol{\beta}$ solely from test prompts (despite $\boldsymbol{\beta}$ varying across different prompts), hence achieving an in-context statistical efficiency that surpasses any non-adaptive (fixed-basis) algorithms such as kernel methods. Moreover, we show that the inference-time sample complexity surpasses the Correlational Statistical Query (CSQ) lower bound, owing to nonlinear label transformations naturally induced by the Softmax self-attention mechanism.

## 1 Introduction

Large language models (LLMs) are capable of In-Context Learning (ICL) (Brown et al., 2020), where they construct variable inference-time algorithms based on examples provided in the test prompt, without updating the model parameters. Recent theoretical studies have shown that pretrained transformers (Vaswani et al., 2017) can implement regression algorithms in their forward passes when test prompts

contain input-output pairs exhibiting certain functional relationships. For instance, a substantial body of research (Garg et al., 2022; Von Oswald et al., 2023; Ahn et al., 2023; Mahankali et al., 2023a; Zhang et al., 2023; Gatmiry et al., 2024) has demonstrated that linear transformers can perform linear regression in-context. As for *nonlinear* transformers, which possess greater expressive power to implement more complex algorithms (Bai et al., 2023; Cheng et al., 2023; Guo et al., 2023; Kim et al., 2024), existing optimization analyses generally do not provide end-to-end statistical guarantees for in-context learning of nontrivial nonlinear function classes (Kim & Suzuki, 2024a; Yang et al., 2024; Bu et al., 2024). A notable exception is Oko et al. (2024b), which characterizes the optimization and sample complexity of in-context learning for Gaussian single-index models using a shallow transformer. Specifically, they showed that for a degree-$P$ single-index function class of dimension $r$, the required in-context sample size of the pretrained transformer scales as $r^{\Theta(P)}$, which is comparable to the performance of kernel methods on an $r$-dimensional subspace.

**Inference-Time Feature Learning.** Our starting point is the observation that for learning low-dimensional target functions (e.g., single-index models) on isotropic data, kernel methods and, more generally, non-adaptive (fixed-feature) estimators are statistically suboptimal. Numerous prior studies have shown that neural networks can outperform kernel methods due to *feature learning* (Ghorbani et al., 2019; Ba et al., 2022; Damian et al., 2022; Abbe et al., 2022), where model parameters adapt to the low-dimensional structure of the learning problem during gradient-based training. In the ICL analysis of Oko et al. (2024b), transformers implement a kernel-like inference algorithm on a nonlinear feature map $\{f_j(\cdot)\}_{j=1}^m$ *fixed* across tasks, constructing the predictor $\sum_j a_j f_j(\cdot)$ by fitting the coefficients $a_j$ in-context. In contrast, we expect that better sample complexity can be achieved by an algorithm that *adaptively* selects the feature map at inference time, i.e., varying $\{f_j(\cdot)\}_{j=1}^m$ based on the in-context examples.

Motivated by the importance of adaptivity in the statistical efficiency of neural networks (Bach, 2017; Suzuki, 2019), in this work we theoretically study *inference-time feature learning* – the ability to adaptively extract latent features of the ground truth from test prompts without any parameter updates. We ask the following question.

---

*Equal contribution  [1]The University of Tokyo, Tokyo, Japan  [2]RIKEN AIP, Tokyo, Japan  [3]Unversity of California, Berkeley  [4]New York University  [5]Flatiron Institute.  Correspondence to: Naoki Nishikawa <nishikawa-naoki259@g.ecc.u-tokyo.ac.jp>, Yujin Song <y.song.research@gmail.com>.

*Proceedings of the 42nd International Conference on Machine Learning*, Vancouver, Canada. PMLR 267, 2025. Copyright 2025 by the author(s).

**Q1:** *Can pretrained transformers implement in-context algorithms capable of feature learning, and outperform non-adaptive methods such as kernel models?*

We study the Gaussian single-index model setting where the class of target functions is given by $f_* : \mathbb{R}^d \to \mathbb{R}$, $f_*(\boldsymbol{x}) = \sigma_*(\langle \boldsymbol{\beta}, \boldsymbol{x} \rangle)$. Here, $\sigma : \mathbb{R} \to \mathbb{R}$ is a nonlinear link function, and $f_*$ depends only on the projection along $\boldsymbol{\beta} \in \mathbb{R}^d$, referred to as the *feature vector*. The transformer model is provided with test prompts (in-context examples) in the following format:

$$\underbrace{\boldsymbol{x}_1, \boldsymbol{y}_1, \ldots, \boldsymbol{x}_N, \boldsymbol{y}_N,}_{\text{examples}} \underbrace{\boldsymbol{x}}_{\text{query}},$$

where the labels satisfy $y_i \simeq \sigma_*(\langle \boldsymbol{\beta}, \boldsymbol{x}_i \rangle)$ with feature vector $\boldsymbol{\beta}$ varying across different prompts. In this setup, inference-time feature learning refers to the model extracting $\boldsymbol{\beta}$ by examining in-context examples and using it to estimate the label $\sigma_*(\langle \boldsymbol{\beta}, \boldsymbol{x} \rangle)$ – all without any parameter updates.

This single-index setting captures scenarios where the ground truth exhibits low-dimensional structure, and has been extensively studied in deep learning theory to demonstrate the adaptivity of neural networks (Ba et al., 2022; Bietti et al., 2022; Berthier et al., 2023; Mahankali et al., 2023b). Furthermore, following Oko et al. (2024b) we assume that the feature vectors are drawn from some $r \leq d$ dimensional subspace, which implies that if pretraining identifies the support of the target function class, the in-context sample complexity should only scale with the subspace dimensionality $r$ instead of the ambient dimensionality $d$.

**Sample Complexity of Feature Learning.** The sample complexity of learning single-index models on isotropic Gaussian data in $\mathbb{R}^d$ has been extensively studied in prior works. For kernel methods, the dimension dependence can be sharply characterized, whereas for adaptive procedures such as gradient-based feature learning, the statistical complexity can be characterized by (variants of) statistical query (SQ) lower bounds (Kearns, 1998; Reyzin, 2020). We summarize these prior results as follow.

- *Kernel methods.* If $\sigma_*$ is a polynomial of degree $\deg(\sigma_*)$, then kernel methods using fixed feature maps require $n \gtrsim d^{\deg(\sigma_*)}$ samples to learn $f_*$ with low estimation error (Ghorbani et al., 2021; Donhauser et al., 2021).

- *CSQ algorithms.* For learners that make correlational queries to the target (i.e., in the form of $\mathbb{E}[\phi(\boldsymbol{x})y]$), such as one-pass SGD on the squared/correlation loss (Ben Arous et al., 2021; Damian et al., 2023) or single-step GD (Damian et al., 2022; Ba et al., 2023), prior works have established a sufficient sample complexity of $n \gtrsim d^{\Theta(\mathrm{ie}(\sigma_*))}$, where $\mathrm{ie}(\sigma_*) \leq \deg(\sigma_*)$ is the *information exponent* defined as the *lowest degree* in the Hermite

expansion of $\sigma_*$ (see Definition 2). This sample complexity improves upon kernel methods, and the exponential dependence on the IE aligns with the CSQ lower bound (Damian et al., 2022; Abbe et al., 2023).

- *SQ algorithms.* For learners that access full statistical queries (i.e., in the form of $\mathbb{E}[\phi(\boldsymbol{x}, y)]$), the complexity of single-index learning is governed by the *generative exponent* $\mathrm{ge}(\sigma_*) \leq \mathrm{ie}(\sigma_*)$ defined as the lowest possible information exponent after arbitrary $L^2$ label transformation (Damian et al., 2024); in particular, $\mathrm{ge}(\sigma_*) \leq 2$ for any polynomial link function. Recent works have shown that gradient-based training on *transformed labels* can achieve a sample complexity of $n \gtrsim d^{\Theta(\mathrm{ge}(\sigma_*))}$, which improves upon the CSQ rate (Dandi et al., 2024; Lee et al., 2024; Arnaboldi et al., 2024; Joshi et al., 2024).

The ICL mechanism in Oko et al. (2024b) operates as a kernel regression algorithm, which entails that the in-context sample complexity depends on $\deg(\sigma_*)$. We expect this dependency to improve with inference-time feature learning. Indeed, drawing from prior works showing that linear transformers can implement single-step GD in-context (Von Oswald et al., 2023; Ahn et al., 2023; Mahankali et al., 2023a; Zhang et al., 2023) – which corresponds to a CSQ-like algorithm – it is natural to conjecture that a sufficient sample size in our single-index setting scales as $r^{\Theta(\mathrm{ie}(\sigma_*))}$. That being said, when considering the effect of *nonlinear* softmax attention, transformers can also introduce nonlinear label transformations at inference time, hence potentially achieving a statistical efficiency that depends only on $\mathrm{ge}(\sigma_*)$ and surpasses the limitations of CSQ-based methods. This motivates us to investigate the following question regarding the ICL sample complexity.

**Q2:** *Can transformers with nonlinear self-attention implement inference-time algorithms that surpass the statistical barrier imposed by CSQ lower bounds?*

### 1.1 Our Contributions

We establish end-to-end optimization and statistical guarantees for the in-context learning of the Gaussian single-index function class using a transformer with softmax attention. Our main result is the following.

**Theorem 1** (informal). *Consider the learning of single-index polynomials $\sigma_*(\langle \boldsymbol{\beta}, \boldsymbol{x} \rangle)$ where the feature vector $\boldsymbol{\beta}$ is drawn from an $r$-dimensional subspace. If we optimize a single-layer transformer with softmax attention via gradient-based pretraining (Algorithm 1) using $T_{\mathrm{pt}} = \tilde{\Omega}(d^{\Theta(\mathrm{ie}(\sigma_*))})$ tasks, where for each task the prompt length satisfies $N_{\mathrm{pt}} = \tilde{\Omega}(d^{\Theta(\mathrm{ie}(\sigma_*))})$, then the model achieves $o_d(1)$ in-context prediction risk if the number of in-context examples satisfies $N_{\mathrm{test}} = \tilde{\Omega}(r^{\Theta(\mathrm{ge}(\sigma_*))})$, where $\mathrm{ge}(\sigma_*) \leq 2$ is the generative exponent of $\sigma_*$.*

We make the following remarks on the main theorem.

- While the pretraining sample complexity scales with the ambient dimension $d$, the required in-context sample size depends only on the support dimension of the target functions $r \leq d$. This aligns with the findings of Oko et al. (2024b), which show that transformers can adapt to low-dimensional structure of the target function class through pretraining, thereby enhancing inference efficiency. We note that while Oko et al. (2024b) restricts the analysis to the regime $r \lesssim \sqrt{d}$, our results applies to arbitrary $r \leq d$.

- The inference-time sample complexity of $r^{\Theta(\mathrm{ge}(\sigma_*))}$ is independent of both the degree and the information exponent of the link function $\sigma_*$; in other words, pretrained transformers can implement SQ algorithms in-context with statistical efficiency that surpasses both kernel methods and CSQ learners. Consequently, our ICL complexity improves upon Oko et al. (2024b) in all regimes.

- Our analysis explicitly shows that softmax self-attention can extract the inner product $\langle \boldsymbol{\beta}, \boldsymbol{x} \rangle$, where $\boldsymbol{\beta}$ varies across tasks, by computing the correlation between nonlinear transformations of both the input $\boldsymbol{x}$ and the label. This mechanism is analogous to the tensor partial trace algorithm studied in Damian et al. (2024); see Section 5 for detailed explanation.

On the technical level, we demonstrate that: $(i)$ the $\mathrm{Softmax}$ attention can be pretrained to compute $\langle \boldsymbol{\beta}, \boldsymbol{x} \rangle$ for each prompt, and $(ii)$ the nonlinearity in the attention reduces the information exponent of the link function $\sigma_*$, which is shown through a careful analysis of the nonlinear correlation computed by the attention scores. Mechanism $(i)$ enables inference-time feature learning, whereas mechanism $(ii)$ leads to an information exponent-free sample complexity.

### 1.2 Related Work

**In-context Learning of Functions.**   ICL abilities of linear transformers have been extensively studied in terms of both expressivity and optimization. They are shown to implement linear regression algorithms, including one gradient descent step (Von Oswald et al., 2023; Ahn et al., 2023; Mahankali et al., 2023a; Zhang et al., 2023), multi-step gradient descent (Ahn et al., 2023; Gatmiry et al., 2024), and sparse regression (Bai et al., 2023). (Zhang et al., 2024) showed that linear transformer can adapt to the prior mean of the coefficient vector. We note several related works on ICL of nonlinear transformers that were not discussed in the introduction: Huang et al. (2023); Chen et al. (2024) analyzed in-context linear regression of softmax transformers, including optimization analysis, while Li et al. (2024) investigated a classification problem (which is also essentially linear). Nichani et al. (2024) examined the ability of optimized softmax transformers to learn causal structures in-context. Bietti et al. (2023b) demonstrated that induction

head mechanisms emerge via associative memory obtained through gradient descent.

**Learning Functions with Sparse Feature.**   Other than single-index models, various function classes with sparse features have been studied, and neural networks' adaptivity to such features via gradient descent have been explored in standard (non-ICL) settings. These function classes include multi-index models (Damian et al., 2022; Abbe et al., 2022; 2023; Bietti et al., 2023a; Mousavi-Hosseini et al., 2023a), additive models (Oko et al., 2024a; Ren et al., 2025), parity functions (Barak et al., 2022; Suzuki et al., 2023; Glasgow, 2024). Kim & Suzuki (2024b) showed that chain-of-thought reasoning of transformers helps solve parity functions efficiently, albeit not in an ICL setting.

## 2 Preliminaries

**Notations.**   $\|\boldsymbol{v}\|$ denotes the $\ell_2$ norm of a vector $\boldsymbol{v}$. For matrix $\boldsymbol{A}$, we denote its $\ell_2$ operator norm and Frobenius norm as $\|\boldsymbol{A}\|_2$ and $\|\boldsymbol{A}\|_F$, respectively. Let $\mathrm{He}_i(z) = (-1)^i e^{\frac{z^2}{2}} \frac{\mathrm{d}^i}{\mathrm{d}z^i} e^{\frac{-z^2}{2}}$ be the degree-$i$ (probabilist's) Hermite polynomial. Given the Hermite expansion of a function $f$, i.e., $f(z) = \sum_{i \geq 0} \frac{c_i}{i!} \mathrm{He}_i(z)$, we define $\mathrm{H}(f, i) := c_i$ as its degree-$i$ coefficient. $\mathbb{S}^{d-1}$ denotes the unit sphere in $\mathbb{R}^d$. $\mathbb{I}(A)$ is the indicator function. $\mathrm{diag}(a_1, \ldots, a_n)$ represents the $n \times n$ diagonal matrix whose $(i, i)$-th element is $a_i$.

### 2.1 Preliminaries on In-Context Learning (ICL)

We consider the setting of learning functional relationships in-context as introduced in Akyürek et al. (2022); Garg et al. (2022). The input consists of $N$ labeled example pairs $(\boldsymbol{x}_i, y_i) \in \mathbb{R}^d \times \mathbb{R}$ for $i = 1, \ldots, N$, referred to as *context* or *in-context examples*, along with a *query* $\boldsymbol{x} \in \mathbb{R}^d$. The entire input sequence $(\boldsymbol{x}_1, y_1, \ldots, \boldsymbol{x}_N, y_N, \boldsymbol{x})$ is referred to a *prompt*, which we sometimes shorten to $(\boldsymbol{X}, \boldsymbol{y}, \boldsymbol{x})$ where $\boldsymbol{X} := \begin{bmatrix} \boldsymbol{x}_1 & \cdots & \boldsymbol{x}_N \end{bmatrix}$ and $\boldsymbol{y} := \begin{bmatrix} y_1 & \cdots & y_N \end{bmatrix}^\top$. The learner is tasked to predict the corresponding output $y \in \mathbb{R}$ for the query $\boldsymbol{x}$. The inputs $\boldsymbol{x}_1, \ldots, \boldsymbol{x}_N, \boldsymbol{x}$ are i.i.d. samples drawn from a specific distribution $\mathcal{D}_{\boldsymbol{x}}$, and the outputs $y_1, \ldots, y_N$ are generated as

$$y_i = f_*(\boldsymbol{x}_i) + \zeta_i \ (i = 1, \ldots, N),$$

where $f_* : \mathbb{R}^d \to \mathbb{R}$ is a function associated with each prompt, and $\zeta_1, \ldots, \zeta_N \sim \mathcal{D}_\zeta$ are i.i.d. noise. The target function $f_*$ varies across prompts and is drawn from $\mathcal{D}_{f_*}$ for each prompt. Our goal is to obtain a model $f(\boldsymbol{X}, \boldsymbol{y}, \boldsymbol{x}; \boldsymbol{\theta})$ with parameters $\boldsymbol{\theta}$, which takes the prompt as input and predicts the query output $f_*(\boldsymbol{x})$ *without updating any parameters during inference time*.

We divide learning into two stages: pretraining and inference. In the *pretraining stage*, the model is optimized on a training set $\{((\boldsymbol{x}_1^t, y_1^t, \ldots, \boldsymbol{x}_{N_{\mathrm{pt}}}^t, y_{N_{\mathrm{pt}}}^t, \boldsymbol{x}^t), y^t)\}_{t=1}^{T_{\mathrm{pt}}}$, which

includes $T_{\text{pt}}$ distinct tasks from $\mathcal{D}_{f_*}$. In the *inference stage*, we present a prompt $(\boldsymbol{x}_1, y_1, \ldots, \boldsymbol{x}_{N_{\text{test}}}, y_{N_{\text{test}}}, \boldsymbol{x})$ with context length $N_{\text{test}}$ to the pretrained model, which aims to predict the label $f_*(\boldsymbol{x})$. To evaluate the model's average performance at inference, we use *test ICL error* defined as

$$\mathcal{R}_{N_{\text{test}}}^{\text{ICL}}(\boldsymbol{\theta}) = \mathbb{E}_{\mathcal{D}_{\boldsymbol{x}}, \mathcal{D}_{f_*}, \mathcal{D}_\zeta}[|f(\boldsymbol{X}_{1:N_{\text{test}}}, \boldsymbol{y}_{1:N_{\text{test}}}, \boldsymbol{x}; \boldsymbol{\theta}) - y|].$$
(2.1)

Based on this setup, we emphasize that there are two distinct notions of "sample complexity" in ICL:

- The ***pretraining task (sample) complexity*** refers to the task (sample) size $T_{\text{pt}}(N_{\text{pt}})$ used by the pretraining algorithm to optimize the parameters $\boldsymbol{\theta}$.

- The ***inference sample complexity*** refers to the context length $N_{\text{test}}$ required at inference time to ensure small $\mathcal{R}_{N_{\text{test}}}^{\text{ICL}}$, characterizing the statistical complexity of the constructed in-context learning algorithm.

We consider the regime where $N_{\text{pt}}$, $T_{\text{pt}}$ and $N_{\text{test}}$ are polynomial in the dimensionality $d$, and track the dimension dependence in the sample complexity for both the pretraining and inference phase.

## 2.2 Single-index Models

For the class of target functions $f_*$, we consider Gaussian *single-index models* where the function value depends only on a single direction $\boldsymbol{\beta}$ in the $d$-dimensional input space. This setting reflects the scenario where learners have to capture the unknown feature vector $\boldsymbol{\beta}$ from high-dimensional input in $\mathbb{R}^d$, and hence has been extensively studied in feature learning theory literature (Bai & Lee, 2019; Ba et al., 2022; Bietti et al., 2022; Mousavi-Hosseini et al., 2023a; Mahankali et al., 2023b; Berthier et al., 2024).

**Definition 2.** *Let* $\boldsymbol{x} \sim \mathcal{N}(0, \boldsymbol{I}_d)$. *(Gaussian) single-index models refer to functions of the form*

$$f_* : \mathbb{R}^d \to \mathbb{R}, \ f_*(\boldsymbol{x}) = \sigma_*(\langle \boldsymbol{\beta}, \boldsymbol{x} \rangle),$$

*where* $\sigma_* : \mathbb{R} \to \mathbb{R}$ *is the link function and* $\boldsymbol{\beta}$ *is the feature vector. We assume* $\sigma_*$ *is a polynomial and satisfies* $\mathbb{E}_{z \sim \mathcal{N}(0,1)}[\sigma_*(z)] = 0$. *We define three key properties:*

1. $\deg(\sigma_*)$ *denotes the (polynomial) degree of* $\sigma_*$.

2. *The* information exponent *(Dudeja & Hsu, 2018; Ben Arous et al., 2021) is defined as*

$$\text{ie}(\sigma_*) := \min\{i \mid \text{H}(\sigma_*, i) \neq 0\}.$$

3. *The* generative exponent *(Damian et al., 2024) is*

$$\text{ge}(\sigma_*) := \min_{h \in L^2} \min\{i \mid \text{H}(h \circ \sigma_*, i) \neq 0\},$$

*where* $L^2$ *denotes the set of all* $L^2(P_Y)$*-measurable transformations from* $\mathbb{R}$ *to* $\mathbb{R}$ *for* $P_Y = \sigma_{*\#}\mathcal{N}(0,1)$ *where* $\sigma_{*\#}$ *is the push-forward by* $\sigma_*$.

Note that, by definition, $\text{ge}(\sigma_*) \leq \text{ie}(\sigma_*) \leq \deg(\sigma_*)$. It is known that for polynomial link functions, there exists a monomial transformation $\sigma_* \to \sigma_*^j$ that reduces the information exponent to 1 or 2.

**Lemma 3** (Lee et al. (2024), Proposition 6)**.** *It holds that*

$$\text{ge}(\sigma_*) = \begin{cases} 1 \ (\textit{if } \sigma_* \text{ is not even}) \\ 2 \ (\textit{if } \sigma_* \text{ is even}) \end{cases}.$$

*Moreover,* $\text{ge}(\sigma_*) = \min_{j \geq 1} \text{ie}(\sigma_*^j)$ *holds.*

**Sample complexity of learning** $f_*$**.** The sample complexity of various learning algorithms are dictated by the quantities in Definition 2. Linear/kernel models on fixed basis require $d^{\deg(\sigma_*)}$ samples to learn $f_*$ (Ghorbani et al., 2021; Donhauser et al., 2021). Online SGD (Ben Arous et al., 2021; Damian et al., 2023) or single-step gradient descent (Damian et al., 2022; Ba et al., 2023; Dandi et al., 2023) on two-layer neural network $\sum_{j=1}^m a_j \sigma(\langle \boldsymbol{w}_j, \boldsymbol{x} \rangle + b_j)$, which can adaptively select feature maps $\{\sigma(\langle \boldsymbol{w}_j, \cdot \rangle + b_j)\}_{j=1}^m$, achieves a better sample complexity of $\tilde{O}(d^{\Theta(\text{ie}(\sigma_*))})$ — such dependence on the information exponent also appears in correlational statistical query (CSQ) lower bounds (Damian et al., 2022). Recent works have shown that data reuse (Arnaboldi et al., 2024; Lee et al., 2024) or loss modification (Joshi et al., 2024) can introduce nonlinear label transformations such that the "effective" information exponent is reduced to $\text{ge}(\sigma_*)$. Consequently, the sample complexity can be improved to $\tilde{O}(d^{\text{ge}(\sigma_*)-1 \vee 1})$ which matches the SQ lower bound $\Omega(d)$ (for polynomial $\sigma_*$) up to polylogarithmic factors.

**ICL sample complexity.** For the ICL setting, (Oko et al., 2024b) studied the single-index function class where $\boldsymbol{\beta}$ is uniformly drawn from the unit sphere in an $r(\lesssim \sqrt{d})$-dimensional linear subspace of $\mathbb{R}^d$. The authors derived a pretraining task and sample complexity of $\tilde{O}(d^{\Theta(\text{ie}(\sigma_*))})$ and inference-time sample complexity of $\tilde{O}(r^{\Theta(\deg(\sigma_*))})$. Note that this inference complexity matches that of *kernel methods* on the $r$-dimensional subspace. Our focus (see **Q1** and **Q2** in the introduction) is to investigate whether transformers can be pretrained to achieve an inference complexity that surpasses kernel methods or CSQ algorithms.

## 3 Problem Setting

### 3.1 Task Setup

We define the task distribution as follows.

**Assumption 4.** *Let* $\tau = O_d(1)$ *be the noise level. A prompt* $(\boldsymbol{x}_1, y_1, \ldots, \boldsymbol{x}_N, y_N, \boldsymbol{x})$ *is generated as:*

$$\begin{aligned} \boldsymbol{x}_1, \ldots, \boldsymbol{x}_N, \boldsymbol{x} &\sim \mathcal{D}_{\boldsymbol{x}} = \mathcal{N}(0, \boldsymbol{I}_d), \\ y_i &= f_*(\boldsymbol{x}_i) + \zeta_i, \quad \zeta_i \sim \text{Unif}(\{-\tau, \tau\}), \\ f_*(\boldsymbol{x}_i) &= \sigma_*(\langle \boldsymbol{\beta}, \boldsymbol{x}_i \rangle). \end{aligned}$$

*For the true function $f_*$, we make the following assumptions:*

- $\sigma_*$ *is a polynomial that remains fixed across tasks. We assume the normalization conditions $\mathbb{E}_{z\sim\mathcal{N}(0,1)}[\sigma_*(z)] = 0$ and $\mathbb{E}_{z\sim\mathcal{N}(0,1)}[\sigma_*^2(z)] = 1$.*

- $\mathcal{S}_r$ *denotes an $r$-dimensional linear subspace of $\mathbb{R}^d$ for $r \leq d$. For each prompt, $\boldsymbol{\beta}$ is drawn uniformly from the unit sphere $\mathrm{Supp}(\boldsymbol{\beta}) := \{\boldsymbol{\beta} \mid \boldsymbol{\beta} \in \mathcal{S}_r, \|\boldsymbol{\beta}\| = 1\}$ in $\mathcal{S}_r$.*

**Remark 5.** *In Assumption 4, a feature vector $\boldsymbol{\beta}$ is newly drawn for each task. We allow for scenarios where $\boldsymbol{\beta}$ has limited ($r$-dimensional) support in $\mathbb{R}^d$, and we will show that the inference-time sample complexity adapts to this support. Note that unlike (Oko et al., 2024b) which established similar adaptivity, we do not assume low-dimensionality ($r = o(d)$); instead our result holds for any $1 \leq r \leq d$.*

### 3.2 Student Model and Pretraining Algorithm

**Student Model.** We train a single-layer transformer (Vaswani et al., 2017) with SoftMax attention. For a prompt $(\boldsymbol{x}_1, y_1, \ldots, \boldsymbol{x}_N, y_N, \boldsymbol{x})$, we construct the embedding $\boldsymbol{E} \in \mathbb{R}^{(d+1)\times(N+1)}$ as

$$\boldsymbol{E} = \begin{bmatrix} \boldsymbol{x}_1 & \cdots & \boldsymbol{x}_N & \boldsymbol{x} \\ y_1 & \cdots & y_N & 1 \end{bmatrix} \in \mathbb{R}^{(d+1)\times(N+1)}.$$

We first convert $\boldsymbol{E}$ to $\mathrm{Attn}(\boldsymbol{E})$ through the Softmax attention layer defined as

$$\begin{aligned}&\mathrm{Attn}(\boldsymbol{E})\\&=\boldsymbol{W}^V \boldsymbol{E} \cdot \mathrm{Softmax}(\mathrm{Mask}(\rho^{-1} \cdot (\boldsymbol{W}^K \boldsymbol{E})^\top \boldsymbol{W}^Q \boldsymbol{E})),\end{aligned}$$

where $\rho > 0$ is the temperature, and $\boldsymbol{W}^K, \boldsymbol{W}^Q, \boldsymbol{W}^V \in \mathbb{R}^{(d+1)\times(d+1)}$ are attention parameters. The Softmax function is applied to each column vector, while the Mask function replaces all entries in the final row of $\rho^{-1} \cdot (\boldsymbol{W}^K \boldsymbol{E})^\top \boldsymbol{W}^Q \boldsymbol{E} \in \mathbb{R}^{(N+1)\times(N+1)}$ with $-\infty$.

**Remark 6.** *Observe that the first $N$ rows of $\rho^{-1} \cdot (\boldsymbol{W}^K \boldsymbol{E})^\top \boldsymbol{W}^Q \boldsymbol{E}$ represent the correlations between $\boldsymbol{x}$ and $\boldsymbol{x}_i$ for $i \in [N]$, while the last row corresponds to the correlation between $\boldsymbol{x}$ and itself. Consequently, the last row is (typically) approximately $\sqrt{d}$ times larger in magnitude than the others. Thus, we apply masking to avoid the Softmax attention from focusing on the uninformative final row.*

Then, we apply a position-wise multi-layer perceptron (MLP) with activation function $\sigma(\cdot)$ and parameters $\boldsymbol{W}^F \in \mathbb{R}^{m\times(d+1)}$, $\boldsymbol{b} \in \mathbb{R}^m$, and $\boldsymbol{a} \in \mathbb{R}^m$, where $m$ is the network width. Throughout this paper, we set $\sigma(z) = \max\{z, 0\}$ (ReLU activation). We use the $(N+1)$-th element of the MLP output as the prediction for the query output. Overall, the model's prediction $f_{\mathrm{TF}}$ is given by

$$\begin{aligned}&f_{\mathrm{TF}}(\boldsymbol{X}, \boldsymbol{y}, \boldsymbol{x}; \boldsymbol{W}^K, \boldsymbol{W}^Q, \boldsymbol{W}^V, \boldsymbol{W}^F, \boldsymbol{a}, \boldsymbol{b})\\&=\mathrm{MLP} \circ \mathrm{Attn}(\boldsymbol{E})_{:,N+1}\end{aligned}$$

$$=\boldsymbol{a}^\top \sigma\big(\boldsymbol{W}^F \mathrm{Attn}(\boldsymbol{E})_{:,N+1} + \boldsymbol{b}\big), \qquad (3.1)$$

where $\sigma$ is applied entry-wise.

We further introduce some simplifications: we merge parameters as $\boldsymbol{W}^{KQ} := (\boldsymbol{W}^K)^\top \boldsymbol{W}^Q \in \mathbb{R}^{(d+1)\times(d+1)}$ and $\boldsymbol{W}^{FV} := \boldsymbol{W}^F \boldsymbol{W}^V \in \mathbb{R}^{m\times(d+1)}$, and write

$$\boldsymbol{W}^{KQ} = \begin{bmatrix} \boldsymbol{\Gamma} & \boldsymbol{0}_{d\times 1} \\ \boldsymbol{0}_{1\times d} & 1 \end{bmatrix}, \ \boldsymbol{W}^{FV} = \begin{bmatrix} \boldsymbol{O}_{m\times d} & \boldsymbol{v} \end{bmatrix},$$

where $\boldsymbol{\Gamma} \in \mathbb{R}^{d\times d}$ and $\boldsymbol{v} \in \mathbb{R}^{m\times 1}$. Similar simplifications (zeroing out specific sub-matrices) have been adopted in recent theoretical works on the expressivity and optimization of transformers (Ahn et al., 2023; Mahankali et al., 2023a; Wu et al., 2023; Kim & Suzuki, 2024a).

Consequently, $f_{\mathrm{TF}}$ can be rewritten as

$$\begin{aligned}&f_{\mathrm{TF}}(\boldsymbol{X}, \boldsymbol{y}, \boldsymbol{x}; \boldsymbol{\Gamma}, \boldsymbol{v}, \boldsymbol{b}, \boldsymbol{a})\\&=\sum_{j=1}^m a_j \sigma\left(v_j \frac{\sum_{i=1}^N y_i \mathrm{e}^{y_i/\rho} \mathrm{e}^{\boldsymbol{x}_i^\top \boldsymbol{\Gamma} \boldsymbol{x}/\rho}}{\sum_{i=1}^N \mathrm{e}^{y_i/\rho} \mathrm{e}^{\boldsymbol{x}_i^\top \boldsymbol{\Gamma} \boldsymbol{x}/\rho}} + b_j\right), \quad (3.2)\end{aligned}$$

with the new parameters $(\boldsymbol{\Gamma}, \boldsymbol{v}, \boldsymbol{b}, \boldsymbol{a})$: see Appendix A for the derivation. Hereafter, we refer to $\boldsymbol{\Gamma}$ as the *attention matrix* and consider $(\boldsymbol{v}, \boldsymbol{b}, \boldsymbol{a})$ as the *MLP parameters*.

**Pretraining Algorithm.** In Algorithm 1, we specify our gradient-based pretraining algorithm to optimize the parameters $(\boldsymbol{\Gamma}, \boldsymbol{v}, \boldsymbol{b}, \boldsymbol{a})$. The algorithm uses $T_{\mathrm{pt}}$ pretraining tasks $\{(\boldsymbol{X}_{1:N_{\mathrm{pt}}}^t, \boldsymbol{y}_{1:N_{\mathrm{pt}}}^t, \boldsymbol{x}^t, y^t)\}_{t=1}^{T_{\mathrm{pt}}}$ and optimizes the ($\ell_2$ regularized) squared loss

$$\begin{aligned}&L_i(\boldsymbol{\Gamma}, \boldsymbol{v}, \boldsymbol{b}, \boldsymbol{a})\\&=\frac{1}{2T_i} \sum_{t=T_{i-1}+1}^{T_{i-1}+T_i} (f_{\mathrm{TF}}(\boldsymbol{X}_{1:N_{\mathrm{pt}}}^t, \boldsymbol{y}_{1:N_{\mathrm{pt}}}^t, \boldsymbol{x}^t; \boldsymbol{\Gamma}, \boldsymbol{v}, \boldsymbol{b}, \boldsymbol{a}) - y^t)^2\end{aligned}$$

for $i = 1, 2$; here $T_0 = 0$ and $T_1 + T_2 = T_{\mathrm{pt}}$. The training procedure is divided into two stages.

- In Stage I, we perform a single gradient descent step on the attention matrix $\boldsymbol{\Gamma}$ using the $\ell_2$-regularized squared loss $L_1(\boldsymbol{\Gamma}, \boldsymbol{v}, \boldsymbol{b}, \boldsymbol{a}) + \frac{\lambda_1}{2}\|\boldsymbol{\Gamma}\|_2^2$ computed over $T_1$ independent pretraining tasks. This optimization problem is nonconvex with respect to $\boldsymbol{\Gamma}$. This single gradient descent update originates from theoretical studies on feature learning Ba et al. (2022); Damian et al. (2022), where it is shown that the first gradient update captures low-dimensional features of the true function. More recent results have highlighted that one gradient step in transformers can encode information for implementing algorithms such as kernel methods on low-dimensional subspaces (Oko et al., 2024b) or induction heads (Bietti et al., 2023b).

- In Stage II, we optimize the MLP parameters. Specifically, we randomize the weight $\boldsymbol{v}$ and bias $\boldsymbol{b}$, and perform ridge

**Algorithm 1** Gradient-based training of transformer

**Input** : Learning rate $\eta_1, \eta_2$, regularization rate $\lambda_1, \lambda_2$, initialization scale $\alpha$, temperature $\rho$

1 **Initialize** $\mathbf{\Gamma}(0) \sim \mathbf{I}_d/\sqrt{d}, \mathbf{v}(0) \sim \text{Unif}(\{\pm 1\}^m), \mathbf{b}(0) = \mathbf{0}_m, \mathbf{a}(0) = \alpha \mathbf{1}_m$

2 **Stage I: Gradient descent on Attention Matrix**

3     $\mathbf{\Gamma}^* \leftarrow \mathbf{\Gamma}(0) - \eta_1 \nabla_{\mathbf{\Gamma}}(L_1(\mathbf{\Gamma}(0), \mathbf{v}(0), \mathbf{b}(0), \mathbf{a}(0)) + \frac{\lambda_1}{2}\|\mathbf{\Gamma}\|_F^2)$

4 **Stage II: Optimization of MLP Layer**

5     **Initialize** $b_j^* \sim \text{Unif}([-1, 1]), \mathbf{v}^* = \mathbf{v}(0)$.

     $\mathbf{a}^* \leftarrow \arg\min_{\mathbf{a}} L_2(\mathbf{\Gamma}^*, \mathbf{v}^*, \mathbf{b}^*, \mathbf{a}) + \frac{\lambda_2}{2}\|\mathbf{a}\|^2$.

**Output :** Prediction $f_{\text{TF}}(\mathbf{X}, \mathbf{y}, \mathbf{x}; \mathbf{\Gamma}^*, \mathbf{v}^*, \mathbf{b}^*, \mathbf{a}^*)$.

regression on the output weights $\mathbf{a}$. Note that the only role of the MLP layer is to fit the polynomial link function $\sigma_*$. Since our primary focus is the optimization of attention matrix, we adopt this simpler algorithm for MLP which admits a closed-form solution due to the convex objective.

# 4 Main Result: Transformer Performs Inference-Time Feature Learning

## 4.1 Main Theorem

Now we present our main theorem, which establishes an optimization guarantee for Algorithm 1 and a bound on the inference-time sample complexity.

**Theorem 1.** *Let* $f_{\text{TF}}(\mathbf{X}, \mathbf{y}, \mathbf{x}; \mathbf{\Gamma}^*, \mathbf{v}^*, \mathbf{b}^*, \mathbf{a}^*)$ *be a transformer pretrained via Algorithm 1 with MLP width* $m = \tilde{\Omega}(r^{2\text{ge}(\sigma_*)})$ *and initialization scale* $\alpha = O(m^{-1}r^{-1}d^{-(\text{ie}(\sigma_*)+1)/2}\log^{-C_\alpha} d)$ *for constant* $C_\alpha$. *Then, there exist hyperparameters* $\lambda_1, \lambda_2, \eta_1,$ *and* $\eta_2$ *such that the following hold with probability at least 0.99 over the training data and random initialization:*

1. *(Optimization) Suppose that the pretraining task size and the pretraining context length satisfy* $T_1, N_{\text{pt}} = \tilde{\Omega}(r^2 d^{\text{ie}(\sigma_*)+2})$ *and* $T_2 = \tilde{\Omega}(r^{3\text{ge}(\sigma_*)/2})$ *(i.e.,* $T_{\text{pt}} = \tilde{\Omega}(r^2 d^{\text{ie}(\sigma_*)+2} \vee r^{3\text{ge}(\sigma_*)/2})$*). Then, Algorithm 1 yields parameters* $(\mathbf{\Gamma}^*, \mathbf{v}^*, \mathbf{b}^*, \mathbf{a}^*)$ *such that the empirical loss*

$$\frac{1}{T_2}\sum_{t=T_1+1}^{T_1+T_2} \left|y^t - f_{\text{TF}}(\mathbf{X}^t, \mathbf{y}^t, \mathbf{x}^t, \mathbf{\Gamma}^*, \mathbf{v}^*, \mathbf{b}^*, \mathbf{a}^*)\right| = o_d(1).$$

*Moreover, we have* $\|\mathbf{a}^*\| = \tilde{O}(r^{3\text{ge}(\sigma_*)/4}m^{-1/2})$.

2. *(Inference-time sample complexity) At inference time, if the (in-context )test prompt length satisfies*

$$N_{\text{test}} = \tilde{\Omega}(r^{3\text{ge}(\sigma_*)/2}),$$

*then the ICL error* (2.1) $\mathcal{R}_{N_{\text{test}}}^{\text{ICL}}(\mathbf{\Gamma}^*, \mathbf{v}^*, \mathbf{b}^*, \mathbf{a}^*) = o_d(1)$.

## 4.2 Implications of the Main Theorem

Our main theorem suggests the statistical capabilities and optimization properties of transformers (see Table 4.1).

**(i) Inference-Time Feature Learning.** First, consider the standard case where $r = d$: the inference-time sample complexity is $d^{\Theta(\text{ge}(\sigma_*))}$. Since $\text{ge}(\sigma_*) \leq \deg(\sigma_*)$ always holds, our derived inference-time sample complexity surpasses the lower bound $d^{\Theta(\deg(\sigma_*))}$ for *non-adaptive algorithms* based on fixed basis such as kernel methods. This implies that pretrained transformers can implement an algorithm capable of *inference-time feature learning*, that is, they can adapt to the feature vector $\boldsymbol{\beta}$ of single-index functions, which varies across test tasks. Beyond merely establishing the inference-time sample complexity, we verify in Section 5 that the attention mechanism indeed captures the feature vector $\boldsymbol{\beta}$.

Moreover, when $\boldsymbol{\beta}$ is restricted to an $r \leq d$-dimensional subspace in $\mathbb{R}^d$ (see Assumption 4), the inference-time sample complexity depends only on $r$. In other words, transformers can significantly reduce the inference-time sample complexity when the support of $\boldsymbol{\beta}$ is low-dimensional, i.e., $r \ll d$.

**(ii) Beyond CSQ.** Our result further shows that the inference-time sample complexity can surpass the sample complexity lower bound $d^{\text{ie}(\sigma_*)/2\vee 1}$ for CSQ algorithms. For instance, the sample complexity upper bounds for one-step gradient descent (Ba et al., 2023) and one-pass SGD (Ben Arous et al., 2021) on two-layer neural networks all depends exponentially on the information exponent. In contrast, in our analysis, the single-layer transformer implements an in-context algorithm that is statistically more efficient than CSQ learners, due to the nonlinear attention.

**(iii) Optimization Guarantee.** While recent works have revealed that nonlinear transformers can implement rich algorithms by constructing specific parameter configurations, our result provides an optimization guarantee for pretraining along with the pretraining task/sample complexity of $d^{\Theta(\text{ie}(\sigma_*))}$ for $T_{\text{pt}}$ and $N_{\text{pt}}$ (see 1. in Theorem 1). We establish this by proving that a single gradient descent step can construct an attention matrix that approximates $\mathbb{E}[\boldsymbol{\beta}\boldsymbol{\beta}^\top]$, which enables inference-time feature learning, and that the MLP layer can successfully fit the link function $\sigma_*$.

**Remark 7.** *For simplicity of analysis, we assume that the link function* $\sigma_*$ *is fixed across tasks. Task-specific link functions may be accommodated by adding an additional attention layer to fit the link function adaptively from the test prompt, as done in (Oko et al., 2024b). An alternative could be to fit the MLP parameter* $\mathbf{a}$ *using each test sample: such* test-time training *constitutes a relatively lightweight optimization, where it only involves training the MLP layer, since the inner attention layers have already extracted the relevant feature information* $\langle \boldsymbol{\beta}, \mathbf{x} \rangle$.

| Regression on test prompt | | |
| :---: | :---: | :---: |
| Kernel | CSQ | SQ |
| $d^{\Theta(\deg(\sigma_*))}$ | $d^{\Theta(\mathrm{ie}(\sigma_*))}$ | $d^{\Theta(\mathrm{ge}(\sigma_*))}$ |
| **In-context learning** | | |
| (Oko et al., 2024b) | | This work |
| Pretraining: $d^{\Theta(\mathrm{ie}(\sigma_*))}$ | | Pretraining: $d^{\Theta(\mathrm{ie}(\sigma_*))}$ |
| Inference: $r^{\Theta(\deg(\sigma_*))}$ [*1] | | Inference: $r^{\Theta(\mathrm{ge}(\sigma_*))}$ [*2] |

*Table 4.1.* Comparison of (inference-time) sample complexity. Here, $r$ denotes the dimensionality of the support from which the task vector $\boldsymbol{\beta}$ is drawn (See Assumption 4). [*1] is only applicable when $r \lesssim \sqrt{d}$, whereas [*2] holds for any $r \leq d$.

### 4.3 Comparison Against Prior Work

Most recent studies on single-index models have focused on the standard supervised regression setting rather than in-context learning, with the exception of (Oko et al., 2024b) which investigates the ICL complexity of single-index learning under a low-dimensional assumption. Here we highlight the distinction from the work; see Table 4.1 for the comparison of relevant sample complexities.

**(i) Improved ICL sample complexity.** In (Oko et al., 2024b), the derived inference-time sample complexity is $O(r^{\Theta(\deg(\sigma_*))})$, which corresponds to the complexity of non-adaptive methods (for instance, kernel methods) on an $r$-dimensional subspace. In contrast, we establish an improved sample complexity of $O(r^{\Theta(\mathrm{ge}(\sigma_*))})$.

**(ii) Removal of low-dimensional assumption.** (Oko et al., 2024b) assumed that the dimensionality $r$ of pretraining task distribution is much smaller than the ambient dimensionality $d$, whereas our analysis do not rely on this assumption.

**(iii) Technical innovations.** To derive the improved inference-time sample complexity, we demonstrate that the nonlinear transformation applied to the output label $\{y_i\}$ reduces the information exponent of the link function $\sigma_*$ to its generative exponent. This requires a careful tracking of the corresponding Hermite coefficients through a SoftMax attention block, which constitutes one of the most technically challenging parts of the analysis. This goes beyond (Oko et al., 2024a) where the authors showed that a linear attention computes correlations on the raw label $\{y_i\}$ and a fixed kernel basis $\{\phi(\boldsymbol{x}_i)\}$ (constructed by a fixed MLP block). See the next section for the technical overview.

## 5 Proof Overview

We provide a proof sketch for our main Theorem 1, where the main goal is to show that the self-attention mechanism provably extracts the feature vector $\boldsymbol{\beta}$ for each prompt.

The key idea is that the exponential transformation in the Softmax attention module leads to both $(i)$ inference-time feature learning capability, and $(ii)$ statistical efficiency that surpasses CSQ algorithms. Observe that the term $\mathrm{e}^{y_i/\rho}\mathrm{e}^{\boldsymbol{x}_i^\top \boldsymbol{\Gamma} \boldsymbol{x}/\rho}$ appears in the architecture (3.2), where $\mathrm{e}^{\boldsymbol{x}_i^\top \boldsymbol{\Gamma} \boldsymbol{x}/\rho}$ generates a series of polynomials in $\boldsymbol{x}$ via a Hermite expansion, which enables detection of the feature direction $\boldsymbol{\beta}$ from the nonlinearly transformed signal $\mathrm{e}^{y_i/\rho}$, yielding $(i)$. On the other hand, the factor $\mathrm{e}^{y_i/\rho}$ reduces the information exponent of $\sigma_*$ to its generative exponent $\mathrm{ge}(\sigma_*)$ through the exponential transformation $\mathrm{e}^{\cdot/\rho}$, yielding $(ii)$. Hereafter, we outline our analysis step by step.

### 5.1 Gradient Descent on Attention Matrix

Assuming that $f_{\mathrm{TF}}$ is sufficiently small at initialization, we can approximate the population (expected) gradient by the population correlational gradient $\mathbb{E}[y\nabla_{\boldsymbol{\Gamma}} f_{\mathrm{TF}}]$. We show that the magnitude of this correlation is $O(d^{-(\mathrm{ie}(\sigma_*)-1)/2})$. A standard matrix concentration argument then yields the required *pretraining* task and sample complexity of $d^{\Theta(\mathrm{ie}(\sigma_*))}$, ensuring that the correlational signal is not hidden within the noise due to finite sample size.

Furthermore, we demonstrate that the single-step gradient on $\boldsymbol{\Gamma}$ is approximately proportional to $\mathbb{E}[\boldsymbol{\beta}\boldsymbol{\beta}^\top]$. This implies that the attention matrix acquires information about the $r$-dimensional support of $\boldsymbol{\beta}$ via pretraining, leading to an inference-time sample complexity independent of $d$.

### 5.2 Feature Extraction by Softmax Attention

Next we show that the pretrained attention module can extract the feature information $\langle \boldsymbol{\beta}, \boldsymbol{x} \rangle$ directly from observing the test prompt. Recall that the model output is given by $f_{\mathrm{TF}} = \sum_{j=1}^m a_j \sigma(v_j g(\boldsymbol{X}, \boldsymbol{y}, \boldsymbol{x}; \boldsymbol{\Gamma}^*) + b_j)$, where $g(\boldsymbol{X}, \boldsymbol{y}, \boldsymbol{x}; \boldsymbol{\Gamma}) := \frac{N^{-1}\sum_{i=1}^N y_i \mathrm{e}^{y_i/\rho}\mathrm{e}^{\boldsymbol{x}_i^\top \boldsymbol{\Gamma} \boldsymbol{x}/\rho}}{N^{-1}\sum_{i=1}^N \mathrm{e}^{y_i/\rho}\mathrm{e}^{\boldsymbol{x}_i^\top \boldsymbol{\Gamma} \boldsymbol{x}/\rho}}$ is the attention output. We establish the following proposition, which demonstrates that $g$ contains the feature information:

**Proposition 8** (Informal)**.** *Suppose $N_{\mathrm{test}} = \tilde{\Omega}(r^{3\mathrm{ge}(\sigma_*)/2})$. For $\boldsymbol{\Gamma}^*$ optimized in Algorithm 1, we have*

$$g(\boldsymbol{X}, \boldsymbol{y}, \boldsymbol{x}; \boldsymbol{\Gamma}^*) \simeq C_a + C_b \left( \frac{\langle \boldsymbol{x}, \boldsymbol{\beta} \rangle}{\sqrt{r}} \right)^{\mathrm{ge}(\sigma_*)}$$

*with high probability over the prompt, where $C_a$ and $C_b$ are $\Theta(1/\mathrm{poly}\log d)$ quantities independent of the data.*

We see that the nonlinearity introduced by the Softmax attention plays a key role in deriving the proposition above.

**Warm-Up: Linear Attention Module.** First consider the counterpart of $g$ with a linear attention module $g_{\mathrm{lin}}(\boldsymbol{X}, \boldsymbol{y}, \boldsymbol{x}; \boldsymbol{\Gamma}) := N^{-1}\sum_{i=1}^N y_i(y_i + \boldsymbol{x}_i^\top \boldsymbol{\Gamma} \boldsymbol{x})$. If the context length $N$ is sufficiently large, the second term concentrates around $\mathbb{E}[y\boldsymbol{x}^\top]\boldsymbol{\Gamma}\boldsymbol{x}$. Moreover, it is known that if $\mathbb{E}_{z\sim\mathcal{N}(0,1)}[z\sigma_*(z)] \neq 0$, or equivalently, $\mathrm{ie}(\sigma_*) = 1$,

then $\mathbb{E}[y\boldsymbol{x}^\top] = \boldsymbol{\beta}^\top$ is satisfied. Overall, the attention module successfully computes $\boldsymbol{\beta}^\top\boldsymbol{\Gamma}\boldsymbol{x} \simeq \boldsymbol{\beta}^\top\boldsymbol{x}$ in-context, if $\boldsymbol{\Gamma}$ is (nearly) spanned by $\boldsymbol{\beta}$. However, if $\mathrm{ie}(\sigma_*) > 1$, then $\mathbb{E}[y\boldsymbol{x}^\top] = \boldsymbol{0}$ and the direction $\boldsymbol{\beta}$ is not identified. Thus we must exploit the nonlinearity to handle high-information-exponent scenarios.

**Exploiting Softmax.** To consider the nonlinear attention, we first show that the exponential transformation can reduce the information exponent of $\sigma_*$ to its generative exponent. Specifically, we show the following:

**Lemma 9** (Informal). *The information exponent of $\exp(\bar{\sigma}_*)$ and $\bar{\sigma}_* \exp(\bar{\sigma}_*)$ is equal to $\mathrm{ge}(\sigma_*)$, where $\bar{\sigma}_*$ is a clipped version of $\sigma_*/\rho$ (see Appendix B).*

Based on this property, we show that Softmax attention can compute $\langle\boldsymbol{x}, \boldsymbol{\beta}\rangle$ in-context for any polynomial link $\sigma_*$; consider the denominator $N^{-1}\sum_{i=1}^N \mathrm{e}^{y_i/\rho}\mathrm{e}^{\boldsymbol{x}_i^\top\boldsymbol{\Gamma}\boldsymbol{x}/\rho}$ of $g$. For $N$ sufficiently large, the quantity approximates

$$\mathbb{E}_{\boldsymbol{x}_1\sim\mathcal{N}(0,\boldsymbol{I}_d)}\left[\exp\bar{\sigma}_*(\langle\boldsymbol{\beta},\boldsymbol{x}_1/\rho\rangle)\exp\left(\langle\boldsymbol{\Gamma}\boldsymbol{x},\boldsymbol{x}_1\rangle/\rho\right)\right].$$

Now we consider Hermite expansions of $\exp\bar{\sigma}_*(\langle\boldsymbol{\beta},\boldsymbol{x}_1/\rho\rangle)$ and $\exp\left(\langle\boldsymbol{\Gamma}\boldsymbol{x},\boldsymbol{x}_1\rangle/\rho\right)$, and apply the property $\mathbb{E}_{\boldsymbol{x}_1}[\mathrm{He}_i(\langle\boldsymbol{w},\boldsymbol{x}_1\rangle)\mathrm{He}_j(\langle\boldsymbol{w}',\boldsymbol{x}_1\rangle)] \propto \mathbb{I}(i=j)\langle\boldsymbol{w},\boldsymbol{w}'\rangle$. Fortunately, $\exp\left(\langle\boldsymbol{\Gamma}\boldsymbol{x},\boldsymbol{x}_1\rangle/\rho\right)$ contains all degrees of Hermite components, and from the lemma above, $\exp\bar{\sigma}_*(\langle\boldsymbol{\beta},\boldsymbol{x}_1/\rho\rangle)$ has a nonzero coefficient for $\mathrm{He}_{\mathrm{ge}(\sigma_*)}$. Thus, the attention module computes $\langle\boldsymbol{\beta},\boldsymbol{x}\rangle^{\mathrm{ge}(\sigma_*)}$ if $\boldsymbol{\Gamma}$ is nearly spanned by $\boldsymbol{\beta}$. Crucially, since the correlational signal strength depends only on the generative exponent $\mathrm{ge}(\sigma_*)$, the required context length $N$ does not scale with $\mathrm{ie}(\sigma_*)$.

We also show that the numerator of $g$ retains this feature information. The remaining technical challenge is to ensure that the information is not lost due to the correlation between the denominator and numerator: we tackle this through a careful order analysis. See Appendix B for details.

### 5.3 Fitting the MLP Layer and Test Error Analysis

To complete the optimization analysis, we show that the MLP layer can fit the unknown link function via ridge regression (line 5 in Algorithm 1). First, we construct an MLP parameter $\boldsymbol{a}'$ such that

$$\frac{1}{T_2}\sum_{t=T_1+1}^{T_1+T_2}\left(y^t - f_{\mathrm{TF}}(\boldsymbol{X}^t,\boldsymbol{y}^t,\boldsymbol{x}^t,\boldsymbol{\Gamma}^*,\boldsymbol{v}^*,\boldsymbol{b}^*,\boldsymbol{a}')\right)^2$$

is sufficiently small. Let $\ell(\boldsymbol{a}')$ be the quantity above: thanks to the equivalence between the $\ell_2$-regularized convex optimization and its norm-constrained counterpart, we can show that there exists a ridge parameter $\lambda_2$ such that $\ell(\boldsymbol{a}^*) \leq \ell(\boldsymbol{a}')$ and $\|\boldsymbol{a}^*\| \leq \|\boldsymbol{a}'\|$ hold, where $\boldsymbol{a}^*$ is the parameter obtained by solving ridge regression.

Finally, we bound the discrepancy between the training and test errors via the Rademacher complexity of the class of transformers, leveraging a norm bound on $\|\boldsymbol{a}^*\|$ derived from that of $\|\boldsymbol{a}'\|$. See Appendices D and E for details.

## 6 Synthetic Experiment

We conduct numerical experiments on synthetic data to compare the in-context learning algorithm implemented by nonlinear transformers against non-adaptive kernel methods. We train a 6-layer GPT-2 model (Radford et al., 2019) to learn the Gaussian single-index task. See Appendix G for detailed settings of the experiments.

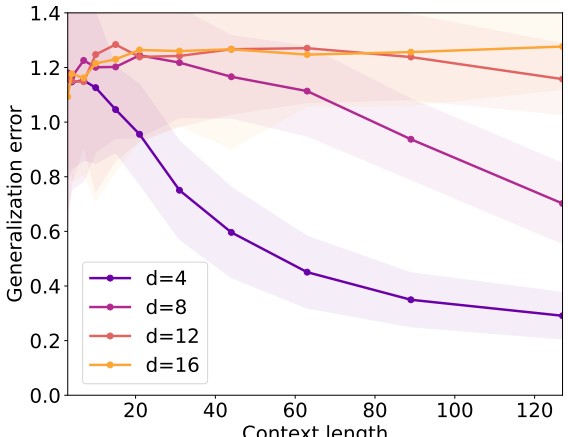

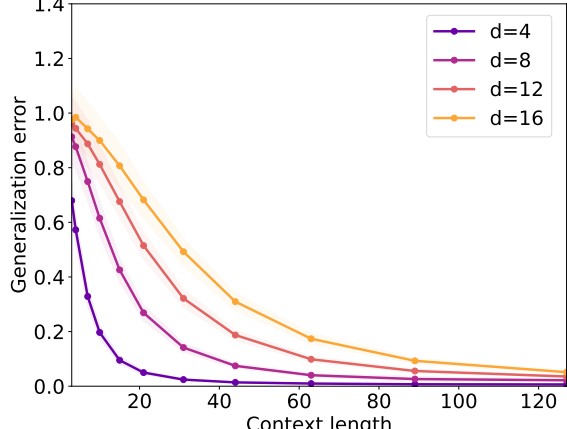

*Figure 6.1.* Generalization error versus test prompt length for kernel ridge regression (Top) and pretrained GPT-2 model (Bottom).

### 6.1 Comparison with Non-Adaptive Algorithm

For each test task $t$, we generate data as $\boldsymbol{x}_1^t,\ldots,\boldsymbol{x}_{N_{\mathrm{test}}}^t, \boldsymbol{x} \sim \mathcal{N}(0,\boldsymbol{I}_d), \boldsymbol{\beta}^t \sim \mathrm{Unif}(\mathbb{S}^{d-1})$ (i.e., $r = d$),with $y_i^t = \sigma_*(\langle\boldsymbol{\beta}^t,\boldsymbol{x}_i^t\rangle) = \mathrm{He}_3(\langle\boldsymbol{\beta}^t,\boldsymbol{x}_i^t\rangle)$ for $i \in [N]$. We compare the performance of two approaches:

- **In-context learning** using a 6-layers GPT-2 model, configured as in (Garg et al., 2022; Oko et al., 2024b). We

pretrain the GPT-2 model using on synthetic datasets $\{\boldsymbol{X}_{1:N_{\text{pt}}}^{t}, \boldsymbol{y}_{1:N_{\text{pt}}}^{t}, \boldsymbol{x}^{t}, y^{t}\}_{t=1}^{T_{\text{pt}}}$, and evaluate its estimation error $\frac{1}{T_{\text{test}}} \sum_{t=1}^{T_{\text{test}}} (y^{t} - f_{\text{GPT2}}(\boldsymbol{X}_{1:N_{\text{test}}}^{t}, \boldsymbol{y}_{1:N_{\text{test}}}^{t}, \boldsymbol{x}^{t}))^{2}$ across different values of $N_{\text{test}}$ and $d$.

- **Kernel method** based on the Gaussian RBF kernel $k(\boldsymbol{x}, \boldsymbol{x}') = \exp\{-\|\boldsymbol{x} - \boldsymbol{x}'\|^{2}/\sigma^{2}\}$. For each task, the kernel is trained from scratch using the test prompt $(\boldsymbol{X}_{1:N_{\text{test}}}^{t}, \boldsymbol{y}_{1:N_{\text{test}}}^{t})$ as training data.

We set $d = 4, 8, 12, 16$ and measure the test error at $N_{\text{test}} = \lfloor 2^{k} - 1 \rfloor$ ($k = 2, 2.5, \ldots, 7$). In Figure 6.1 we observe that the performance of kernel models deteriorates rapidly due to the sample complexity lower bound $d^{\deg(\sigma_{*})} = d^{3}$, whereas the pretrained transformer exhibits a much more favorable dimension scaling in the in-context sample complexity.

### 6.2 Sample Complexity Scaling

To further validate our theoretical claim that pretrained transformers can achieve test-time sample complexity beyond the CSQ lower bound, we conduct a sample complexity probing experiment under the same setting as in Section 6.1. In Figure 6.2, we vary the input dimensionality $d$ and measure the minimal number of samples required to reach specified test error thresholds. The estimated inference-time sample complexity scales approximately as $n \simeq d^{1.1}$, outperforming both the kernel lower bound of $d^{3}$ and the CSQ lower bound of $n \gtrsim d^{1.5}$ for the link function $\sigma_{*}(\cdot) = \text{He}_{3}(\cdot)$ which has information exponent 3.

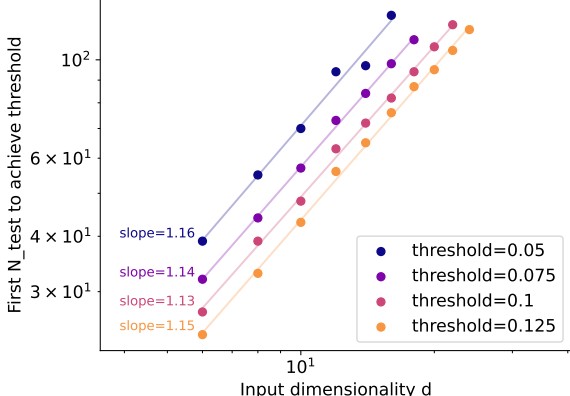

*Figure 6.2.* Sample complexity of pretrained GPT-2 on the degree-3 single-index task $y = \text{He}_{3}(\langle \boldsymbol{\beta}, \boldsymbol{x} \rangle)$. We plot the minimal test-time sample size $N_{\text{test}}$ required to achieve test error thresholds $(0.05, 0.075, 0.1, 0.125)$ against input dimensionality $d$ on log-log scale. The slope of each plot is estimated using least squares.

### 7 Conclusion and Future Direction

We studied in-context learning of single-index polynomials and demonstrated that pretrained transformers achieve inference-time sample complexity that surpasses both kernel methods and CSQ lower bounds. Our analysis highlights

that the Softmax attention can extract feature information by leveraging nonlinear transformations applied to labels.

We outline several future directions. First, the *pretraining* procedure and its current sample complexity of $d^{\Theta(\text{ie}(\sigma_{*}))}$ may potentially be improved to $d^{\Theta(\text{ge}(\sigma_{*}))}$ by going beyond one gradient step (Dandi et al., 2024; Arnaboldi et al., 2024; Lee et al., 2024). Second, extending our analysis to broader function classes – such as non-polynomial $\sigma_{*}$, multi-index targets, or anisotropic covariates (Ghorbani et al., 2020; Refinetti et al., 2021; Mousavi-Hosseini et al., 2023b) – is another interesting direction. Finally, while our study demonstrates both "memorization" of the $r$-dimensional support of the target function class and inference-time feature learning, it remains an open question whether a tradeoff or certain interaction exists between these two mechanisms.

### Acknowledgements

NN was partially supported by JST ACT-X (JPMJAX24CK) and JST BOOST (JPMJBS2418). YS was partially supported by JST CREST (JPMJCR2115). TS was partially supported by JSPS KAKENHI (24K02905) and JST CREST (JPMJCR2015). This research is supported by the National Research Foundation, Singapore and the Ministry of Digital Development and Information under the AI Visiting Professorship Programme (award number AIVP-2024-004). Any opinions, findings and conclusions or recommendations expressed in this material are those of the author(s) and do not reflect the views of National Research Foundation, Singapore and the Ministry of Digital Development and Information.

### Impact Statement

This paper presents work whose goal is to advance the field of Machine Learning. There are many potential societal consequences of our work, none which we feel must be specifically highlighted here.

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

# A Preliminaries

## A.1 High Probability Events

We frequently use the term "with high probability" or "w.h.p.":

**Definition 10.** *We state that an event $A$ occurs* with high probability, *when*

$$\mathbb{P}[A] \geq 1 - d^{-C_{\text{whp}}}$$

*holds for a sufficiently large constant $C_{\text{whp}}$. Here, $C_{\text{whp}}$ depends on neither $d$ nor $r$.*

An illustrative example is Gaussian tail bound: for $z \sim \mathcal{N}(0,1)$ and $t \geq 0$, it holds that

$$\mathbb{P}[z > t] \leq \exp\{-t^2/2\}.$$

Plugging $\sqrt{2C_{\text{whp}} \log d}$ into $t$ above, we can see that $z \leq \sqrt{2C_{\text{whp}} \log d}$ holds with probability at least $1 - d^{-C_{\text{whp}}}$. In this situation, we say that "$z \leq O(\sqrt{\log d})$ with high probability"—this means that we can redefine $C_{\text{whp}}$ to be sufficiently large, by changing the hidden constant in $O(\sqrt{\log d})$.

Throughout this paper, whenever we take the intersection of high-probability events $A_1, \ldots, A_m$, $m = O(\text{poly} d)$ is always ensured. Thus, (by redefining $C_{\text{whp}}$,) we can state that $A_1 \cap \cdots \cap A_m$ also occurs with high probability.

Moreover, we consider the *high-dimensional setting*, i.e., our result holds for all $d \geq C_{\text{hd}}$ where $C_{\text{hd}}$ is a constant independent of both $d$ and $r$.

Unless otherwise specified, the notations $O, \Theta, \Omega, \tilde{O}, \tilde{\Theta}, \tilde{\Omega}$ are considered with respect to $d$. Note that we allow $r$ to scale with $d$, while treating $\deg(\sigma_*)$ as an $O(1)$ quantity.

## A.2 Estimator Constructed by Single-Layer Transformer

We now show that the estimator constructed by the transformer (3.1) is explicitly written as in (3.2), given our parameter configuration

$$\boldsymbol{W}^{KQ} = \begin{bmatrix} \boldsymbol{\Gamma} & \boldsymbol{0}_{d \times 1} \\ \boldsymbol{0}_{1 \times d} & 1 \end{bmatrix}, \ \boldsymbol{W}^{FV} = \begin{bmatrix} \boldsymbol{O}_{m \times d} & \boldsymbol{v} \end{bmatrix}.$$

From (3.1) and the configuration above, the estimator $f_{\text{TF}}$ is written as, recalling that $\boldsymbol{E} = \begin{bmatrix} \boldsymbol{x}_1 & \cdots & \boldsymbol{x}_N & \boldsymbol{x} \\ y_1 & \cdots & y_N & 1 \end{bmatrix}$,

$$f_{\text{TF}} = \boldsymbol{a}^\top \sigma(\boldsymbol{W}^{FV} \boldsymbol{E} \text{Softmax}(\text{Mask}(\rho^{-1} \boldsymbol{E}^\top \boldsymbol{W}^{KQ} \boldsymbol{E}_{:,N+1})) + \boldsymbol{b}$$

$$= \boldsymbol{a}^\top \sigma \left( \boldsymbol{W}^{FV} \boldsymbol{E} \text{Softmax} \left( \begin{bmatrix} \boldsymbol{x}_1^\top \boldsymbol{\Gamma} \boldsymbol{x}/\rho + y_1/\rho \\ \vdots \\ \boldsymbol{x}_N^\top \boldsymbol{\Gamma} \boldsymbol{x}/\rho + y_N/\rho \\ -\infty \end{bmatrix} \right) + \boldsymbol{b} \right)$$

$$= \boldsymbol{a}^\top \sigma \left( \begin{bmatrix} \boldsymbol{v}y_1 & \cdots & \boldsymbol{v}y_N & \boldsymbol{v} \end{bmatrix} \text{Softmax} \left( \begin{bmatrix} \boldsymbol{x}_1^\top \boldsymbol{\Gamma} \boldsymbol{x}/\rho + y_1/\rho \\ \vdots \\ \boldsymbol{x}_N^\top \boldsymbol{\Gamma} \boldsymbol{x}/\rho + y_N/\rho \\ -\infty \end{bmatrix} \right) + \boldsymbol{b} \right)$$

$$= \sum_{j=1}^m a_j \sigma \left( v_j \frac{\sum_{i=1}^N y_i e^{y_i/\rho} e^{\boldsymbol{x}_i^\top \boldsymbol{\Gamma} \boldsymbol{x}/\rho}}{\sum_{i=1}^N e^{y_i/\rho} e^{\boldsymbol{x}_i^\top \boldsymbol{\Gamma} \boldsymbol{x}/\rho}} + b_j \right).$$

Hence we obtain (3.2).

# B In-Context Feature Extraction via Softmax Self-Attention Module

We show that the softmax self-attention has the ability to extract the feature vector $\boldsymbol{\beta}$ of the single-index function $\sigma_*(\langle \boldsymbol{x}, \boldsymbol{\beta} \rangle)$ (this section precedes the optimization analysis, as the techniques developed here are later utilized in the optimization theory). Crucially, we show that the nonlinear transformation applied to $\boldsymbol{y}$ via softmax attention module reduces the effective

information exponent. Consequently, the generative exponent (Damian et al., 2024) of the link function $\sigma_*$ governs the inference-time complexity, which surpasses the CSQ limit (Damian et al., 2022). Let

$$g(\boldsymbol{X}, \boldsymbol{y}, \boldsymbol{x}; \boldsymbol{\Gamma}) := \frac{\sum_{i=1}^{N} y_i e^{y_i/\rho} e^{\boldsymbol{x}_i^\top \boldsymbol{\Gamma} \boldsymbol{x}/\rho}}{\sum_{i=1}^{N} e^{y_i/\rho} e^{\boldsymbol{x}_i^\top \boldsymbol{\Gamma} \boldsymbol{x}/\rho}} \tag{B.1}$$

be the output of the attention layer. The goal of this section is to show the following Proposition 11, stating that $g$ can extract the inner product between $\boldsymbol{x}$ and $\boldsymbol{\beta}$ which varies across tasks.

**Proposition 11.** *Let $\rho = \Theta(\log^{C_\rho} d)$ and $\kappa = \Theta(\log^{C_\kappa} d)$ where $C_\rho$ and $C_\kappa$ are any constants satisfying $C_\rho \geq \deg(\sigma_*)/2 + 1$ and $C_\kappa \geq \max\{2C_\rho e_{\mathrm{ge}(\sigma_*)}(\sigma_*) + 1, 4\deg(\sigma_*) - 4\}$ ($e_i(\sigma_*)$ is defined in Definition 14). Suppose $\boldsymbol{\Gamma} = \frac{1}{\kappa\sqrt{r}}\boldsymbol{G}$ for a matrix $\boldsymbol{G}$ written as*

$$\boldsymbol{G} = \boldsymbol{U}^\top \boldsymbol{D} \boldsymbol{U} + \boldsymbol{N}, \tag{B.2}$$

*where $\boldsymbol{D} := \mathrm{diag}(\underbrace{1,\ldots,1}_{r}, \underbrace{0,\ldots,0}_{d-r})$, $\boldsymbol{U}$ is an orthogonal matrix, and $\boldsymbol{N}$ is a matrix satisfying $\|\boldsymbol{N}\|_F = O(1/\sqrt{d})$.*

*Now, if $N = \tilde{\Omega}(r^{3\mathrm{ge}(\sigma_*)/2})$, then with high probability over length-$N$ prompts $(\boldsymbol{X}_{1:N}, \boldsymbol{y}_{1:N}, \boldsymbol{x})$, it holds that*

$$g(\boldsymbol{X}, \boldsymbol{y}, \boldsymbol{x}; \boldsymbol{\Gamma})$$
$$= P_1 + P_2 \left(\frac{\langle \boldsymbol{G}\boldsymbol{x}, \boldsymbol{\beta} \rangle}{\sqrt{r}}\right)^{\mathrm{ge}(\sigma_*)} + o_d(P_2 r^{-3\mathrm{ge}(\sigma_*)/4} \log^{-2\deg(\sigma_*)+2} d). \tag{B.3}$$

*Here, $P_1$ and $P_2$ are independent of the data and satisfies $P_1 = o_d(1)$ and $P_2 = \Theta_d\left((\log d)^{-C_{P_2}}\right)$, where $C_{P_2}$ is a constant dependent on $\sigma_*$ but is uniformly upper bounded using $\deg(\sigma_*)$ and $\mathrm{odd}(\sigma_*) := \mathbb{E}_{z\sim\mathcal{N}(0,1)}[\sigma_{*,\mathrm{odd}}(z)^2]$, where $\sigma_{*,\mathrm{odd}}(\cdot)$ is a polynomial constructed by extracting only the odd parts of $\sigma_*$.*

**Remark 12.** *Later we will show that $\boldsymbol{\Gamma}$ after pretraining satisfies $\boldsymbol{\Gamma} \simeq \sqrt{r}\mathbb{E}_\beta[\boldsymbol{\beta}\boldsymbol{\beta}^\top] = \boldsymbol{U}^\top\boldsymbol{D}\boldsymbol{U}/\sqrt{r}$, which matches the condition of Proposition 11. Since $\boldsymbol{U}$ is spanned by the basis of the support of $\boldsymbol{\beta}$ in this case, $\langle \boldsymbol{G}\boldsymbol{x}, \boldsymbol{\beta}\rangle$ in (B.3) satisfies $\langle \boldsymbol{G}\boldsymbol{x}, \boldsymbol{\beta}\rangle \simeq \langle \boldsymbol{x}, \boldsymbol{\beta}\rangle$. This implies that, if the context length at inference time is $\tilde{\Omega}(r^{\mathrm{ge}(\sigma_*)})$, then the pretrained attention layer can extract the feature vector $\boldsymbol{\beta}$ which varies across different prompts and compute $\langle \boldsymbol{x}, \boldsymbol{\beta}\rangle$. This highlights the inference-time feature learning ability of the pretrained attention layer. Moreover, the inference-time sample complexity depends on neither $\deg(\sigma_*)$ nor $\mathrm{ie}(\sigma_*)$, which surpasses kernel and CSQ methods. The factor $\log^{-2\deg(\sigma_*)+2} d$ in the residual term is introduced just for compatibility with later stages.*

**Preparations.** We use the following property, which states that there exists a polynomial transformation which reduces the information exponent of $\sigma_*$ to its generative exponent. Recall that $\mathrm{H}(f, i)$ denotes the degree-$i$ Hermite coefficient of $f$ (that is, $f(z) = \sum_{i\geq 0} \frac{\mathrm{H}(f,i)}{i!}\mathrm{He}_i(z)$).

**Lemma 13** ((Lee et al., 2024), Proposition 6). *The following holds:*

- *If $\mathrm{ge}(\sigma_*) = 1$, then there exists $i \leq C_1(\deg(\sigma_*), \mathrm{odd}(\sigma_*))$ such that $|\mathrm{H}(\sigma_*^i, 1)| \geq D_1(\deg(\sigma_*), \mathrm{odd}(\sigma_*))$.*

- *If $\mathrm{ge}(\sigma_*) = 2$, then there exists $i \leq C_2(\deg(\sigma_*))$ such that $|\mathrm{H}(\sigma_*^i, 2)| \geq D_2(\deg(\sigma_*))$.*

*Here, $C_1(\deg(\sigma_*), \mathrm{odd}(\sigma_*))$ and $D_1(\deg(\sigma_*), \mathrm{odd}(\sigma_*))$ are constants which only depend on $\deg(\sigma_*)$ and $\mathrm{odd}(\sigma_*) := \mathbb{E}_{z\sim\mathcal{N}(0,1)}[\sigma_{*,\mathrm{odd}}(z)^2]$ where $\sigma_{*,\mathrm{odd}}(\cdot)$ is a polynomial constructed by extracting only the odd parts of $\sigma_*$. Similarly, $C_2(\deg(\sigma_*))$ and $D_2(\deg(\sigma_*))$ are constants depending only on $\deg(\sigma_*)$.*

**Definition 14.** *For $i \geq 1$, let $e_i(\sigma_*)$ be the minimal $j \geq 1$ such that $\mathrm{H}(\sigma_*^j, i) \neq 0$ holds. If such $j$ does not exist, then define $e_i(\sigma_*) = \infty$. (Especially from Lemma 13, $e_{\mathrm{ge}(\sigma_*)}(\sigma_*)$ can be uniformly upper bounded using the degree and the odd part of $\sigma_*$.)*

Another key property is that $\boldsymbol{y}$ is bounded with high probability.

**Lemma 15** ((Oko et al., 2024b), Corollary 17). *If $\|\boldsymbol{\beta}\| = 1$ and $\boldsymbol{x} \sim \mathcal{N}(0, \boldsymbol{I}_d)$, then $|\sigma_*(\langle \boldsymbol{\beta}, \boldsymbol{x}\rangle)| \lesssim (\log d)^{\deg(\sigma_*)/2}$ holds with high probability.*

This motivates us to introduce a new function $\bar{\sigma}_*(z) := \begin{cases} \frac{\sigma_*(z)}{\rho} & \left(\text{if } \left|\frac{\sigma_*(z)}{\rho}\right| \leq \frac{1}{\log d}\right) \\ 0 \text{ (otherwise)} \end{cases}$ . Note that $\bar{\sigma}_*(\langle \boldsymbol{\beta}, \boldsymbol{x}\rangle) = \sigma_*(\langle \boldsymbol{\beta}, \boldsymbol{x}\rangle)/\rho$ holds with high probability from Lemma 15 and the condition of $\rho$.

Here we show the following lemma appeared in Section 5, stating that the nonlinear transformation $\exp(\cdot)$ can reduce the information exponent.

**Lemma 9.** *For $p \geq 1$, if $e_p(\sigma_*) < \infty$, then $\mathrm{H}(\exp(\bar{\sigma}_*), p) = \Theta\left(\frac{1}{(\log d)^{C_\rho e_p(\sigma_*)}}\right)$ and $\mathrm{H}(\bar{\sigma}_* \exp(\bar{\sigma}_*), p) = \Theta\left(\frac{1}{(\log d)^{C_\rho e_p(\sigma_*)}}\right)$ holds. Here, the hidden constants in $\Theta$ notations may depend on $\sigma_*$ and $p$. If $e_i(\sigma_*) = \infty$, then $\mathrm{H}(\exp(\bar{\sigma}_*), i)$ can be taken to be of $O(d^{-C})$ where $C$ is a sufficiently large constant.*

*This immediately implies that the information exponent of $\exp(\bar{\sigma}_*)$ and $\bar{\sigma}_* \exp(\bar{\sigma}_*)$ is equal to $\mathrm{ge}(\sigma_*)$.*

*Proof.* We consider $\mathrm{H}(\exp(\bar{\sigma}_*), p)$ for $e_p(\sigma_*) < \infty$. The other cases can be derived similarly. From the Taylor expansion, we obtain

$$\mathrm{H}(\exp(\bar{\sigma}_*), p) = \underbrace{\frac{1}{e_p(\sigma_*)!}\mathrm{H}(\bar{\sigma}_*^{e_p(\sigma_*)}, p)}_{\text{(I)}} + \underbrace{\sum_{i=e_p(\sigma_*)+1}^{C_\rho e_p(\sigma_*)}\frac{1}{i!}\mathrm{H}(\bar{\sigma}_*^i, p)}_{\text{(II)}} + \underbrace{\sum_{i \geq C_\rho e_p(\sigma_*)+1}\frac{1}{i!}\mathrm{H}(\bar{\sigma}_*^i, p)}_{\text{(III)}}.$$

For (I),

$$\mathrm{H}(\bar{\sigma}_*^{e_p(\sigma_*)}, p) = \mathbb{E}_{z \sim \mathcal{N}(0,1)}[\mathrm{He}_p(z)\bar{\sigma}_*^{e_p(\sigma_*)}(z)]$$
$$= \frac{1}{\rho^{e_p(\sigma_*)}}\mathbb{E}[\mathrm{He}_p(z)\sigma_*^{e_p(\sigma_*)}(z)] - \frac{1}{\rho^{e_p(\sigma_*)}}\mathbb{E}[\mathrm{He}_p(z)\sigma_*^{e_p(\sigma_*)}(z)\mathbb{I}(|\sigma_*(z)| \geq \rho(\log d)^{-1})]$$

holds. Here, from Hölder's inequality, we know

$$\mathbb{E}[\mathrm{He}_p(z)\sigma_*^{e_p(\sigma_*)}(z)\mathbb{I}(|\sigma_*(z)| \geq \rho(\log d)^{-1})] \leq \mathbb{E}[\mathrm{He}_p(z)^4]^{1/4}\mathbb{E}[\sigma_*^{4e_p(\sigma_*)}(z)]^{1/4}\mathbb{E}[\mathbb{I}(|\sigma_*(z)| \geq \rho(\log d)^{-1})]^{1/2} \quad \text{(B.4)}$$

Moreover, $\mathbb{E}[\mathbb{I}(|\sigma_*(z)| \geq \rho(\log d)^{-1})] = O(d^{-1})$ holds due to the high probability property. Consequently, (I) is of $\Theta((\log d)^{-C_\rho e_p(\sigma_*)})$.

For (II), note that $\frac{1}{i!}\mathrm{H}(\bar{\sigma}_*^i, p)$ is dominated by $\frac{1}{i!\rho^i}\mathbb{E}[\mathrm{He}_p(z)\sigma_*^i(z)]$ from the similar argument as above. Moreover, we can see from Hölder's inequality that

$$\left|\frac{1}{i!\rho^i}\mathbb{E}[\mathrm{He}_p(z)\sigma_*^i(z)]\right| \lesssim \frac{1}{i!\rho^i}\sqrt{\mathbb{E}[\sigma_*^{2i}(z)]}.$$

Here, as $M(\sigma_*) = \max_{e_p(\sigma_*)+1 \leq i \leq C_\rho e_p(\sigma_*)} \sqrt{\mathbb{E}[\sigma_*^{2i}(z)]}$ is the constant depending only on $\sigma_*$ and $p$, we can state that (II) is of $O((\log d)^{-C_\rho(e_p(\sigma_*)+1)})$.

Finally, let us bound (III). From the definition of $\bar{\sigma}_*$, we get

$$\left|\frac{1}{i!}\mathrm{H}(\bar{\sigma}_*^i, p)\right| \leq \frac{1}{i!}\sqrt{\mathbb{E}[\mathrm{He}_p(z)^2]}\sqrt{\mathbb{E}[\bar{\sigma}_*^{2i}(z)]}$$
$$\lesssim \frac{1}{i!}(\log d)^{-i}.$$

Moreover, from the bound of residual terms of the Taylor expansion,

$$\sum_{i \geq C_\rho e_p(\sigma_*)+1}\frac{1}{i!}(\log d)^{-i} \lesssim (\log d)^{-C_\rho e_p(\sigma_*)-1}$$

is satisfied.

Overall, we conclude that $\mathrm{H}(\exp(\bar{\sigma}_*), p) = \Theta\left(\frac{1}{(\log d)^{C_\rho e_p(\sigma_*)}}\right)$. $\qquad\square$

Lemma 9 does not deal with the case where $p = 0$. Here we treat this case:

**Lemma 16.** $\mathrm{H}(\bar{\sigma}_* \exp \bar{\sigma}_*, 0) = \frac{1}{\rho^2}(1 + o_d(1))$ and $\mathrm{H}(\exp \bar{\sigma}_*, 0) = 1 + o_d(1)$ holds.

*Proof.* Note that $\mathrm{H}(\sigma_*/\rho, 0) = 0$ and $\mathrm{H}((\sigma_*/\rho)^2, 0) = \frac{1}{\rho^2}$ from Assumption 4. For $\sigma_* \exp \bar{\sigma}_* = \sum_{i \geq 0} \frac{1}{i!} \bar{\sigma}_*^{i+1}$ and $\exp \bar{\sigma}_* = \sum_{i \geq 0} \frac{1}{i!} \bar{\sigma}_*^i$, conducting the same argument as Lemma 9 yields the assertion. $\square$

Let us derive bounds for some important quantities.

**Lemma 17.** Suppose $x, x' \sim \mathcal{N}(0, I_d)$ and $\beta \sim \mathrm{Supp}(\beta) \coloneqq \{\beta \mid \beta \in \mathcal{S}_r, \|\beta\| = 1\}$. Then, $\frac{x^\top G x'}{\rho \kappa \sqrt{r}} = o(1)$ and $\beta^\top G x = O(\sqrt{\log d})$ holds with high probability.

*Proof.* First we can confirm that

$$\frac{x^\top G x'}{\rho \kappa \sqrt{r}} = x^\top U^\top DU x'/\rho \kappa \sqrt{r} + x^\top N x'/\rho \kappa \sqrt{r} = o(1)$$

holds with high probability: from the rotational invariance, $y = Ux$ and $y' = Ux'$ are standard Gaussian vectors. Then, the first term is $o(1)$ with high probability from Lemma 29 and $\rho \kappa = \Omega(\log d)$. To show that the second term is $o(1)$, we can again use Lemma 29 for the case $r = d$ and the fact $\|N\|_2 = O(d^{-1/2})$.

Next we show

$$\beta^\top G x = \beta^\top U^\top DU x + \beta^\top N x = O(\sqrt{\log d}).$$

For the first term, note that $\|\beta^\top U^\top DU\| \leq 1$ holds. Then, (for any fixed $\beta$) $\beta^\top U^\top DU x$ is the Gaussian with variance at most one. Thus, Gaussian tail bound indicates that $\beta^\top U^\top DU x \lesssim \sqrt{\log d}$ with high probability. The second term can be also derived using the fact that $\|N x\| = O(1)$ holds with high probability from Lemma 28. $\square$

**Lemma 18.** With high probability over the prompt $(X, y, x)$,

$$\left| N^{-1} \sum_{i=1}^N \exp\left(\frac{y_i}{\rho}\right) \exp\left(\frac{x_i^\top G x}{\rho \kappa \sqrt{r}}\right) - \mathbb{E}_{x_1, \zeta_1}\left[\exp\left(\bar{\sigma}_*(\langle \beta, x_1 \rangle) + \frac{\zeta_1}{\rho}\right) \exp\left(\frac{x_1^\top G x}{\rho \kappa \sqrt{r}}\right)\right] \right| = \tilde{O}\left(N^{-\frac{1}{2}}\right)$$

*and*

$$\left| N^{-1} \sum_{i=1}^N \left(\frac{y_i}{\rho}\right) \exp\left(\frac{y_i}{\rho}\right) \exp\left(\frac{x_i^\top G x}{\rho \kappa \sqrt{r}}\right) - \mathbb{E}_{x_1, \zeta_1}\left[\bar{\sigma}_*(\langle \beta, x_1 \rangle) \exp\left(\bar{\sigma}_*(\langle \beta, x_1 \rangle) + \frac{\zeta_1}{\rho}\right) \exp\left(\frac{x_1^\top G x}{\rho \kappa \sqrt{r}}\right)\right] \right|$$
$$= \tilde{O}\left(N^{-\frac{1}{2}}\right)$$

*holds.*

*Proof.* Let us show the first assertion: the second one is derived in the same manner.

From Lemma 17, $N^{-1} \sum_{i=1}^N \exp(y_i/\rho) \exp(x_i^\top G x/\rho \kappa \sqrt{r}) = N^{-1} \sum_{i=1}^N \exp(\bar{\sigma}_*(\langle \beta, x_i \rangle) + \zeta_i/\rho) \exp(\mathrm{clip}(x_i^\top G x/\rho \kappa \sqrt{r}, 1))$ with high probability where $\mathrm{clip}(x, c) = \begin{cases} x \ (\text{if } |x| \leq c) \\ 0 \ (\text{if } |x| > c) \end{cases}$. Conditioned on this, with high probability,

$$N^{-1} \sum_{i=1}^N \exp(y_i/\rho) \exp(x_i^\top G x/\rho \kappa \sqrt{r}) - \mathbb{E}_{x_1, \zeta_1}\left[\exp(\bar{\sigma}_*(\langle \beta, x_1 \rangle) + \zeta_1/\rho) \exp(x_1^\top G x/\rho \kappa \sqrt{r})\right]$$

$$= \underbrace{N^{-1} \sum_{i=1}^N e^{\bar{\sigma}_*(\langle \beta, x_i \rangle) + \zeta_i/\rho} e^{\mathrm{clip}(x_i^\top G x/\rho \kappa \sqrt{r}, 1)} - \mathbb{E}_{x_1, \zeta_1}\left[e^{\bar{\sigma}_*(\langle \beta, x_1 \rangle) + \zeta_1/\rho} e^{\mathrm{clip}(x_1^\top G x/\rho \kappa \sqrt{r}, 1)}\right]}_{(\mathrm{I})}$$

$$+ \underbrace{\mathbb{E}_{\boldsymbol{x}_1,\zeta_1}\left[e^{\bar{\sigma}_*(\langle\boldsymbol{\beta},\boldsymbol{x}_1\rangle)+\zeta_1/\rho}e^{\mathrm{clip}(\boldsymbol{x}_1^\top\boldsymbol{G}\boldsymbol{x}/\rho\kappa\sqrt{r},1)}\right] - \mathbb{E}_{\boldsymbol{x}_1,\zeta_1}\left[e^{\bar{\sigma}_*(\langle\boldsymbol{\beta},\boldsymbol{x}_1\rangle)+\zeta_1/\rho}e^{\boldsymbol{x}_1^\top\boldsymbol{G}\boldsymbol{x}/\rho\kappa\sqrt{r}}\right]}_{\text{(II)}}$$

holds. (I) is of $\tilde{O}(N^{-\frac{1}{2}})$ with high probability from Hoeffding's inequality, noting that $\exp(\bar{\sigma}_*(\langle\boldsymbol{\beta},\boldsymbol{x}_i\rangle) + \zeta_i/\rho)\exp(\mathrm{clip}(\boldsymbol{x}_i^\top\boldsymbol{G}\boldsymbol{x}/\rho\kappa\sqrt{r},1))$ is $O(1)$-bounded. (II) is upper-bounded as, using Hölder's inequality,

$$\mathbb{E}_{\boldsymbol{x}_1}\left[\left|\exp(\bar{\sigma}_*(\langle\boldsymbol{\beta},\boldsymbol{x}_1\rangle) + \zeta_1/\rho)(1 - \exp(\boldsymbol{x}_1^\top\boldsymbol{G}\boldsymbol{x}/\rho\kappa\sqrt{r}))\mathbb{I}((\boldsymbol{x}_1^\top\boldsymbol{G}\boldsymbol{x}/\rho\kappa\sqrt{r}) \geq 1)\right|\right]$$
$$\leq \mathbb{E}_{\boldsymbol{x}_1}\left[\exp(\bar{\sigma}_*(\langle\boldsymbol{\beta},\boldsymbol{x}_1\rangle) + \zeta_1/\rho)^4\right]^{1/4}\mathbb{E}_{\boldsymbol{x}_1}\left[(1 - \exp(\boldsymbol{x}_1^\top\boldsymbol{G}\boldsymbol{x}/\rho\kappa\sqrt{r}))^4\right]^{1/4}\mathbb{E}[\mathbb{I}((\boldsymbol{x}_1^\top\boldsymbol{G}\boldsymbol{x}/\rho\kappa\sqrt{r}) \geq 1)^2]^{1/2}$$
$$\leq \mathbb{E}_{\boldsymbol{x}_1}\left[(1 - \exp(\boldsymbol{x}_1^\top\boldsymbol{G}\boldsymbol{x}/\rho\kappa\sqrt{r}))^4\right]^{1/4} \cdot O(d^{-C^*})$$

where $C^*$ is a constant which can be taken to be sufficiently large from the definition of high probability events. We conclude that (II) is negligible by showing that $\mathbb{E}_{\boldsymbol{x}_1}\left[(1 - \exp(\boldsymbol{x}_1^\top\boldsymbol{G}\boldsymbol{x}/\rho\kappa\sqrt{r}))^4\right] = O(1)$ with high probability over $\boldsymbol{x}$. It suffices to show that $\mathbb{E}_{\boldsymbol{x}_1}\left[(\exp(\boldsymbol{x}_1^\top\boldsymbol{G}\boldsymbol{x}/\rho\kappa\sqrt{r}))^l\right] = \mathbb{E}_{\boldsymbol{x}_1}\left[\exp(l\boldsymbol{x}_1^\top\boldsymbol{G}\boldsymbol{x}/\rho\kappa\sqrt{r})\right] = O(1)$ with high probability for $l \in [4]$. Here, it holds that

$$\mathbb{E}_{\boldsymbol{x}_1}\left[\exp(l\boldsymbol{x}_1^\top\boldsymbol{G}\boldsymbol{x}/\rho\kappa\sqrt{r})\right]$$
$$= \mathbb{E}_{\boldsymbol{x}_1\sim\mathcal{N}(0,\boldsymbol{I}_d)}\left[\exp(l\boldsymbol{x}_1^\top\boldsymbol{U}^\top\boldsymbol{D}\boldsymbol{U}\boldsymbol{x}/\rho\kappa\sqrt{r})\exp(l\boldsymbol{x}_1^\top\boldsymbol{N}\boldsymbol{x}/\rho\kappa\sqrt{r})\right]$$
$$\leq \mathbb{E}_{\boldsymbol{x}_1\sim\mathcal{N}(0,\boldsymbol{I}_d)}\left[\exp(2l\boldsymbol{x}_1^\top\boldsymbol{D}\boldsymbol{U}\boldsymbol{x}/\rho\kappa\sqrt{r})\right]^{1/2}\mathbb{E}_{\boldsymbol{x}_1\sim\mathcal{N}(0,\boldsymbol{I}_d)}\left[\exp(2l\boldsymbol{x}_1^\top\boldsymbol{N}\boldsymbol{x}/\rho\kappa\sqrt{r})\right]^{1/2}. \tag{B.5}$$

Note that $\|2l\boldsymbol{D}\boldsymbol{U}\boldsymbol{x}/\rho\kappa\sqrt{r}\| \leq 1$ and $\|2l\boldsymbol{N}\boldsymbol{x}/\rho\kappa\sqrt{r}\| \leq 1$ holds with high probability over $\boldsymbol{x} \sim \mathcal{N}(0,\boldsymbol{I}_d)$: for the former one, we can use Lemma 28 as $\boldsymbol{U}\boldsymbol{x} \sim \mathcal{N}(0,\boldsymbol{I}_d)$. For the latter one, we again apply Lemma 28 (for the case where $r = d$), together with $\|\boldsymbol{N}\|_F = O(1/\sqrt{d})$. In this case, (B.5) can be upper-bounded by

$$\mathbb{E}_{z\sim\mathcal{N}(0,1)}[\exp(\lambda_1 z)]^{1/2}\mathbb{E}_{z\sim\mathcal{N}(0,1)}[\exp(\lambda_2 z)]^{1/2}$$

where $\lambda_1, \lambda_2 \leq 1$, which is known to be $O(1)$. This completes the proof. $\qquad\square$

**Proof of Proposition 11.** Now we prove Proposition 11. We first naively evaluate the denominator and numerator of (B.1) separately, and deal with their correlation afterwards.

*Proof.* Plugging the parameter values into (B.1) yields

$$(\text{B.1}) = \rho\frac{N^{-1}\sum_{i=1}^N (y_i/\rho)\exp(y_i/\rho)\exp(\boldsymbol{x}_i^\top\boldsymbol{G}\boldsymbol{x}/\rho\kappa\sqrt{r})}{N^{-1}\sum_{i=1}^N \exp(y_i/\rho)\exp(\boldsymbol{x}_i^\top\boldsymbol{G}\boldsymbol{x}/\rho\kappa\sqrt{r})}.$$

First, let us evaluate the denominator $N^{-1}\sum_{i=1}^N \exp(y_i/\rho)\exp(\boldsymbol{x}_i^\top\boldsymbol{G}\boldsymbol{x}/\rho\kappa\sqrt{r})$. Note that, with high probability over the distribution of $(\boldsymbol{x}_1,\ldots,\boldsymbol{x}_N)$, this is equal to the clipped version $N^{-1}\sum_{i=1}^N \exp(\bar{\sigma}_*(\langle\boldsymbol{\beta},\boldsymbol{x}_i\rangle) + \zeta_i/\rho)\exp(\boldsymbol{x}_i^\top\boldsymbol{G}\boldsymbol{x}/\rho\kappa\sqrt{r})$.

Its expectation (assume $\boldsymbol{x}$ is fixed now) is calculated as, using $\exp(\lambda z) = \sum_{i\geq 0}\frac{\exp(\lambda^2/2)}{i!}\lambda^i\mathrm{He}_i(z)$,

$$\mathbb{E}_{\boldsymbol{x}_1,\zeta_1}\left[\exp(\zeta_1/\rho)\exp(\bar{\sigma}_*(\langle\boldsymbol{\beta},\boldsymbol{x}_1\rangle))\exp(\boldsymbol{x}_1^\top\boldsymbol{G}\boldsymbol{x}/\rho\kappa\sqrt{r})\right]$$
$$= \frac{1}{2}\left(\exp(\tau/\rho) + \exp(-\tau/\rho)\right)\mathbb{E}_{\boldsymbol{x}_1}\underbrace{\left[\sum_{i\geq 0}\frac{1}{i!}\mathrm{H}(\exp\bar{\sigma}_*,i)\mathrm{He}_i(\langle\boldsymbol{\beta},\boldsymbol{x}_1\rangle)\exp\left(\frac{\|\boldsymbol{G}\boldsymbol{x}\|}{\rho\kappa\sqrt{r}} \cdot \frac{\boldsymbol{x}_1^\top\boldsymbol{G}\boldsymbol{x}}{\|\boldsymbol{G}\boldsymbol{x}\|}\right)\right]}_{(*)}$$

and

$$(*) = \mathbb{E}_{\boldsymbol{x}_1}\left[\sum_{i\geq 0}\frac{1}{i!}\mathrm{H}(\exp\bar{\sigma}_*,i)\mathrm{He}_i(\langle\boldsymbol{\beta},\boldsymbol{x}_1\rangle)\sum_{j\geq 0}\frac{1}{j!}\left(\frac{\|\boldsymbol{G}\boldsymbol{x}\|}{\rho\kappa\sqrt{r}}\right)^j\exp\left(\frac{\|\boldsymbol{G}\boldsymbol{x}\|^2}{2\rho^2\kappa^2 r}\right)\mathrm{He}_j(\langle\boldsymbol{x}_1,\boldsymbol{G}\boldsymbol{x}/\|\boldsymbol{G}\boldsymbol{x}\|\rangle)\right]$$

$$
=\sum_{i\geq 0}\frac{1}{i!}\mathrm{H}(\exp\bar{\sigma}_*,i)\left(\frac{\|\boldsymbol{Gx}\|}{\rho\kappa\sqrt{r}}\right)^i\exp\left(\frac{\|\boldsymbol{Gx}\|^2}{2\rho^2\kappa^2 r}\right)(\langle\boldsymbol{\beta},\boldsymbol{Gx}/\|\boldsymbol{Gx}\|\rangle)^i\ (Lemma\ 34)
$$

$$
=\mathrm{H}(\exp\bar{\sigma}_*,0)\exp\left(\frac{\|\boldsymbol{Gx}\|^2}{2\rho^2\kappa^2 r}\right)+\frac{\mathrm{H}(\exp\bar{\sigma}_*,\mathrm{ge}(\sigma_*))}{\mathrm{ge}(\sigma_*)!}\left(\frac{\|\boldsymbol{Gx}\|}{\rho\kappa\sqrt{r}}\right)^{\mathrm{ge}(\sigma_*)}\exp\left(\frac{\|\boldsymbol{Gx}\|^2}{2\rho^2\kappa^2 r}\right)(\langle\boldsymbol{\beta},\boldsymbol{Gx}/\|\boldsymbol{Gx}\|\rangle)^{\mathrm{ge}(\sigma_*)}
$$

$$
+\underbrace{\sum_{i\geq\mathrm{ge}(\sigma_*)+1}\frac{\mathrm{H}(\exp\bar{\sigma}_*,i)}{i!}\left(\frac{\|\boldsymbol{Gx}\|}{\rho\kappa\sqrt{r}}\right)^i\exp\left(\frac{\|\boldsymbol{Gx}\|^2}{2\rho^2\kappa^2 r}\right)(\langle\boldsymbol{\beta},\boldsymbol{Gx}/\|\boldsymbol{Gx}\|\rangle)^i}_{=:\Xi}. \tag{B.6}
$$

We can upper bound the final term $\Xi$: indeed, as $\frac{\|\boldsymbol{Gx}\|}{\rho\kappa\sqrt{r}}\leq 1$ with high probability,

$$
|\Xi|=\left|\sum_{i\geq\mathrm{ge}(\sigma_*)+1}\frac{\mathrm{H}(\exp\bar{\sigma}_*,i)}{i!}\left(\frac{\|\boldsymbol{Gx}\|}{\rho\kappa\sqrt{r}}\right)^i\exp\left(\frac{\|\boldsymbol{Gx}\|^2}{2\rho^2\kappa^2 r}\right)(\langle\boldsymbol{\beta},\boldsymbol{Gx}/\|\boldsymbol{Gx}\|\rangle)^i\right|
$$

$$
\lesssim\sum_{i\geq\mathrm{ge}(\sigma_*)+1}\left|\frac{\mathrm{H}(\exp\bar{\sigma}_*,i)}{i!}(\langle\boldsymbol{\beta},\boldsymbol{Gx}/\rho\kappa\sqrt{r}\rangle)^i\right|
$$

holds. Moreover, from Lemma 33, we know that

$$
\left(\frac{\mathrm{H}(\exp\bar{\sigma}_*,i)}{i!}\right)^2\leq\sum_{i\geq 0}\frac{(\mathrm{H}(\exp\bar{\sigma}_*,i))^2}{i!}=\mathbb{E}_{z\sim N(0,1)}[(\exp\bar{\sigma}_*(z))^2]\leq\mathrm{e}^2.
$$

Then, we get

$$
|\Xi|\lesssim\sum_{i\geq\mathrm{ge}(\sigma_*)+1}\left|\frac{\mathrm{H}(\exp\bar{\sigma}_*,i)}{i!}(\langle\boldsymbol{\beta},\boldsymbol{Gx}/\rho\kappa\sqrt{r}\rangle)^i\right|\lesssim\sum_{i\geq\mathrm{ge}(\sigma_*)+1}|\langle\boldsymbol{\beta},\boldsymbol{Gx}/\rho\kappa\sqrt{r}\rangle|^i\lesssim\langle\boldsymbol{\beta},\boldsymbol{Gx}/\rho\kappa\sqrt{r}\rangle^{\mathrm{ge}(\sigma_*)+1}
$$

with high probability.

Overall, letting $k_\rho:=\frac{1}{2}\left(\exp(\tau/\rho)+\exp(-\tau/\rho)\right)=\Theta_d(1)$ (recall that we assumed $\tau=O(1)$), with high probability we obtain from Lemma 18 that

$$
N^{-1}\sum_{i=1}^{N}\exp(y_i/\rho)\exp(\boldsymbol{x}_i^\top\boldsymbol{Gx}/\rho\kappa\sqrt{r})
$$

$$
=k_\rho\mathrm{H}(\exp\bar{\sigma}_*,0)\exp\left(\frac{\|\boldsymbol{Gx}\|^2}{2\rho^2\kappa^2 r}\right)+k_\rho\frac{\mathrm{H}(\exp\bar{\sigma}_*,\mathrm{ge}(\sigma_*))}{\mathrm{ge}(\sigma_*)!}\exp\left(\frac{\|\boldsymbol{Gx}\|^2}{2\rho^2\kappa^2 r}\right)(\langle\boldsymbol{\beta},\boldsymbol{Gx}/\rho\kappa\sqrt{r}\rangle)^{\mathrm{ge}(\sigma_*)}+\Xi+\tilde{O}(N^{-1/2}).
$$

Next we turn to the numerator: the expectation of $N^{-1}\sum_{i=1}^{N}(y_i/\rho)\exp(y_i/\rho)\exp(\boldsymbol{x}_i^\top\boldsymbol{Gx}/\rho\kappa\sqrt{r})$ is

$$
\mathbb{E}_{\boldsymbol{x}_1,\zeta_1}\left[(\bar{\sigma}_*(\langle\boldsymbol{\beta},\boldsymbol{x}_1\rangle)+\zeta_1/\rho)\exp(\zeta_1/\rho)\exp(\bar{\sigma}_*(\langle\boldsymbol{\beta},\boldsymbol{x}_1\rangle))\exp(\boldsymbol{x}_1^\top\boldsymbol{Gx}/\rho\kappa\sqrt{r})\right]
$$

$$
=k_\rho\mathbb{E}_{\boldsymbol{x}_1}\left[\sum_{i\geq 0}\frac{1}{i!}\mathrm{H}(\bar{\sigma}_*\exp\bar{\sigma}_*,i)\mathrm{He}_i(\langle\boldsymbol{\beta},\boldsymbol{x}_1\rangle)\exp\left(\frac{\|\boldsymbol{Gx}\|}{\rho\kappa\sqrt{r}}\cdot\frac{\boldsymbol{x}_1^\top\boldsymbol{Gx}}{\|\boldsymbol{Gx}\|}\right)\right]
$$

$$
+k_\rho'\mathbb{E}_{\boldsymbol{x}_1}\left[\sum_{i\geq 0}\frac{1}{i!}\mathrm{H}(\exp\bar{\sigma}_*,i)\mathrm{He}_i(\langle\boldsymbol{\beta},\boldsymbol{x}_1\rangle)\exp\left(\frac{\|\boldsymbol{Gx}\|}{\rho\kappa\sqrt{r}}\cdot\frac{\boldsymbol{x}_1^\top\boldsymbol{Gx}}{\|\boldsymbol{Gx}\|}\right)\right]
$$

where $k_\rho':=\frac{1}{2}\frac{\tau}{\rho}\left(\exp(\tau/\rho)-\exp(-\tau/\rho)\right)=\Theta_d(\tau^2/\rho^2)$. Hence, by the same procedure as the analysis of the denominator, we obtain

$$
N^{-1}\sum_{i=1}^{N}(y_i/\rho)\exp(y_i/\rho)\exp(\boldsymbol{x}_i^\top\boldsymbol{Gx}/\rho\kappa\sqrt{r})
$$

$$=k_\rho \mathrm{H}(\bar\sigma_* \exp\bar\sigma_*, 0)\exp\left(\frac{\|\boldsymbol{Gx}\|^2}{2\rho^2\kappa^2 r}\right) + k_\rho \frac{\mathrm{H}(\bar\sigma_* \exp\bar\sigma_*, \mathrm{ge}(\sigma_*))}{\mathrm{ge}(\sigma_*)!}\exp\left(\frac{\|\boldsymbol{Gx}\|^2}{2\rho^2\kappa^2 r}\right)(\langle\boldsymbol{\beta},\boldsymbol{Gx}/\rho\kappa\sqrt{r}\rangle)^{\mathrm{ge}(\sigma_*)}$$

$$+ k'_\rho \mathrm{H}(\exp\bar\sigma_*, 0)\exp\left(\frac{\|\boldsymbol{Gx}\|^2}{2\rho^2\kappa^2 r}\right) + k'_\rho \frac{\mathrm{H}(\exp\bar\sigma_*, \mathrm{ge}(\sigma_*))}{\mathrm{ge}(\sigma_*)!}\exp\left(\frac{\|\boldsymbol{Gx}\|^2}{2\rho^2\kappa^2 r}\right)(\langle\boldsymbol{\beta},\boldsymbol{Gx}/\rho\kappa\sqrt{r}\rangle)^{\mathrm{ge}(\sigma_*)} + \Xi' + \tilde O(N^{-1/2}),$$

where $|\Xi'| \lesssim \langle\boldsymbol{\beta},\boldsymbol{Gx}/\rho\kappa\sqrt{r}\rangle^{\mathrm{ge}(\sigma_*)+1}$ with high probability. Let us derive the claim of the proposition by analyzing the correlation between the numerator and the denominator. Let $A_0 = \mathrm{H}(\bar\sigma_* \exp\bar\sigma_*, 0), A_{\mathrm{ge}(\sigma_*)} = \mathrm{H}(\bar\sigma_* \exp\bar\sigma_*, \mathrm{ge}(\sigma_*))/(\mathrm{ge}(\sigma_*)!), B_0 = \mathrm{H}(\exp\bar\sigma_*, 0), B_{\mathrm{ge}(\sigma_*)} = \mathrm{H}(\exp\bar\sigma_*, \mathrm{ge}(\sigma_*))/(\mathrm{ge}(\sigma_*)!)$. Note that $A_{\mathrm{ge}(\sigma_*)}$ and $B_{\mathrm{ge}(\sigma_*)}$ are of $\Theta\left(\frac{1}{(\log d)^{C_\rho e_{\mathrm{ge}(\sigma_*)}(\sigma_*)}}\right)$ and then

$$\begin{aligned}
|\Xi|, |\Xi'| &\lesssim \langle\boldsymbol{\beta},\boldsymbol{Gx}/\rho\kappa\sqrt{r}\rangle^{\mathrm{ge}(\sigma_*)+1}\\
&\in O(\log^{\mathrm{ge}(\sigma_*)/2+1/2} d \cdot (\rho\kappa)^{-\mathrm{ge}(\sigma_*)} \cdot (\rho\kappa)^{-1} r^{-\mathrm{ge}(\sigma_*)/2-1/2}) \;(\because \text{ Lemma 17})\\
&\in O(\rho^{-1}\log^{\mathrm{ge}(\sigma_*)/2+1/2} d \cdot (\rho\kappa)^{-\mathrm{ge}(\sigma_*)} \cdot \kappa^{-1} r^{-3\mathrm{ge}(\sigma_*)/4}) \;(\because \mathrm{ge}(\sigma_*) \le 2)\\
&\in o((\rho\kappa)^{-\mathrm{ge}(\sigma_*)} \cdot \kappa^{-1} r^{-3\mathrm{ge}(\sigma_*)/4}) \;(\because \rho = \omega(\log^{\mathrm{ge}(\sigma_*)/2+1/2} d))
\end{aligned}$$

holds from Lemma 17. Moreover, set $N = \tilde\Omega(r^{3\mathrm{ge}(\sigma_*)/2})$ to ensure that $N^{-1/2}$ term is also of $o((\rho\kappa)^{-\mathrm{ge}(\sigma_*)} \cdot \kappa^{-1} r^{-3\mathrm{ge}(\sigma_*)/4})$. In this case, $\Xi, \Xi', N^{-1/2}$ are all of $o((\rho\kappa)^{-\mathrm{ge}(\sigma_*)} A_{\mathrm{ge}(\sigma_*)} r^{-3\mathrm{ge}(\sigma_*)/4}\kappa^{-1/2})$ from the condition $\kappa^{1/2} = \omega(A_{\mathrm{ge}(\sigma_*)}^{-1})$. Then we get, letting $z = \langle\boldsymbol{\beta},\boldsymbol{Gx}\rangle/(\rho\kappa\sqrt{r})$,

$$\begin{aligned}
&g(\boldsymbol{X}, \boldsymbol{y}, \boldsymbol{x}; \boldsymbol{\Gamma})\\
&=\rho\frac{k'_\rho B_0 + k_\rho A_0 + k'_\rho B_{\mathrm{ge}(\sigma_*)} z^{\mathrm{ge}(\sigma_*)} + k_\rho A_{\mathrm{ge}(\sigma_*)} z^{\mathrm{ge}(\sigma_*)} + (\text{h.o.t.})}{k_\rho B_0 + k_\rho B_{\mathrm{ge}(\sigma_*)} z^{\mathrm{ge}(\sigma_*)} + (\text{h.o.t.})}\\
&=\rho\frac{k'_\rho B_0 + k_\rho A_0 + k'_\rho B_{\mathrm{ge}(\sigma_*)} z^{\mathrm{ge}(\sigma_*)} + k_\rho A_{\mathrm{ge}(\sigma_*)} z^{\mathrm{ge}(\sigma_*)} + (\text{h.o.t.})}{k_\rho B_0(1 + \frac{B_{\mathrm{ge}(\sigma_*)}}{B_0} z^{\mathrm{ge}(\sigma_*)} + (\text{h.o.t.}))}\\
&=\rho\frac{k'_\rho B_0 + k_\rho A_0 + k'_\rho B_{\mathrm{ge}(\sigma_*)} z^{\mathrm{ge}(\sigma_*)} + k_\rho A_{\mathrm{ge}(\sigma_*)} z^{\mathrm{ge}(\sigma_*)} + (\text{h.o.t.})}{k_\rho B_0}\left(1 - \frac{B_{\mathrm{ge}(\sigma_*)}}{B_0} z^{\mathrm{ge}(\sigma_*)} + (\text{h.o.t.})\right)\\
&=\rho\left(\frac{k'_\rho B_0 + k_\rho A_0}{k_\rho B_0} + \frac{k_\rho B_0 A_{\mathrm{ge}(\sigma_*)} + k'_\rho B_0 B_{\mathrm{ge}(\sigma_*)} - (k'_\rho B_0 B_{\mathrm{ge}(\sigma_*)} + k_\rho A_0 B_{\mathrm{ge}(\sigma_*)})}{k_\rho B_0^2} z^{\mathrm{ge}(\sigma_*)} + (\text{h.o.t.})\right)\\
&=\rho\left(\frac{k'_\rho B_0 + k_\rho A_0}{k_\rho B_0} + \frac{k_\rho B_0 A_{\mathrm{ge}(\sigma_*)} - k_\rho A_0 B_{\mathrm{ge}(\sigma_*)}}{k_\rho B_0^2} z^{\mathrm{ge}(\sigma_*)} + (\text{h.o.t.})\right)
\end{aligned}$$

Where (h.o.t.) means the terms of $o((\rho\kappa)^{-\mathrm{ge}(\sigma_*)} A_{\mathrm{ge}(\sigma_*)} r^{-3\mathrm{ge}(\sigma_*)/4}\kappa^{-1/2})$. Finally, let us care about the correlation between the numerator and the denominator, that is, verify that $k_\rho B_0 A_{\mathrm{ge}(\sigma_*)} - k_\rho A_0 B_{\mathrm{ge}(\sigma_*)}$ does not vanish. Here recall that $A_0 = \frac{1}{\rho^2}(1 + o_d(1))$ and $B_0 = 1 + o_d(1)$ from Lemma 16, and that $A_{\mathrm{ge}(\sigma_*)}$ and $B_{\mathrm{ge}(\sigma_*)}$ are of the same order. Hence, $k_\rho B_0 A_{\mathrm{ge}(\sigma_*)} = \Theta(A_{\mathrm{ge}(\sigma_*)})$ and $k_\rho A_0 B_{\mathrm{ge}(\sigma_*)} = \Theta((1/\rho^2)A_{\mathrm{ge}(\sigma_*)})$ holds. Then we can conclude that $k_\rho B_0 A_{\mathrm{ge}(\sigma_*)} = \Theta(A_{\mathrm{ge}(\sigma_*)})$ dominates these terms. Then, taking

$$P_1 = \rho\frac{k'_\rho B_0 + k_\rho A_0}{k_\rho B_0}$$

and

$$P_2 = \rho(\rho\kappa)^{-\mathrm{ge}(\sigma_*)}\frac{k_\rho B_0 A_{\mathrm{ge}(\sigma_*)} - k_\rho A_0 B_{\mathrm{ge}(\sigma_*)}}{k_\rho B_0^2}(= \Theta(\rho(\rho\kappa)^{-\mathrm{ge}(\sigma_*)} A_{\mathrm{ge}(\sigma_*)}))$$

yields the assertion. Finally, the order of $\rho\cdot(\text{h.o.t})$ is $o(P_2 r^{-3\mathrm{ge}(\sigma_*)/4}\kappa^{-1/2})$. Using $\kappa^{1/2} = \Omega(\log^{2\deg(\sigma_*)-2} d)$ completes the proof. $\qquad\square$

We are ready to show the following Proposition 8 outlined in Section 5.

**Proposition 8** (Formal). *Suppose $N, \rho$ and $\kappa$ are satisfying the condition in Proposition 11. Let*

$$\mathbf{\Gamma} = \frac{r\mathbb{E}_\beta[\boldsymbol{\beta}\boldsymbol{\beta}^\top] + \mathbf{N}}{\kappa\sqrt{r}}$$

*for some matrix $\mathbf{N}$ satisfying $\|\mathbf{N}\|_F = O(1/\sqrt{d})$. Then,*

$$g(\mathbf{X}, \mathbf{y}, \mathbf{x}; \mathbf{\Gamma}) = P_1 + P_2 \left(\frac{\langle \mathbf{x}, \boldsymbol{\beta}\rangle}{\sqrt{r}}\right)^{\mathrm{ge}(\sigma_*)} + o_d(P_2 r^{-3\mathrm{ge}(\sigma_*)/4} \log^{-2\deg(\sigma_*)+2} d)$$

*holds with high probability over the distribution of the prompt $(\mathbf{X}, \mathbf{y}, \mathbf{x})$, where $P_1 = o_d(1)$ and $P_2 = \Theta_d\left((\log d)^{-C_{P_2}}\right)$.*

*Proof.* From the assumption on the support of $\boldsymbol{\beta}$ (Assumption 4), there exists a $d \times d$ orthogonal matrix $\mathbf{U}$ such that $\mathbf{U}\boldsymbol{\beta} \sim \mathrm{Unif}\{(\alpha_1, \alpha_2, \ldots, \alpha_r, 0, \ldots, 0)|\alpha_1^2 + \cdots + \alpha_r^2 = 1\}$. Here note that $\mathbb{E}[\mathbf{U}\boldsymbol{\beta}\boldsymbol{\beta}^\top\mathbf{U}^\top] = r^{-1}\mathrm{diag}(\underbrace{1, \ldots, 1}_{r}, \underbrace{0, \ldots, 0}_{d-r})$

holds: then, $\mathbf{\Gamma}$ satisfies the decomposition (B.2) with $\mathbf{G} = r\mathbb{E}_\beta[\boldsymbol{\beta}\boldsymbol{\beta}^\top] + \mathbf{N}$.

Let us simplify $\left(\frac{\langle \mathbf{G}\mathbf{x}, \boldsymbol{\beta}\rangle}{\sqrt{r}}\right)^{\mathrm{ge}(\sigma_*)}$ in (B.3). Note that $\langle r\mathbb{E}_\beta[\boldsymbol{\beta}\boldsymbol{\beta}^\top]\mathbf{x}, \boldsymbol{\beta}\rangle = \boldsymbol{\beta}^\top\mathbf{U}^\top\mathrm{diag}(\underbrace{1, \ldots, 1}_{r}, \underbrace{0, \ldots, 0}_{d-r})\mathbf{U}\mathbf{x} =$
$\boldsymbol{\beta}^\top\mathbf{U}^\top\mathbf{U}\mathbf{x} = \langle \mathbf{x}, \boldsymbol{\beta}\rangle$. Moreover, $\frac{\langle \mathbf{N}\mathbf{x}, \boldsymbol{\beta}\rangle}{\sqrt{r}} = o(r^{-1/2}\log^{-2\deg(\sigma_*)+1} d)$ holds with high probability from Lemma 30. Overall,

$$g(\mathbf{X}, \mathbf{y}, \mathbf{x}; \mathbf{\Gamma}) = P_1 + P_2\left(\frac{\langle \mathbf{x}, \boldsymbol{\beta}\rangle}{\sqrt{r}} + o(r^{-1/2}\log^{-2\deg(\sigma_*)+1} d)\right)^{\mathrm{ge}(\sigma_*)} + o_d(P_2 r^{-\mathrm{ge}(\sigma_*)/2}\log^{-2\deg(\sigma_*)+2} d)$$

is satisfied. Then, the case where $\mathrm{ge}(\sigma_*) = 1$ is derived immediately. For the case where $\mathrm{ge}(\sigma_*) = 2$, noting that $\frac{\langle \mathbf{x}, \boldsymbol{\beta}\rangle}{\sqrt{r}} = o(r^{-1/2}\log d)$ with high probability, we can use $(p + q)^2 - p^2 = q(2p + q)$ for the second term. $\square$

## C  Optimizing the Attention Matrix

In this section we develop the optimization guarantee for the attention matrix $\mathbf{\Gamma}$: see Section 5 for the proof sketch. We show that one gradient descent step contains information about $\mathbb{E}[\boldsymbol{\beta}\boldsymbol{\beta}^\top]$, to satisfy the conditions in Proposition 8.

If $f_{\mathrm{TF}}$ is sufficiently small at the initialization, we can approximate the full gradient using the correlational gradient $\frac{1}{T_1}\sum_{t=1}^{T_1} y^t \nabla_{\mathbf{\Gamma}} f_{\mathrm{TF}}(\mathbf{X}^t, \mathbf{y}^t, \mathbf{x}^t; \mathbf{\Gamma}, \mathbf{v}(0), \mathbf{b}(0), \mathbf{a}(0))\big|_{\mathbf{\Gamma}=\mathbf{\Gamma}(0)}$. Hence, we first analyze the model gradient $\nabla_{\mathbf{\Gamma}} f_{\mathrm{TF}}(\mathbf{X}^t, \mathbf{y}^t, \mathbf{x}^t; \mathbf{\Gamma}, \mathbf{v}(0), \mathbf{b}(0), \mathbf{a}(0))$, utilizing the technique developed in Appendix B.

**Lemma 19.** *Let*

$$\boldsymbol{\xi}_1 := \frac{1}{N}\sum_{i=1}^N (y_i/\rho)e^{y_i/\rho}e^{\mathbf{x}_i^\top\mathbf{x}/\rho\sqrt{d}}\mathbf{x}_i, \quad \boldsymbol{\xi}_2 := \frac{1}{N}\sum_{i=1}^N e^{y_i/\rho}e^{\mathbf{x}_i^\top\mathbf{x}/\rho\sqrt{d}}\mathbf{x}_i.$$

*Also, let $A_i = \mathrm{H}(\bar{\sigma}_* \exp\bar{\sigma}_*, i), B_i = \mathrm{H}(\exp\bar{\sigma}_*, i)$ for $i \geq 0$ where $\bar{\sigma}_*(z) := \begin{cases} \frac{\sigma_*(z)}{\rho} & \left(\text{if } \left|\frac{\sigma_*(z)}{\rho}\right| \leq \frac{1}{\log d}\right) \\ 0 & \text{(otherwise)} \end{cases}$.*

*Then, using $k_\rho := \frac{1}{2}(\exp(\tau/\rho) + \exp(-\tau/\rho)) = \Theta_d(1), k'_\rho := \frac{1}{2}\frac{\tau}{\rho}(\exp(\tau/\rho) - \exp(-\tau/\rho)) = \Theta_d(\tau^2/\rho^2)$ and $z := \frac{\langle \boldsymbol{\beta}, \mathbf{x}\rangle}{\rho\sqrt{d}}$, it holds that*

$$\begin{aligned}
\boldsymbol{\xi}_1 =&\, k_\rho \exp\left(\frac{\|\mathbf{x}\|^2}{2\rho^2 d}\right)\left(\sum_{i\geq 0}\frac{A_{i+1}}{i!}z^i\right)\boldsymbol{\beta} + \frac{k_\rho}{\rho\sqrt{d}}\exp\left(\frac{\|\mathbf{x}\|^2}{2\rho^2 d}\right)\left(\sum_{i\geq 0}\frac{A_i}{i!}z^i\right)\mathbf{x} \\
&\, + k'_\rho \exp\left(\frac{\|\mathbf{x}\|^2}{2\rho^2 d}\right)\left(\sum_{i\geq 0}\frac{B_{i+1}}{i!}z^i\right)\boldsymbol{\beta} + \frac{k'_\rho}{\rho\sqrt{d}}\exp\left(\frac{\|\mathbf{x}\|^2}{2\rho^2 d}\right)\left(\sum_{i\geq 0}\frac{B_i}{i!}z^i\right)\mathbf{x}
\end{aligned}$$

$$+ \tilde{O}\left(d^{1/2}N^{-1/2}\right),$$

*and*

$$
\boldsymbol{\xi}_2 = k_\rho \exp\left(\frac{\|\boldsymbol{x}\|^2}{2\rho^2 d}\right) \left(\sum_{i \geq 0} \frac{B_{i+1}}{i!} z^i\right) \boldsymbol{\beta}
$$

$$
+ \frac{k_\rho}{\rho\sqrt{d}} \exp\left(\frac{\|\boldsymbol{x}\|^2}{2\rho^2 d}\right) \left(\sum_{i \geq 0} \frac{B_i}{i!} z^i\right) \boldsymbol{x} + \tilde{O}\left(d^{1/2}N^{-1/2}\right)
$$

*with high probability over the data distribution* [1].

*Proof.* Let us analyze $\boldsymbol{\xi}_2$ first. Note that $\boldsymbol{\xi}_2 = \frac{1}{N}\sum_{i=1}^N \mathrm{e}^{\bar{\sigma}_*(\langle\boldsymbol{\beta},\boldsymbol{x}_i\rangle)+\zeta_i/\rho}\mathrm{e}^{\boldsymbol{x}_i^\top\boldsymbol{x}/\rho\sqrt{d}}\boldsymbol{x}_i$ holds with high probability. The expectation of this quantity is calculated as

$$
\mathbb{E}_{\boldsymbol{x}_1}\left[\mathrm{e}^{\bar{\sigma}_*(\langle\boldsymbol{\beta},\boldsymbol{x}_1\rangle)+\zeta_i/\rho}\mathrm{e}^{\boldsymbol{x}_1^\top\boldsymbol{x}/\rho\sqrt{d}}\boldsymbol{x}_1\right]
$$

$$
= k_\rho\mathbb{E}_{\boldsymbol{x}_1}\left[\mathrm{e}^{\bar{\sigma}_*(\langle\boldsymbol{\beta},\boldsymbol{x}_1\rangle)}\mathrm{e}^{\boldsymbol{x}_1^\top\boldsymbol{x}/\rho\sqrt{d}}\boldsymbol{x}_1\right]
$$

$$
= k_\rho\mathbb{E}_{\boldsymbol{x}_1}\left[\sum_{i \geq 0} \frac{B_i}{i!}\mathrm{He}_i(\langle\boldsymbol{\beta},\boldsymbol{x}_1\rangle)\mathrm{e}^{\boldsymbol{x}_1^\top\boldsymbol{x}/\rho\sqrt{d}}\boldsymbol{x}_1\right]
$$

$$
\overset{(a)}{=} k_\rho\boldsymbol{\beta}\mathbb{E}_{\boldsymbol{x}_1}\left[\sum_{i \geq 0} \frac{B_{i+1}}{i!}\mathrm{He}_i(\langle\boldsymbol{\beta},\boldsymbol{x}_1\rangle)\mathrm{e}^{\boldsymbol{x}_1^\top\boldsymbol{x}/\rho\sqrt{d}}\right] + k_\rho\frac{\boldsymbol{x}}{\rho\sqrt{d}}\mathbb{E}_{\boldsymbol{x}_1}\left[\sum_{i \geq 0} \frac{B_i}{i!}\mathrm{He}_i(\langle\boldsymbol{\beta},\boldsymbol{x}_1\rangle)\mathrm{e}^{\boldsymbol{x}_1^\top\boldsymbol{x}/\rho\sqrt{d}}\right]
$$

$$
\overset{(b)}{=} k_\rho\exp\left(\frac{\|\boldsymbol{x}\|^2}{2\rho^2 d}\right)\left(\sum_{i \geq 0} \frac{B_{i+1}}{i!}z^i\right)\boldsymbol{\beta} + \frac{k_\rho}{\rho\sqrt{d}}\exp\left(\frac{\|\boldsymbol{x}\|^2}{2\rho^2 d}\right)\left(\sum_{i \geq 0} \frac{B_i}{i!}z^i\right)\boldsymbol{x} + \tilde{O}\left(\sqrt{d/N}\right).
$$

Here we used Stein's lemma to derive (a), and the argument to derive (B.6) for (b). Finally, the deviation from the expectation is upper-bounded using the exactly same technique as the proof of Lemma 18: the only difference is that we now treat $d$-dimensional vectors, meaning that the bound coming from Hoeffding's inequality is of $\tilde{O}\left(\sqrt{d/N}\right)$ (Lemma 31).

Similarly, we can analyze $\boldsymbol{\xi}_1$ as

$$
\mathbb{E}_{\boldsymbol{x}_1}\left[(\bar{\sigma}_*(\langle\boldsymbol{\beta},\boldsymbol{x}_1\rangle)+\zeta_i/\rho)\mathrm{e}^{\bar{\sigma}_*(\langle\boldsymbol{\beta},\boldsymbol{x}_1\rangle)+\zeta_i/\rho}\mathrm{e}^{\boldsymbol{x}_1^\top\boldsymbol{x}/\rho\sqrt{d}}\boldsymbol{x}_1\right]
$$

$$
= k_\rho\mathbb{E}_{\boldsymbol{x}_1}\left[\bar{\sigma}_*(\langle\boldsymbol{\beta},\boldsymbol{x}_1\rangle)\mathrm{e}^{\bar{\sigma}_*(\langle\boldsymbol{\beta},\boldsymbol{x}_1\rangle)}\mathrm{e}^{\boldsymbol{x}_1^\top\boldsymbol{x}/\rho\sqrt{d}}\boldsymbol{x}_1\right] + k_\rho'\mathbb{E}_{\boldsymbol{x}_1}\left[\mathrm{e}^{\bar{\sigma}_*(\langle\boldsymbol{\beta},\boldsymbol{x}_1\rangle)}\mathrm{e}^{\boldsymbol{x}_1^\top\boldsymbol{x}/\rho\sqrt{d}}\boldsymbol{x}_1\right]
$$

$$
= k_\rho\exp\left(\frac{\|\boldsymbol{x}\|^2}{2\rho^2 d}\right)\left(\sum_{i \geq 0} \frac{A_{i+1}}{i!}z^i\right)\boldsymbol{\beta} + \frac{k_\rho}{\rho\sqrt{d}}\exp\left(\frac{\|\boldsymbol{x}\|^2}{2\rho^2 d}\right)\left(\sum_{i \geq 0} \frac{A_i}{i!}z^i\right)\boldsymbol{x}
$$

$$
+ k_\rho'\exp\left(\frac{\|\boldsymbol{x}\|^2}{2\rho^2 d}\right)\left(\sum_{i \geq 0} \frac{B_{i+1}}{i!}z^i\right)\boldsymbol{\beta} + \frac{k_\rho'}{\rho\sqrt{d}}\exp\left(\frac{\|\boldsymbol{x}\|^2}{2\rho^2 d}\right)\left(\sum_{i \geq 0} \frac{B_i}{i!}z^i\right)\boldsymbol{x} + \tilde{O}\left(\sqrt{d/N}\right).
$$

$\square$

**Lemma 20.** *Let* $\boldsymbol{\Gamma}(0) = \boldsymbol{I}_d/\sqrt{d}$ *and* $\boldsymbol{v}(0), \boldsymbol{b}(0)$ *and* $\boldsymbol{a}(0)$ *are determined in Algorithm* 1. *Then, assuming* $N_{\mathrm{pt}} = \tilde{\Omega}(r^2 d^{\mathrm{ie}(\sigma_*)+2})$, *with high probability over the data* $(\boldsymbol{X}_{1:N_{\mathrm{pt}}}, \boldsymbol{y}_{1:N_{\mathrm{pt}}}, \boldsymbol{x})$ *(simply written as* $(\boldsymbol{X}, \boldsymbol{y}, \boldsymbol{x})$), *it holds that*

$$
\nabla_{\boldsymbol{\Gamma}} f_{\mathrm{TF}}(\boldsymbol{X}, \boldsymbol{y}, \boldsymbol{x}; \boldsymbol{\Gamma}, \boldsymbol{v}(0), \boldsymbol{b}(0), \boldsymbol{a}(0))|_{\boldsymbol{\Gamma}=\boldsymbol{\Gamma}(0)=\boldsymbol{I}_d/\sqrt{d}}
$$

---

[1] For a vector $\boldsymbol{v}$, $\boldsymbol{v} = \tilde{O}(\cdots)$ means $\|\boldsymbol{v}\| = \tilde{O}(\cdots)$.

$$= \alpha L_m (P_0 + P_1 z + \cdots + P_{\mathrm{ie}(\sigma_*)-1} z^{\mathrm{ie}(\sigma_*)-1} + \cdots) \boldsymbol{\beta} \boldsymbol{x}^\top + \alpha L_m \boldsymbol{n} \boldsymbol{x}^\top$$

*where* $L_m := |\{j \in [m] \mid v_j(0) = 1\}|$, $P_0, \ldots, P_{\mathrm{ie}(\sigma_*)-2} = O(1), P_{\mathrm{ie}(\sigma_*)-1} = \Theta(\rho^{-1})$ *and* $\|\boldsymbol{n}\| \leq O(r^{-1} d^{-(\mathrm{ie}(\sigma_*)+1)/2} \log^{-C_n} d)$ *(where $C_n$ can be taken to be sufficiently large).*

*Proof.* For simplicity we define $N := N_{\mathrm{pt}}$. We can calculate the model gradient as

$$\nabla_{\boldsymbol{\Gamma}} f_{\mathrm{TF}}(\boldsymbol{X}, \boldsymbol{y}, \boldsymbol{x}; \boldsymbol{\Gamma}, \boldsymbol{v}(0), \boldsymbol{b}(0), \boldsymbol{a}(0))|_{\boldsymbol{\Gamma}=\boldsymbol{I}_d/\sqrt{d}}$$

$$= \alpha \sum_{j=1}^m \sigma'\left( v_j \frac{\sum_{i=1}^N y_i \mathrm{e}^{y_i/\rho} \mathrm{e}^{\boldsymbol{x}_i^\top \boldsymbol{x}/\rho\sqrt{d}}}{\sum_{i=1}^N \mathrm{e}^{y_i/\rho} \mathrm{e}^{\boldsymbol{x}_i^\top \boldsymbol{x}/\rho\sqrt{d}}} \right) \cdot v_j \boldsymbol{s}(\boldsymbol{X}, \boldsymbol{y}, \boldsymbol{x}, y) \boldsymbol{x}^\top,$$

where

$$\boldsymbol{s}(\boldsymbol{X}, \boldsymbol{y}, \boldsymbol{x}, y)$$
$$= \frac{(\sum_{i=1}^N (y_i/\rho) \mathrm{e}^{y_i/\rho} \mathrm{e}^{\boldsymbol{x}_i^\top \boldsymbol{x}/\rho\sqrt{d}} \boldsymbol{x}_i)(\sum_{i=1}^N \mathrm{e}^{y_i/\rho} \mathrm{e}^{\boldsymbol{x}_i^\top \boldsymbol{x}/\rho\sqrt{d}}) - (\sum_{i=1}^N (y_i/\rho) \mathrm{e}^{y_i/\rho} \mathrm{e}^{\boldsymbol{x}_i^\top \boldsymbol{x}/\rho\sqrt{d}})(\sum_{i=1}^N \mathrm{e}^{y_i/\rho} \mathrm{e}^{\boldsymbol{x}_i^\top \boldsymbol{x}/\rho\sqrt{d}} \boldsymbol{x}_i)}{(\sum_{i=1}^N \mathrm{e}^{y_i/\rho} \mathrm{e}^{\boldsymbol{x}_i^\top \boldsymbol{x}/\rho\sqrt{d}})^2}.$$

First, we can see from the same procedure as the Proposition 11 (with $r = d$) that the content of $\sigma'(\cdot)$ above is dominated by $P_1(1 + o_d(1))$ with high probability, where $P_1$ in Proposition 11 is positive, which means that $\sigma'\left( v_j \frac{\sum_{i=1}^N y_i \mathrm{e}^{y_i/\rho} \mathrm{e}^{\boldsymbol{x}_i^\top \boldsymbol{x}/\rho\sqrt{d}}}{\sum_{i=1}^N \mathrm{e}^{y_i/\rho} \mathrm{e}^{\boldsymbol{x}_i^\top \boldsymbol{x}/\rho\sqrt{d}}} \right) = 1$ if and only if $v_j = 1$. Then,

$$\nabla_{\boldsymbol{\Gamma}} f_{\mathrm{TF}}(\boldsymbol{X}, \boldsymbol{y}, \boldsymbol{x}; \boldsymbol{\Gamma}, \boldsymbol{v}(0), \boldsymbol{b}(0), \boldsymbol{a}(0))|_{\boldsymbol{\Gamma}=\boldsymbol{I}_d/\sqrt{d}} = \alpha L_m \boldsymbol{s}(\boldsymbol{X}, \boldsymbol{y}, \boldsymbol{x}, y) \boldsymbol{x}^\top$$

holds with high probability. We proceed to calculate $\boldsymbol{s}(\boldsymbol{X}, \boldsymbol{y}, \boldsymbol{x}, y)$ above. From the proof of Proposition 11 and Lemma 19, we have

$$\boldsymbol{s}(\boldsymbol{X}, \boldsymbol{y}, \boldsymbol{x}, y) = (\pi_1(\boldsymbol{X}, \boldsymbol{y}, \boldsymbol{x}, y) \boldsymbol{\xi}_2(\boldsymbol{X}, \boldsymbol{y}, \boldsymbol{x}, y) - \pi_2(\boldsymbol{X}, \boldsymbol{y}, \boldsymbol{x}, y) \boldsymbol{\xi}_1(\boldsymbol{X}, \boldsymbol{y}, \boldsymbol{x}, y))(\pi_2(\boldsymbol{X}, \boldsymbol{y}, \boldsymbol{x}, y))^{-2}, \quad \text{(C.1)}$$

where

$$\pi_1(\boldsymbol{X}, \boldsymbol{y}, \boldsymbol{x}, y) = k_\rho \exp\left( \frac{\|\boldsymbol{x}\|^2}{2\rho^2 d} \right) \left( \sum_{i\geq 0} \frac{A_i}{i!} z^i \right) + k_\rho' \exp\left( \frac{\|\boldsymbol{x}\|^2}{2\rho^2 d} \right) \left( \sum_{i\geq 0} \frac{B_i}{i!} z^i \right) + \tilde{O}(N^{-1/2}),$$

$$\pi_2(\boldsymbol{X}, \boldsymbol{y}, \boldsymbol{x}, y) = k_\rho \exp\left( \frac{\|\boldsymbol{x}\|^2}{2\rho^2 d} \right) \left( \sum_{i\geq 0} \frac{B_i}{i!} z^i \right) + \tilde{O}(N^{-1/2}),$$

and

$$\boldsymbol{\xi}_1 = k_\rho \exp\left( \frac{\|\boldsymbol{x}\|^2}{2\rho^2 d} \right) \left( \sum_{i\geq 0} \frac{A_{i+1}}{i!} z^i \right) \boldsymbol{\beta} + \frac{k_\rho}{\rho\sqrt{d}} \exp\left( \frac{\|\boldsymbol{x}\|^2}{2\rho^2 d} \right) \left( \sum_{i\geq 0} \frac{A_i}{i!} z^i \right) \boldsymbol{x}$$

$$+ k_\rho' \exp\left( \frac{\|\boldsymbol{x}\|^2}{2\rho^2 d} \right) \left( \sum_{i\geq 0} \frac{B_{i+1}}{i!} z^i \right) \boldsymbol{\beta} + \frac{k_\rho'}{\rho\sqrt{d}} \exp\left( \frac{\|\boldsymbol{x}\|^2}{2\rho^2 d} \right) \left( \sum_{i\geq 0} \frac{B_i}{i!} z^i \right) \boldsymbol{x}$$

$$+ \tilde{O}\left( d^{1/2} N^{-1/2} \right),$$

and

$$\boldsymbol{\xi}_2 = k_\rho \exp\left( \frac{\|\boldsymbol{x}\|^2}{2\rho^2 d} \right) \left( \sum_{i\geq 0} \frac{B_{i+1}}{i!} z^i \right) \boldsymbol{\beta}$$

$$+ \frac{k_\rho}{\rho\sqrt{d}} \exp\left(\frac{\|\boldsymbol{x}\|^2}{2\rho^2 d}\right) \left(\sum_{i\geq 0} \frac{B_i}{i!} z^i\right) \boldsymbol{x} + \tilde{O}\left(d^{1/2} N^{-1/2}\right)$$

(recall $z := \frac{\langle \boldsymbol{\beta}, \boldsymbol{x}\rangle}{\rho\sqrt{d}}$).

---

Before proceeding to the further analysis, we note the intuition for the goal of this calculation. We can see that terms proportional to $\boldsymbol{x}$ will be canceled out in (C.1). Therefore, we can expect that (C.1) can be expanded using $z$ as $\sum_{k\geq 0} c_k z^k \cdot \boldsymbol{\beta}$. Thus, $\nabla_{\boldsymbol{\Gamma}} f_{\mathrm{TF}}(\boldsymbol{X}, \boldsymbol{y}, \boldsymbol{x}; \boldsymbol{\Gamma}, \boldsymbol{v}(0), \boldsymbol{b}(0), \boldsymbol{a}(0)) \simeq \alpha L_m \sum_{k\geq 0} c_k z^k \cdot \boldsymbol{\beta}\boldsymbol{x}^\top$ holds. Then, to calculate the expected correlational gradient $\mathbb{E}[y \nabla_{\boldsymbol{\Gamma}} f_{\mathrm{TF}}]$, it suffices to calculate $\mathbb{E}[y \cdot z^k \boldsymbol{\beta}\boldsymbol{x}^\top] = \mathbb{E}[\sigma_*(\langle \boldsymbol{\beta}, \boldsymbol{x}\rangle)(\boldsymbol{\beta}^\top \boldsymbol{x}/\rho\sqrt{d})^k \boldsymbol{\beta}\boldsymbol{x}^\top] = \mathbb{E}[\sigma_*'(\langle \boldsymbol{\beta}, \boldsymbol{x}\rangle)(\boldsymbol{\beta}^\top \boldsymbol{x}/\rho\sqrt{d})^k \boldsymbol{\beta}\boldsymbol{\beta}^\top] + \mathbb{E}[\sigma_*(\langle \boldsymbol{\beta}, \boldsymbol{x}\rangle)k(\boldsymbol{\beta}^\top \boldsymbol{x}/\rho\sqrt{d})^{k-1} \boldsymbol{\beta}\boldsymbol{\beta}^\top/\rho\sqrt{d}]$.

Now, if $k < \mathrm{ie}(\sigma_*) - 1$, the first term $\mathbb{E}[\sigma_*'(\langle \boldsymbol{\beta}, \boldsymbol{x}\rangle)(\boldsymbol{\beta}^\top \boldsymbol{x}/\rho\sqrt{d})^k \boldsymbol{\beta}\boldsymbol{\beta}^\top]$ is zero because a degree-$k$ polynomial does not have Hermite component of degree $\geq \mathrm{ie}(\sigma_*') = \mathrm{ie}(\sigma_*) - 1$. If $k = \mathrm{ie}(\sigma_*) - 1$, then the expectation is nonzero and proportional to $\mathbb{E}[\boldsymbol{\beta}\boldsymbol{\beta}^\top]$, as desired. We can also show that this is the leading term in the entire gradient. Therefore, we focus on checking $z^{\mathrm{ie}(\sigma_*)-1}$ term in (C.1).

---

Let us proceed with the strategy above. Set $N = \tilde{\Omega}(r^2 d^{\mathrm{ie}(\sigma_*)+2})$ to ensure that all the noise terms in $\pi_1, \pi_2, \boldsymbol{\xi}_1, \boldsymbol{\xi}_2$ are of $O(r^{-1} d^{-(\mathrm{ie}(\sigma_*)+1)/2} \log^{-C} d)$ for sufficiently large $C$. Then, we can simplify $s(\boldsymbol{X}, \boldsymbol{y}, \boldsymbol{x}, y)$ as $s(\boldsymbol{X}, \boldsymbol{y}, \boldsymbol{x}, y) = r(\boldsymbol{X}, \boldsymbol{y}, \boldsymbol{x}, y)\boldsymbol{\beta} + O(r^{-1} d^{-(\mathrm{ie}(\sigma_*)+1)/2} \log^{-C} d)$, where

$$r(\boldsymbol{X}, \boldsymbol{y}, \boldsymbol{x}, y)$$
$$= \left[\left(\underbrace{\left(k_\rho\left(\sum_{i=0}^{\mathrm{ie}(\sigma_*)-1} \frac{A_i}{i!} z^i\right) + k_\rho'\left(\sum_{i=0}^{\mathrm{ie}(\sigma_*)-1} \frac{B_i}{i!} z^i\right)\right)}_{(a_1)} \cdot \underbrace{k_\rho\left(\sum_{i=0}^{\mathrm{ie}(\sigma_*)-1} \frac{B_{i+1}}{i!} z^i\right)}_{(a_2)}\right.$$
$$\left. - \underbrace{k_\rho\left(\sum_{i=0}^{\mathrm{ie}(\sigma_*)-1} \frac{B_i}{i!} z^i\right)}_{(b_1)} \underbrace{\left(k_\rho\left(\sum_{i=0}^{\mathrm{ie}(\sigma_*)-1} \frac{A_{i+1}}{i!} z^i\right) + k_\rho'\left(\sum_{i=0}^{\mathrm{ie}(\sigma_*)-1} \frac{B_{i+1}}{i!} z^i\right)\right)}_{(b_2)}\right]$$
$$\cdot \left[\underbrace{(k_\rho B_0)^{-2}(1 - 2w + 3w^2 + \cdots + (-1)^{\mathrm{ie}(\sigma_*)-1} w^{\mathrm{ie}(\sigma_*)-1})}_{(c)}\right] + O(z^{\mathrm{ie}(\sigma_*)})$$

where $w := \frac{B_1}{B_0} z + \frac{B_2}{2B_0} z^2 + \cdots + \frac{B_{\mathrm{ie}(\sigma_*)-1}}{(\mathrm{ie}(\sigma_*)-1)! B_0} z^{\mathrm{ie}(\sigma_*)-1}$. Here we used the fact that $\pi_2 / \exp\left(\frac{\|\boldsymbol{x}\|^2}{2\rho^2 d}\right) = k_\rho B_0\left(1 + \frac{B_1}{B_0} z + \frac{B_2}{2B_0} z^2 + \cdots + \frac{B_{\mathrm{ie}(\sigma_*)-1}}{(\mathrm{ie}(\sigma_*)-1)! B_0} z^{\mathrm{ie}(\sigma_*)-1}\right) + O(z^{\mathrm{ie}(\sigma_*)})$.

There are a lot of terms of degree $z^{\mathrm{ie}(\sigma_*)-1}$ above, but we can show that the terms

$$(k_\rho A_0 + k_\rho' B_0) \cdot k_\rho \frac{B_{\mathrm{ie}(\sigma_*)}}{(\mathrm{ie}(\sigma_*)-1)!} z^{\mathrm{ie}(\sigma_*)-1} \cdot (k^\rho B_0)^{-2} - k_\rho B_0\left(k_\rho \frac{A_{\mathrm{ie}(\sigma_*)}}{(\mathrm{ie}(\sigma_*)-1)!} + k_\rho' \frac{B_{\mathrm{ie}(\sigma_*)}}{(\mathrm{ie}(\sigma_*)-1)!}\right) z^{\mathrm{ie}(\sigma_*)-1} \cdot (k_\rho B_0)^{-2}, \tag{C.2}$$

made from the first term of $(a_1)$ (and $(b_1)$), the last term of $(a_2)$ (and $(b_2)$), and the first term of $(c)$, dominates over other terms, i.e., other terms having degree $z^{\mathrm{ie}(\sigma_*)-1}$ are of $o_d((\text{C.2}))$ considering the coefficients.

First we calculate the order of (C.2). From Lemma 9, $A_i, B_i = \Theta\left(\frac{1}{\rho^{e_i(\sigma_*)}}\right)$ ($i \geq 1$) holds where $e_i$ is defined in Definition 14. Moreover, from Lemma 16 it holds that $A_0 = \Theta(\rho^{-2})$ and $B_0 = \Theta(1)$. Note that from the definition of $e_i$ and the information exponent, we can see that $e_{\mathrm{ie}(\sigma_*)} = 1$ and then $A_{\mathrm{ie}(\sigma_*)}, B_{\mathrm{ie}(\sigma_*)} = O(\rho^{-1})$ holds. Thus, noting that $k_\rho = \Theta(1)$ and $k_\rho' = O(\rho^{-2})$, indeed the term $k_\rho B_0 k_\rho \frac{A_{\mathrm{ie}(\sigma_*)}}{(\mathrm{ie}(\sigma_*)-1)!} (k_\rho B_0)^{-2} = \Theta(\rho^{-1})$ dominates over other terms.

Next we show the superiority of this term over other $z^{\mathrm{ie}(\sigma_*)-1}$ terms (in terms of the degree of $\rho$ in coefficients): all the terms are made by multiplying a term in $(a_1)$ (or $(b_1)$), a term in $(a_2)$ (or $(b_2)$), and a term in $(c)$. Let us see the $\rho$ factor in

the coefficients: note that again from the definition of the information exponent, $e_i(\sigma_*) > 1$ (and then $A_i, B_i = O(\rho^{-2})$) if $1 \le i < \mathrm{ie}(\sigma_*)$. Hence, to be comparable to $\rho^{-1}$ we should pick out the zeroth-order term from $(a_1)$ and $(b_1)$. Here the only optimal way is to pick out the $z^{\mathrm{ie}(\sigma_*)-1}$ term in $(a_2)$ and $(b_2)$, which is nothing but (C.2). Otherwise, we should pick out also the zeroth-order term in $(a_2)$ and $(b_2)$. In this case, we should use $z^{\mathrm{ie}(\sigma_*)-1}$ term in $(c)$, but this is suboptimal because $B_i/B_0$ in $w$ is already $O(\rho^{-2})$.

Getting back to (C.1), we now showed that

$$s(\boldsymbol{X}, \boldsymbol{y}, \boldsymbol{x}, y) = (P_0 + P_1 z + \cdots + P_{\mathrm{ie}(\sigma_*)-1} z^{\mathrm{ie}(\sigma_*)-1} + \cdots)\boldsymbol{\beta} + O(r^{-1} d^{-(\mathrm{ie}(\sigma_*)+1)/2} \log^{-C_n} d)$$

where $P_{\mathrm{ie}(\sigma_*)-1} = \Theta(\rho^{-1})$, which completes the proof. $\qquad\square$

**Lemma 21.** *Let* $\boldsymbol{\Gamma}(0) = \boldsymbol{I}_d/\sqrt{d}$ *and* $\boldsymbol{v}(0), \boldsymbol{b}(0)$ *and* $\boldsymbol{a}(0)$ *are determined in Algorithm 1. Then, assuming* $N_{\mathrm{pt}} = \tilde{\Omega}(r^2 d^{\mathrm{ie}(\sigma_*)+2})$*, with high probability over the data* $(\boldsymbol{X}, \boldsymbol{y}, \boldsymbol{x})$*, it holds that*

$$
\left. \nabla_{\boldsymbol{\Gamma}} \frac{1}{2T_1} \sum_{t=1}^{T_1} \left( f_{\mathrm{TF}}(\boldsymbol{X}^t, \boldsymbol{y}^t, \boldsymbol{x}^t; \boldsymbol{\Gamma}, \boldsymbol{v}(0), \boldsymbol{b}(0), \boldsymbol{a}(0)) - y^t \right)^2 \right|_{\boldsymbol{\Gamma}=\boldsymbol{\Gamma}(0)}
$$
$$
= \Theta(\alpha m d^{-(\mathrm{ie}(\sigma_*)-1)/2} \rho^{-(\mathrm{ie}(\sigma_*))}) \mathbb{E}_{\boldsymbol{\beta}}[\boldsymbol{\beta}\boldsymbol{\beta}^\top] + O(\alpha m r^{-1} d^{-\mathrm{ie}(\sigma_*)/2} \log^{-C_n} d) + \tilde{O}(\alpha^2 m^2 \sqrt{d})
$$

*with high probability over the pretraining data distribution and random initialization, where* $C_n$ *is a sufficiently large constant.*

*Proof.* Now we have

$$
\left. \nabla_{\boldsymbol{\Gamma}} \frac{1}{2T_1} \sum_{t=1}^{T_1} \left( f_{\mathrm{TF}}(\boldsymbol{X}^t, \boldsymbol{y}^t, \boldsymbol{x}^t; \boldsymbol{\Gamma}, \boldsymbol{v}(0), \boldsymbol{b}(0), \boldsymbol{a}(0)) - y^t \right)^2 \right|_{\boldsymbol{\Gamma}=\boldsymbol{\Gamma}(0)} = \frac{1}{T_1} \sum_{t=1}^{T_1} f^t \boldsymbol{G}^t - \frac{1}{T_1} \sum_{t=1}^{T_1} y^t \boldsymbol{G}^t
$$

where $f^t := f_{\mathrm{TF}}(\boldsymbol{X}^t, \boldsymbol{y}^t, \boldsymbol{x}^t; \boldsymbol{\Gamma}(0), \boldsymbol{v}(0), \boldsymbol{b}(0), \boldsymbol{a}(0))$ and $\boldsymbol{G}^t = \nabla_{\boldsymbol{\Gamma}} f_{\mathrm{TF}}(\boldsymbol{X}^t, \boldsymbol{y}^t, \boldsymbol{x}^t; \boldsymbol{\Gamma}, \boldsymbol{v}(0), \boldsymbol{b}(0), \boldsymbol{a}(0))|_{\boldsymbol{\Gamma}=\boldsymbol{\Gamma}(0)}$. We first upper-bound the first term $\frac{1}{T_1} \sum_{t=1}^{T_1} f^t \boldsymbol{G}^t$. From Lemma 15, with high probability $y_i^t = \tilde{O}(1)$ holds for all $i$ and $t$. Therefore,

$$
|f_{\mathrm{TF}}(\boldsymbol{X}^t, \boldsymbol{y}^t, \boldsymbol{x}^t; \boldsymbol{\Gamma}, \boldsymbol{v}(0), \boldsymbol{b}(0), \boldsymbol{a}(0))| \le \alpha m \left| \frac{\sum_{i=1}^{N_{\mathrm{pt}}} y_i e^{c y_i/\rho} e^{\boldsymbol{x}_i^\top \boldsymbol{\Gamma} \boldsymbol{x}/\rho}}{\sum_{i=1}^{N_{\mathrm{pt}}} e^{c y_i/\rho} e^{\boldsymbol{x}_i^\top \boldsymbol{\Gamma} \boldsymbol{x}/\rho}} \right| = \tilde{O}(\alpha m)
$$

holds with high probability.

Also, from Lemma 20, we have

$$
\nabla_{\boldsymbol{\Gamma}} f_{\mathrm{TF}}(\boldsymbol{X}, \boldsymbol{y}, \boldsymbol{x}; \boldsymbol{\Gamma}, \boldsymbol{v}(0), \boldsymbol{b}(0), \boldsymbol{a}(0))|_{\boldsymbol{\Gamma}=\boldsymbol{\Gamma}(0)=\boldsymbol{I}_d/\sqrt{d}}
$$
$$
= \alpha L_m (P_0 + P_1 z + \cdots + P_{\mathrm{ie}(\sigma_*)-1} z^{\mathrm{ie}(\sigma_*)-1} + \cdots)\boldsymbol{\beta}\boldsymbol{x}^\top + \alpha L_m \boldsymbol{n}\boldsymbol{x}^\top
$$

where $P_0, \ldots, P_{\mathrm{ie}(\sigma_*)-2} = O(1)$, $P_{\mathrm{ie}(\sigma_*)-1} = \Theta(\rho^{-1})$ and $\|\boldsymbol{n}\| \le O(r^{-1} d^{-(\mathrm{ie}(\sigma_*)+1)/2} \log^{-C_n} d)$. Moreover, $\|\boldsymbol{x}\| = O(\sqrt{d})$ holds with high probability from Lemma 28, meaning $\|\boldsymbol{G}^t\|_F = O(\alpha m \sqrt{d})$. Overall,

$$
\|\frac{1}{T_1} \sum_{t=1}^{T_1} f^t \boldsymbol{G}^t\|_F = \tilde{O}(\alpha^2 m^2 \sqrt{d}) \tag{C.3}
$$

is ensured with high probability.

Next let us calculate

$$
\frac{1}{T_1} \sum_{t=1}^{T_1} y^t \boldsymbol{G}^t = \frac{\alpha L_m}{T_1} \sum_{t=1}^{T_1} \left( \underbrace{\left( y^t (P_0 + P_1(z^t) + \cdots + P_{\mathrm{ie}(\sigma_*)-1}(z^t)^{\mathrm{ie}(\sigma_*)-1} + \cdots)\boldsymbol{\beta}^t \right)(\boldsymbol{x}^t)^\top}_{=:\boldsymbol{m}^t} + y^t \boldsymbol{n}^t (\boldsymbol{x}^t)^\top \right).
$$

Note that the norm of $\boldsymbol{m}^t$ is $\tilde{O}(1)$ with high probability. Then, $\left|\frac{\alpha L_m}{T_1}\sum_{t=1}^{T_1}\boldsymbol{m}^t(\boldsymbol{x}^t)^\top - \mathbb{E}[\alpha L_m \boldsymbol{m}^1(\boldsymbol{x}^1)^\top]\right| = \tilde{O}(\alpha m d/\sqrt{T_1})$ holds with high probability from Corollary 32. Moreover, $\|\boldsymbol{n}^t(\boldsymbol{x}^t)^\top\|_F \leq O(r^{-1}d^{-\mathrm{ie}(\sigma_*)/2}\log^{-C_n}d\log^{\deg(\sigma_*)/2}d)$ holds (additional $\log^{\deg(\sigma_*)/2}d$ comes from the order of $y$; see Lemma 15. Hereafter we absorb this by retaking $C_n$). Thus, if $T_1 = \tilde{\Omega}(r^2 d^{\mathrm{ie}(\sigma_*)+2})$ then

$$\frac{1}{T_1}\sum_{t=1}^{T_1}y^t\boldsymbol{G}^t = \alpha L_m\mathbb{E}[\boldsymbol{m}\boldsymbol{x}^\top] + O(\alpha m r^{-1}d^{-\mathrm{ie}(\sigma_*)/2}\log^{-C_n}d) \tag{C.4}$$

is satisfied. It remains to calculate the expectation $\mathbb{E}[yz^k\boldsymbol{\beta}\boldsymbol{x}^\top] = (\rho\sqrt{d})^{-k}\mathbb{E}[\sigma_*(\langle\boldsymbol{\beta},\boldsymbol{x}\rangle)\langle\boldsymbol{\beta},\boldsymbol{x}\rangle^k\boldsymbol{\beta}\boldsymbol{x}^\top]$ to analyze $\mathbb{E}[\boldsymbol{m}\boldsymbol{x}^\top]$. From Stein's lemma, we have

$$(\rho\sqrt{d})^{-k}\mathbb{E}_{\boldsymbol{\beta},\boldsymbol{x}}[\sigma_*(\langle\boldsymbol{\beta},\boldsymbol{x}\rangle)\langle\boldsymbol{\beta},\boldsymbol{x}\rangle^k\boldsymbol{\beta}\boldsymbol{x}^\top]$$
$$=(\rho\sqrt{d})^{-k}\mathbb{E}_{\boldsymbol{\beta}}[\mathbb{E}_{\boldsymbol{x}}[\sigma_*'(\langle\boldsymbol{\beta},\boldsymbol{x}\rangle)\langle\boldsymbol{\beta},\boldsymbol{x}\rangle^k]\boldsymbol{\beta}\boldsymbol{\beta}^\top]+(\rho\sqrt{d})^{-k}k\mathbb{E}_{\boldsymbol{\beta}}[\mathbb{E}_{\boldsymbol{x}}[\sigma_*(\langle\boldsymbol{\beta},\boldsymbol{x}\rangle)\langle\boldsymbol{\beta},\boldsymbol{x}\rangle^{k-1}]\boldsymbol{\beta}\boldsymbol{\beta}^\top],$$

and from Lemma 34 and the definition of the information exponent, $\mathbb{E}_{\boldsymbol{x}}[\sigma_*'(\langle\boldsymbol{\beta},\boldsymbol{x}\rangle)\langle\boldsymbol{\beta},\boldsymbol{x}\rangle^k]$ is zero if $k < \mathrm{ie}(\sigma_*') = \mathrm{ie}(\sigma_*)-1$ and nonzero if $k = \mathrm{ie}(\sigma_*)-1$. If $k = \mathrm{ie}(\sigma_*)-1$, then $(\rho\sqrt{d})^{-k}\mathbb{E}_{\boldsymbol{\beta}}[\mathbb{E}_{\boldsymbol{x}}[\sigma_*'(\langle\boldsymbol{\beta},\boldsymbol{x}\rangle)\langle\boldsymbol{\beta},\boldsymbol{x}\rangle^k]\boldsymbol{\beta}\boldsymbol{\beta}^\top] \asymp (\rho\sqrt{d})^{-(\mathrm{ie}(\sigma_*)-1)}\mathbb{E}_{\boldsymbol{\beta}}[\boldsymbol{\beta}\boldsymbol{\beta}^\top]$ is satisfied from Lemma 34. We can observe that other terms are also proportional to $\mathbb{E}[\boldsymbol{\beta}\boldsymbol{\beta}^\top]$ but dominated by this term. From $L_m \asymp m$ with high probability and $P_{\mathrm{ie}(\sigma_*)-1} = \Theta(\rho^{-1})$, we have

$$\alpha L_m\mathbb{E}[\boldsymbol{m}\boldsymbol{x}^\top] = \Theta(\alpha m d^{-(\mathrm{ie}(\sigma_*)-1)/2}\rho^{-(\mathrm{ie}(\sigma_*))})\mathbb{E}_{\boldsymbol{\beta}}[\boldsymbol{\beta}\boldsymbol{\beta}^\top]. \tag{C.5}$$

Putting (C.3), (C.4) and (C.5) together completes the proof. $\qquad\square$

Setting the hyperparameters for Lemma 21 immediately yields the following conclusion of the optimization of $\boldsymbol{\Gamma}$:

**Proposition 22.** *By setting the hyperparameters as $\alpha m = O(r^{-1}d^{-(\mathrm{ie}(\sigma_*)+1)/2}\log^{-C_\alpha}d)$, $N_{\mathrm{pt}}, T_1 = \tilde{\Omega}(r^2 d^{\mathrm{ie}(\sigma_*)+2})$, $\eta_1 = \Theta(\alpha^{-1}m^{-1}d^{(\mathrm{ie}(\sigma_*)-1)/2}\log^{-C_{\eta_1}}d)r^{1/2}$ and $\lambda_1 = \eta_1^{-1}$ for constants $C_{\eta_1}$ and $C_\alpha$, after one gredient descent step (Line 3 in Algorithm 1) it holds that*

$$\boldsymbol{\Gamma}^* = \frac{1}{r^{1/2}\log^{C_\kappa}d}\left(r\mathbb{E}_{\boldsymbol{\beta}}[\boldsymbol{\beta}\boldsymbol{\beta}^\top] + \boldsymbol{N}\right)$$

*with high probability over the data distribution, where $\|\boldsymbol{N}\|_F = O_d(1/\sqrt{d})$ holds, where $C_\kappa$ can be taken to be sufficiently large.*

This proposition can be immediately connected to Proposition 8.

## D  Optimizing the MLP Layer

In this section, we give an optimization guarantee for the MLP layer. We begin with constructing the output layer parameter $\boldsymbol{a} \in \mathbb{R}^m$ yielding a sufficient approximation guarantee.

**Lemma 23** (Construction of Continuous Output Layer). *Let $l = 1$ or $2$ and assume $\mathrm{ge}(\sigma_*) = l$. Suppose that there exists $g(\boldsymbol{x})$ such that*

$$g(\boldsymbol{x}) = P_1 + P_2\frac{(\langle\boldsymbol{\beta},\boldsymbol{x}\rangle)^l}{r^{l/2}} + n(\boldsymbol{x}),$$

*where $P_1 = o_d(1), P_2 = \Theta_d\left((\log d)^{-C_{P_2}}\right)$, and $|n(\boldsymbol{x})| = o_d(P_2 r^{-3l/4}\log^{-2\deg(\sigma_*)+2}d)$ holds (these conditions are indeed identical to the conclusion of Proposition 8). Then, there exists $\pi(v,b)$ such that*

$$\left|\mathbb{E}_{v\sim\mathrm{Unif}\{\pm 1\},b\sim[-1,1]}[\pi(v,b)\sigma(v\cdot g(\boldsymbol{x})+b)] - \sigma_*(\langle\boldsymbol{\beta},\boldsymbol{x}\rangle)\right| = o_d(1)$$

*with high probability over $\boldsymbol{x} \sim \mathcal{N}(0,\boldsymbol{I}_d)$. Moreover, $\sup_{v,b}|\pi(v,b)| = \tilde{O}(r^l)$ holds.*

*Proof.* Note that if $l = 2$, then $\sigma_*$ is even: this means that there exists a polynomial $\varsigma_*$ such that $\sigma_*(z) = \varsigma_*(z^l)$ for $l = 1,2$. Here let $\varsigma_*(z) = \sum_{k=0}^{\deg(\varsigma_*)}s_k z^k$. Now from Lemma 9 in Damian et al. (2022), there exists $\pi_k'(v,b)$ such that $\sup_{v,b}|\pi_k'(v,b)| = O(1)$ and

$$\mathbb{E}_{v\sim\mathrm{Unif}\{\pm 1\},b\sim[-1,1]}[\pi_k'(v,b)\sigma(vz+b)] = z^k$$

for any $|z| \le 1$. If we define

$$\pi'(v, b) = \sum_{k=0}^{\deg(\varsigma_*)} s_k \frac{\pi'_k(v, b\alpha^{-1} \log^{-2} d)}{\alpha \log^2 d} \log^{2k} d,$$

where $\alpha = P_2 r^{-l/2}$, then

$$\mathbb{E}_{v \sim \mathrm{Unif}\{\pm 1\}, b \sim [-\alpha \log^2 d, \alpha \log^2 d]}[\pi'(v, b)\sigma(v(\alpha z) + b)]$$

$$= \sum_{k=0}^{\deg(\varsigma_*)} s_k \mathbb{E}_{v \sim \mathrm{Unif}\{\pm 1\}, b \sim [-\alpha \log^2 d, \alpha \log^2 d]} \left[ \frac{\pi'_k(v, b\alpha^{-1} \log^{-2} d)}{\alpha \log^2 d} \log^{2k} d\sigma(v(\alpha z) + b) \right]$$

$$= \sum_{k=0}^{\deg(\varsigma_*)} s_k \log^{2k} d \, \mathbb{E}_{v \sim \mathrm{Unif}\{\pm 1\}, b \sim [-1,1]} \left[ \pi'_k(v, b)\sigma(vz \log^{-2} d + b) \right] = \sum_{k=0}^{\deg(\varsigma_*)} s_k z^k = \varsigma_*(z),$$

if $|z \log^{-2} d| \le 1$. Note that $\sup_{v,b} |\pi'(v, b)| = O(\alpha^{-1} \log^{2\deg(\sigma_*)-2} d)$ holds.

Furthermore, let us define events $E_-(v, b)$ and $E_+(v, b)$ as

$$E_-(v, b) = (v = -1 \wedge b \in [P_1 \pm \alpha \log^2 d]),$$

$$E_+(v, b) = (v = 1 \wedge b \in [-P_1 \pm \alpha \log^2 d]),$$

and then define

$$\pi(v, b) = \frac{1}{2\alpha \log^2 d}(\mathbb{I}(E_-(v, b))\pi'(-1, b - P_1) + \mathbb{I}(E_+(v, b))\pi'(1, b + P_1)). \tag{D.1}$$

Now it holds that

$$\mathbb{E}_{v \sim \mathrm{Unif}\{\pm 1\}, b \sim [-1,1]}[\pi(v, b)\sigma(v \cdot g(\boldsymbol{x}) + b)]$$

$$= \frac{1}{2} \mathbb{E}_{b \sim [P_1 - \alpha \log^2 d, P_1 + \alpha \log^2 d]} \left[ \pi'(-1, b - P_1)\sigma(-P_1 - \alpha(\langle \boldsymbol{\beta}, \boldsymbol{x} \rangle)^l - n(\boldsymbol{x}) + b) \right]$$

$$+ \frac{1}{2} \mathbb{E}_{b \sim [-P_1 - \alpha \log^2 d, -P_1 + \alpha \log^2 d]} \left[ \pi'(1, b + P_1)\sigma(P_1 + \alpha(\langle \boldsymbol{\beta}, \boldsymbol{x} \rangle)^l + n(\boldsymbol{x}) + b) \right]$$

$$\overset{(*)}{=} \mathbb{E}_{v \sim \mathrm{Unif}\{\pm 1\}, b \sim [-\alpha \log^2 d, \alpha \log^2 d]}[\pi'(v, b)\sigma(v \cdot \alpha \langle \boldsymbol{\beta}, \boldsymbol{x} \rangle^l + b)] + o_d(1) = \sigma_*(\langle \boldsymbol{\beta}, \boldsymbol{x} \rangle) + o_d(1)$$

holds (note that $|\langle \boldsymbol{\beta}, \boldsymbol{x} \rangle^l| \le \log^2 d$ with high probability from the tail bound for the standard Gaussian).

Here, we used the Lipschitz continuity of the ReLU function and the condition $|n(\boldsymbol{x})| = o_d(\alpha \log^{-2\deg(\sigma_*)+2} d)$ to derive (*). □

**Lemma 24** (Construction of Discrete Output Layer). *Under the condition of Lemma 23, there exists $\boldsymbol{a}^* \in \mathbb{R}^m$ such that*

$$\left| \sum_{j=1}^{m} a_j^* \sigma(v_j \cdot g(\boldsymbol{x}) + b_j) - \sigma_*(\langle \boldsymbol{\beta}, \boldsymbol{x} \rangle) \right| = \tilde{O}(r^l m^{-1/2}) + o_d(1) \tag{D.2}$$

*with high probability over $\boldsymbol{x} \sim \mathcal{N}(0, \boldsymbol{I}_d)$. Moreover, $\|\boldsymbol{a}\|^2 = \tilde{O}(r^{2l} m^{-3/2} + r^{3l/2} m^{-1})$ holds with high probability.*

*Proof.* Let $a_j = m^{-1}\pi(v_j, b_j)$ where $\pi$ is constructed in Lemma 23. As $g(\boldsymbol{x}) = O(1)$ and $\sup_{v,b} \pi(v, b) = \tilde{O}(r^l)$, from Hoeffding's inequality we have

$$\left| m^{-1} \sum_{j=1}^{m} \pi(v_j, b_j)\sigma(v_j \cdot g(\boldsymbol{x}) + b_j) - \mathbb{E}_{v,b}[\pi(v, b)\sigma(v \cdot g(\boldsymbol{x}) + b)] \right| = \tilde{O}(r^l m^{-1/2})$$

with high probability and then we obtain (D.2).

Let us upper bound $\|\boldsymbol{a}\|^2 = m^{-1} \cdot m^{-1} \sum_{j=1}^m \pi(v_j, b_j)^2$. Again from Hoeffding's inequality, we obtain

$$\left| m^{-1} \sum_{j=1}^m \pi(v_j, b_j)^2 - \mathbb{E}_{v,b}[\pi(v, b)^2] \right| = \tilde{O}(r^{2l} m^{-1/2})$$

and then it ramains to upper-bound $\mathbb{E}_{v,b}[\pi(v, b)^2]$. Naively $\sup_{v,b} \pi(v, b)^2 = \tilde{O}(r^{2l})$ holds, but from the definition (D.1), $\pi(v, b)$ is nonzero with probability $\tilde{O}(r^{-l/2})$, meaning $\mathbb{E}_{v,b}[\pi(v, b)^2] = \tilde{O}(r^{3l/2})$. Overall, $\|\boldsymbol{a}\|^2 = \tilde{O}(r^{2l} m^{-3/2} + r^{3l/2} m^{-1})$ is satisfied with high probability. $\qquad\square$

**Proof of the Optimization Part of Theorem 1.**   Now we are ready to show the optimization part (part 1) in Theorem 1.

*Proof of Part 1, Theorem 1.* We note that the condition in Lemma 23 is satisfied from Proposition 8, and the condition in Proposition 8 is ensured by Proposition 22. Let $\boldsymbol{a}'$ be the output parameter constructed in Lemma 24: from the equivalence between $\ell_2$-regularized and norm-constrained optimization algorithms, there exists $\lambda_2$ such that

$$\left( \frac{1}{T_2} \sum_{t=T_1+1}^{T_1+T_2} |y^t - f_{\mathrm{TF}}(\boldsymbol{X}^t, \boldsymbol{y}^t, \boldsymbol{x}^t, \boldsymbol{\Gamma}^*, \boldsymbol{v}^*, \boldsymbol{b}^*, \boldsymbol{a}^*)| \right)^2 \leq \frac{1}{T_2} \sum_{t=T_1+1}^{T_1+T_2} \left( y^t - f_{\mathrm{TF}}(\boldsymbol{X}^t, \boldsymbol{y}^t, \boldsymbol{x}^t, \boldsymbol{\Gamma}^*, \boldsymbol{v}^*, \boldsymbol{b}^*, \boldsymbol{a}^*) \right)^2$$

$$\leq \frac{1}{T_2} \sum_{t=T_1+1}^{T_1+T_2} \left( y^t - f_{\mathrm{TF}}(\boldsymbol{X}^t, \boldsymbol{y}^t, \boldsymbol{x}^t, \boldsymbol{\Gamma}^*, \boldsymbol{v}^*, \boldsymbol{b}^*, \boldsymbol{a}') \right)^2$$

$$\leq (\tau + o_d(1))^2 \quad (\because \text{ Lemma } 24, m = \tilde{\Omega}(r^{2\mathrm{ge}(\sigma_*)}))$$

and it holds that $\|\boldsymbol{a}^*\| \leq \|\boldsymbol{a}'\| = \tilde{O}(r^{\mathrm{ge}(\sigma_*)} m^{-3/4} + r^{3\mathrm{ge}(\sigma_*)/4} m^{-1/2})$. Note that as we assumed $m = \tilde{\Omega}(r^{2\mathrm{ge}(\sigma_*)})$, the second term $r^{3\mathrm{ge}(\sigma_*)/4} m^{-1/2}$ is dominant. $\qquad\square$

# E   Inference-Time Estimation Error Analysis

We are ready to analyze the ICL loss at the inference time defined by

$$\mathcal{R}_{N_{\mathrm{test}}}^{\mathrm{ICL}}(\boldsymbol{\Gamma}^*, \boldsymbol{v}^*, \boldsymbol{b}^*, \boldsymbol{a}^*) = \mathbb{E}_{\mathcal{D}_{\boldsymbol{x}}, \mathcal{D}_{f_*}, \mathcal{D}_{\zeta}}[|f_{\mathrm{TF}}(\boldsymbol{X}_{1:N_{\mathrm{test}}}, \boldsymbol{y}_{1:N_{\mathrm{test}}}, \boldsymbol{x}, \boldsymbol{\Gamma}^*, \boldsymbol{v}^*, \boldsymbol{b}^*, \boldsymbol{a}^*) - y|].$$

In particular, our primary interest is in deriving the inference-time sample complexity, i.e., the number of in-context examples $N_{\mathrm{test}}$ in the test prompt needed to ensure that the ICL loss is $o_d(1)$.

## E.1   ICL Error at $N_{\mathrm{test}} = N_{\mathrm{pt}}$

We first treat the special case where $N_{\mathrm{test}} = N_{\mathrm{pt}}$ holds: in this case, as we have an optimization guarantee at context length $N_{\mathrm{pt}}$, it is sufficient to bound generalization gap via a standard Ramemacher complexity analysis.

Recall that $g$ is defined as the attention output, i.e.,

$$g(\boldsymbol{X}, \boldsymbol{y}, \boldsymbol{x}; \boldsymbol{\Gamma}) = \frac{\sum_{i=1}^N y_i \mathrm{e}^{y_i/\rho} \mathrm{e}^{\boldsymbol{x}_i^\top \boldsymbol{\Gamma} \boldsymbol{x}_i/\rho}}{\sum_{i=1}^N \mathrm{e}^{y_i/\rho} \mathrm{e}^{\boldsymbol{x}_i^\top \boldsymbol{\Gamma} \boldsymbol{x}_i/\rho}}.$$

In this section, we fix $\boldsymbol{\Gamma}, \boldsymbol{v}$ and $\boldsymbol{b}$. We define $\mathcal{F}_A$ to describe the set of transformers where the norm of the output layer of the MLP is constrained:

$$\mathcal{F}_A = \left\{ (\boldsymbol{X}, \boldsymbol{y}, \boldsymbol{x}) \mapsto \sum_{j=1}^m a_j \sigma(v_j \cdot g(\boldsymbol{X}, \boldsymbol{y}, \boldsymbol{x}; \boldsymbol{\Gamma}) + b_j) \,\middle|\, \|\boldsymbol{a}\| \leq A \right\}.$$

Moreover, let

$$\mathrm{Rad}_T(\mathcal{F}_A) = \mathbb{E}_{\{\boldsymbol{X}^t\}, \{\boldsymbol{y}^t\}, \{\boldsymbol{x}^t\}, \{\epsilon^t\}} \left[ \sup_{f \in \mathcal{F}_A} \frac{1}{T} \sum_{t=1}^T \epsilon^t f(\boldsymbol{X}^t, \boldsymbol{y}^t, \boldsymbol{x}^t) \right]$$

be its Rademacher complexity, where $\epsilon^t \sim \mathrm{Unif}(\{\pm 1\})$. Here, we fix the context length to be $N = O(\mathrm{poly} d)$, i.e, suppose $\boldsymbol{X} \in \mathbb{R}^{d \times N}$ and $\boldsymbol{y} \in \mathbb{R}^N$. Then, we obtain the upper bound of $\mathrm{Rad}_T(\mathcal{F}_A)$ as follows.

**Lemma 25.** *When $\|\boldsymbol{b}\| \leq B$ and $v_j = \pm 1$, then it holds that*

$$\mathrm{Rad}_T(\mathcal{F}_A) = \tilde{O}\left(\frac{A(B + \sqrt{m})}{\sqrt{T}}\right).$$

*Proof.* From the definition of $\mathcal{F}_A$, we have

$$\mathrm{Rad}_T(\mathcal{F}_A) = \mathbb{E}_{\{\boldsymbol{X}^t\},\{\boldsymbol{y}^t\},\{\boldsymbol{x}^t\},\{\epsilon^t\}}\left[\sup_{\boldsymbol{a}:\|\boldsymbol{a}\|\leq A}\frac{1}{T}\sum_{t=1}^{T}\epsilon^t\sum_{j=1}^{m}a_j\sigma\big(v_j\cdot g(\boldsymbol{X}^t,\boldsymbol{y}^t,\boldsymbol{x}^t;\boldsymbol{\Gamma})+b_j\big)\right]$$

$$= \mathbb{E}_{\{\boldsymbol{X}^t\},\{\boldsymbol{y}^t\},\{\boldsymbol{x}^t\},\{\epsilon^t\}}\left[\sup_{\boldsymbol{a}:\|\boldsymbol{a}\|\leq A}\sum_{j=1}^{m}a_j\cdot\left(\frac{1}{T}\sum_{t=1}^{T}\epsilon^t\sigma\big(v_j\cdot g(\boldsymbol{X}^t,\boldsymbol{y}^t,\boldsymbol{x}^t;\boldsymbol{\Gamma})+b_j\big)\right)\right].$$

From Cauchy–Schwarz inequality and Jensen's inequality, we have

$$\mathrm{Rad}_T(\mathcal{F}_A) \leq \mathbb{E}_{\{\boldsymbol{X}^t\},\{\boldsymbol{y}^t\},\{\boldsymbol{x}^t\},\{\epsilon^t\}}\left[\sup_{\boldsymbol{a}:\|\boldsymbol{a}\|\leq A}\|a\|\cdot\sqrt{\sum_{j=1}^{m}\left(\frac{1}{T}\sum_{t=1}^{T}\epsilon^t\sigma(v_j\cdot g(\boldsymbol{X}^t,\boldsymbol{y}^t,\boldsymbol{x}^t;\boldsymbol{\Gamma})+b_j)\right)^2}\right]$$

$$\leq A\sqrt{\mathbb{E}_{\{\boldsymbol{X}^t\},\{\boldsymbol{y}^t\},\{\boldsymbol{x}^t\},\{\epsilon^t\}}\sum_{j=1}^{m}\left(\frac{1}{T}\sum_{t=1}^{T}\epsilon^t\sigma(v_j\cdot g(\boldsymbol{X}^t,\boldsymbol{y}^t,\boldsymbol{x}^t;\boldsymbol{\Gamma})+b_j)\right)^2}.$$

Next, we obtain the upper bound of the component in the square root. Since $\epsilon^t$ and $\epsilon^{t'}$ are independent if $t \neq t'$, we have

$$\mathbb{E}_{\{\boldsymbol{X}^t\},\{\boldsymbol{y}^t\},\{\boldsymbol{x}^t\},\{\epsilon^t\}}\left[\sum_{j=1}^{m}\left(\frac{1}{T}\sum_{t=1}^{T}\epsilon^t\sigma\big(v_j\cdot g(\boldsymbol{X}^t,\boldsymbol{y}^t,\boldsymbol{x}^t;\boldsymbol{\Gamma})+b_j\big)\right)^2\right]$$

$$= \mathbb{E}_{\{\boldsymbol{X}^t\},\{\boldsymbol{y}^t\},\{\boldsymbol{x}^t\},\{\epsilon^t\}}\left[\frac{1}{T^2}\sum_{j=1}^{m}\sum_{t=1}^{T}\sum_{t'=1}^{T}\epsilon^t\epsilon^{t'}\sigma\big(v_j\cdot g(\boldsymbol{X}^t,\boldsymbol{y}^t,\boldsymbol{x}^t;\boldsymbol{\Gamma})+b_j\big)\sigma\big(v_j\cdot g(\boldsymbol{X}^{t'},\boldsymbol{y}^{t'},\boldsymbol{x}^{t'};\boldsymbol{\Gamma})+b_j\big)\right]$$

$$= \mathbb{E}_{\{\boldsymbol{X}^t\},\{\boldsymbol{y}^t\},\{\boldsymbol{x}^t\}}\left[\frac{1}{T^2}\sum_{j=1}^{m}\sum_{t=1}^{T}\sigma\big(v_j\cdot g(\boldsymbol{X}^t,\boldsymbol{y}^t,\boldsymbol{x}^t;\boldsymbol{\Gamma})+b_j\big)^2\right].$$

Now, $\sigma$ is a ReLU activation function, which means that $\sigma(z)^2 \leq z^2$ for $z \in \mathbb{R}$. Therefore, we have

$$\mathbb{E}_{\{\boldsymbol{X}^t\},\{\boldsymbol{y}^t\},\{\boldsymbol{x}^t\}}\left[\frac{1}{T^2}\sum_{j=1}^{m}\sum_{t=1}^{T}\sigma\big(v_j\cdot g(\boldsymbol{X}^t,\boldsymbol{y}^t,\boldsymbol{x}^t;\boldsymbol{\Gamma})+b_j\big)^2\right]$$

$$\leq \frac{1}{T^2}\mathbb{E}_{\{\boldsymbol{X}^t\},\{\boldsymbol{y}^t\},\{\boldsymbol{x}^t\}}\left[\sum_{j=1}^{m}\sum_{t=1}^{T}\big(v_j\cdot g(\boldsymbol{X}^t,\boldsymbol{y}^t,\boldsymbol{x}^t;\boldsymbol{\Gamma})+b_j\big)^2\right]$$

$$\leq \frac{2}{T^2}\mathbb{E}_{\{\boldsymbol{X}^t\},\{\boldsymbol{y}^t\},\{\boldsymbol{x}^t\}}\left[\sum_{j=1}^{m}\sum_{t=1}^{T}\big(v_j^2\cdot g(\boldsymbol{X}^t,\boldsymbol{y}^t,\boldsymbol{x}^t;\boldsymbol{\Gamma})^2+b_j^2\big)\right]$$

$$\leq \frac{2}{T^2}\left(B^2T+m\sum_{t=1}^{T}\mathbb{E}_{\{\boldsymbol{X}^t\},\{\boldsymbol{y}^t\},\{\boldsymbol{x}^t\}}\big[g(\boldsymbol{X}^t,\boldsymbol{y}^t,\boldsymbol{x}^t;\boldsymbol{\Gamma})^2\big]\right).$$

Finally, we bound the term $\mathbb{E}[g(\boldsymbol{X}^t,\boldsymbol{y}^t,\boldsymbol{x}^t;\boldsymbol{\Gamma})^2]$. For $i = 1,\ldots,N$, let

$$p_i(\boldsymbol{X},\boldsymbol{y},\boldsymbol{x}) = \frac{\mathrm{e}^{y_i/\rho}\mathrm{e}^{\boldsymbol{x}_i^\top\boldsymbol{\Gamma}\boldsymbol{x}/\rho}}{\sum_{i'=1}^{N}\mathrm{e}^{y_{i'}/\rho}\mathrm{e}^{\boldsymbol{x}_{i'}^\top\boldsymbol{\Gamma}\boldsymbol{x}/\rho}}.$$

Then, we have $g(\boldsymbol{X}, \boldsymbol{y}, \boldsymbol{x}; \boldsymbol{\Gamma}) = \sum_{i=1}^{N} p_i(\boldsymbol{X}, \boldsymbol{y}, \boldsymbol{x}) y_i$ and $\sum_{i=1}^{N} p_i(\boldsymbol{X}, \boldsymbol{y}, \boldsymbol{x}) = 1$. Hence, from Jensen's inequality, we have

$$
g(\boldsymbol{X}^t, \boldsymbol{y}^t, \boldsymbol{x}^t; \boldsymbol{\Gamma})^2 = \left( \sum_{i=1}^{N} p_i(\boldsymbol{X}^t, \boldsymbol{y}^t, \boldsymbol{x}^t) y_i^t \right)^2
$$
$$
\leq \sum_{i=1}^{N} p_i(\boldsymbol{X}^t, \boldsymbol{y}^t, \boldsymbol{x}^t)(y_i^t)^2.
$$

Now, we define random variables $Z_i^t, \hat{Z}_i^t \; (i = 1, \ldots N)$ as

$$
Z_i^t = p_i(\boldsymbol{X}^t, \boldsymbol{y}^t, \boldsymbol{x}^t)(y_i^t)^2, \quad \hat{Z}_i^t = p_i(\boldsymbol{X}^t, \boldsymbol{y}^t, \boldsymbol{x}^t)(y_i^t)^2 \cdot \mathbb{I}((y_i^t)^2 \leq (\log d)^{\deg(\sigma_*)+1}).
$$

Thus, using Hölder's inequality, we have

$$
\mathbb{E}_{\{\boldsymbol{X}^t\}, \{\boldsymbol{y}^t\}, \{\boldsymbol{x}^t\}}[Z_i^t - \hat{Z}_i^t]
$$
$$
\leq \mathbb{E}_{\{\boldsymbol{X}^t\}, \{\boldsymbol{y}^t\}, \{\boldsymbol{x}^t\}} \left[ p_i(\boldsymbol{X}^t, \boldsymbol{y}^t, \boldsymbol{x}^t)(y_i^t)^2 \mathbb{I}((y_i^t)^2 \geq (\log d)^{\deg(\sigma_*)+1}) \right]
$$
$$
\leq \mathbb{E}_{\{\boldsymbol{X}^t\}, \{\boldsymbol{y}^t\}, \{\boldsymbol{x}^t\}} \left[ p_i(\boldsymbol{X}^t, \boldsymbol{y}^t, \boldsymbol{x}^t)^4 \right]^{1/4} \mathbb{E}_{\{\boldsymbol{X}^t\}, \{\boldsymbol{y}^t\}, \{\boldsymbol{x}^t\}} \left[ (y_i^t)^4 \right]^{1/2} \mathbb{P} \left[ (y_i^t)^2 \geq (\log d)^{\deg(\sigma_*)+1} \right]^{1/4}
$$
$$
\leq \mathbb{E}_{\{\boldsymbol{X}^t\}, \{\boldsymbol{y}^t\}, \{\boldsymbol{x}^t\}} \left[ (y_i^t)^4 \right]^{1/2} \mathbb{P} \left[ (y_i^t)^2 \geq (\log d)^{\deg(\sigma_*)+1} \right]^{1/4}.
$$

Using Lemma 15 and the definition of high probability events, we can set the quantity above to be $O(N^{-2})$ (note that we assume $N = \mathrm{poly}(d)$). Consequently, we have

$$
\mathbb{E}_{\{\boldsymbol{X}^t\}, \{\boldsymbol{y}^t\}, \{\boldsymbol{x}^t\}} \left[ g(\boldsymbol{X}^t, \boldsymbol{y}^t, \boldsymbol{x}^t; \boldsymbol{\Gamma})^2 \right] \leq \sum_{i=1}^{N} \mathbb{E}[Z_i^t]
$$
$$
\leq O(1) + \sum_{i=1}^{N} \mathbb{E}[\hat{Z}_i^t]
$$
$$
\leq O(1) + \mathbb{E} \left[ \max_{i=1,\ldots,N} \left\{ (y_i^t)^2 \cdot \mathbb{I}((y_i^t)^2 \leq (\log d)^{\deg(\sigma_*)+1}) \right\} \right]
$$
$$
\leq O((\log d)^{\deg(\sigma_*)+1}). \tag{E.1}
$$

In conclusion, we obtain

$$
\mathrm{Rad}_T(\mathcal{F}_A) \leq A \cdot \sqrt{\frac{2}{T^2} \left( B^2 T + mT \cdot \tilde{O}(1) \right)} = \tilde{O} \left( \frac{A(B + \sqrt{m})}{\sqrt{T}} \right).
$$

$\square$

Then, we arrive at the following lemma:

**Lemma 26.** *Suppose running Algorithm 1 under the conditions specified in Theorem 1. Then, with probability at least 0.995 over the training data and the random initialization of Algorithm 1, it holds that $\mathcal{R}_{N_{\mathrm{pt}}}^{\mathrm{ICL}}(\boldsymbol{\Gamma}^*, \boldsymbol{v}^*, \boldsymbol{b}^*, \boldsymbol{a}^*) = o_d(1)$.*

*Proof.* Now we know from Part 1 of Theorem 1 that $\frac{1}{T_2} \sum_{t=T_1+1}^{T_1+T_2} |y^t - f_{\mathrm{TF}}(\boldsymbol{X}^t, \boldsymbol{y}^t, \boldsymbol{x}^t, \boldsymbol{\Gamma}^*, \boldsymbol{v}^*, \boldsymbol{b}^*, \boldsymbol{a}^*)| = o_d(1)$ with high probability. Moreover, we have parameter norm bounds $\|\boldsymbol{a}^*\| = \tilde{O}(r^{3\mathrm{ge}(\sigma_*)/4} m^{-1/2})$ and $\|\boldsymbol{b}\| \leq \sqrt{m}$. Plugging these values into $A$ and $B$ in Lemma 25 yields

$$
\mathrm{Rad}_{T_2}(\mathcal{F}_A) = \tilde{O} \left( \frac{r^{3\mathrm{ge}(\sigma_*)/4}}{\sqrt{T_2}} \right).
$$

Using standard symmetrization technique (cf: Appendix D.3 in (Oko et al., 2024b)) yields $\mathcal{R}_{N_{\mathrm{pt}}}^{\mathrm{ICL}}(\boldsymbol{\theta}) = o_d(1) + \tilde{O} \left( \frac{r^{3\mathrm{ge}(\sigma_*)/4}}{\sqrt{T_2}} \right)$ with probability at least 0.995. Noting that $T_2 = \tilde{\Omega}(r^{3\mathrm{ge}(\sigma_*)/2})$, we arrive at the assertion. $\square$

### E.2 ICL Error at General $N_{\text{test}}$

Finally, we extend the generalization result to any inference-time context length $N_{\text{test}}$, which concludes the proof of Theorem 1.

**Lemma 27.** *For $N \le \text{poly} d$,*

$$\mathbb{E}_{\boldsymbol{X}, \boldsymbol{y}, \boldsymbol{x}}[f_{\text{TF}}(\boldsymbol{X}_{1:N}, \boldsymbol{y}_{1:N}, \boldsymbol{x}; \boldsymbol{\Gamma}^*, \boldsymbol{v}^*, \boldsymbol{b}^*, \boldsymbol{a}^*)^2] \le \tilde{O}(r^{3\text{ge}(\sigma_*)/2})$$

*holds.*

*Proof.* Note that

$$f_{\text{TF}}(\boldsymbol{X}_{1:N}, \boldsymbol{y}_{1:N}, \boldsymbol{x}; \boldsymbol{\Gamma}^*, \boldsymbol{v}^*, \boldsymbol{b}^*, \boldsymbol{a}^*) = \sum_{j=1}^{m} a_j^* \sigma(v_j^* g(\boldsymbol{X}_{1:N}, \boldsymbol{y}_{1:N}, \boldsymbol{x}; \boldsymbol{\Gamma}^*) + b_j^*)$$

where $\mathbb{E}[g(\boldsymbol{X}_{1:N}, \boldsymbol{y}_{1:N}, \boldsymbol{x}; \boldsymbol{\Gamma}^*)^2] = \tilde{O}(1)$ from (E.1). Then,

$$\mathbb{E}[f_{\text{TF}}(\boldsymbol{X}_{1:N}, \boldsymbol{y}_{1:N}, \boldsymbol{x}; \boldsymbol{\Gamma}^*, \boldsymbol{v}^*, \boldsymbol{b}^*, \boldsymbol{a}^*)^2]$$

$$\le \mathbb{E}\left[\|\boldsymbol{a}^*\|^2 \cdot \sum_{j=1}^{m} \sigma(v_j^* g(\boldsymbol{X}_{1:N}, \boldsymbol{y}_{1:N}, \boldsymbol{x}; \boldsymbol{\Gamma}^*) + b_j^*)^2\right]$$

$$\le \mathbb{E}\left[\|\boldsymbol{a}^*\|^2 \cdot \sum_{j=1}^{m} (2g(\boldsymbol{X}_{1:N}, \boldsymbol{y}_{1:N}, \boldsymbol{x}; \boldsymbol{\Gamma}^*)^2 + 2(b_j^*)^2)\right] \le \tilde{O}(r^{3\text{ge}(\sigma_*)/2})$$

holds. $\qquad\square$

Finally, we prove the Part 2 of Theorem 1.

*Proof of Part 2, Theorem 1.* Note that

$$|\mathcal{R}_{N_{\text{pt}}}^{\text{ICL}}(\boldsymbol{\Gamma}^*, \boldsymbol{v}^*, \boldsymbol{b}^*, \boldsymbol{a}^*) - \mathcal{R}_{N_{\text{test}}}^{\text{ICL}}(\boldsymbol{\Gamma}^*, \boldsymbol{v}^*, \boldsymbol{b}^*, \boldsymbol{a}^*)|$$

$$\le \mathbb{E}[|f_{\text{TF}}(\boldsymbol{X}_{1:N_{\text{pt}}}, \boldsymbol{y}_{1:N_{\text{pt}}}, \boldsymbol{x}; \boldsymbol{\Gamma}^*, \boldsymbol{v}^*, \boldsymbol{b}^*, \boldsymbol{a}^*) - f_{\text{TF}}(\boldsymbol{X}_{1:N_{\text{test}}}, \boldsymbol{y}_{1:N_{\text{test}}}, \boldsymbol{x}; \boldsymbol{\Gamma}^*, \boldsymbol{v}^*, \boldsymbol{b}^*, \boldsymbol{a}^*)|]$$

holds. From Proposition 11, if $N_{\text{pt}}, N_{\text{test}} = \tilde{\Omega}(r^{3\text{ge}(\sigma_*)/2})$, then

$$|g(\boldsymbol{X}_{1:N_{\text{test}}}, \boldsymbol{y}_{1:N_{\text{test}}}, \boldsymbol{x}; \boldsymbol{\Gamma}^*) - g(\boldsymbol{X}_{1:N_{\text{pt}}}, \boldsymbol{y}_{1:N_{\text{pt}}}, \boldsymbol{x}; \boldsymbol{\Gamma}^*)| \le o_d(r^{-3\text{ge}(\sigma_*)/4} \log^{-C} d)$$

holds with high probability (here, $C$ can be made sufficiently large by taking the constant $C_\kappa$ to be sufficiently large), where the upper bound quantity at right hand side does not depend on the data. Note that

$$|f_{\text{TF}}(\boldsymbol{X}_{1:N_{\text{pt}}}, \boldsymbol{y}_{1:N_{\text{pt}}}, \boldsymbol{x}; \boldsymbol{\Gamma}^*, \boldsymbol{v}^*, \boldsymbol{b}^*, \boldsymbol{a}^*) - f_{\text{TF}}(\boldsymbol{X}_{1:N_{\text{test}}}, \boldsymbol{y}_{1:N_{\text{test}}}, \boldsymbol{x}; \boldsymbol{\Gamma}^*, \boldsymbol{v}^*, \boldsymbol{b}^*, \boldsymbol{a}^*)|$$

$$\le \sum_{j=1}^{m} |a_j^*| \left|\sigma(v_j^* g(\boldsymbol{X}_{1:N_{\text{pt}}}, \boldsymbol{y}_{1:N_{\text{pt}}}, \boldsymbol{x}; \boldsymbol{\Gamma}^*) + b_j^*) - \sigma(v_j^* g(\boldsymbol{X}_{1:N_{\text{test}}}, \boldsymbol{y}_{1:N_{\text{test}}}, \boldsymbol{x}; \boldsymbol{\Gamma}^*) + b_j^*)\right|$$

$$\le \|\boldsymbol{a}^*\| \cdot \sqrt{m} \cdot o_d(r^{-3\text{ge}(\sigma_*)/4} \log^{-C} d) \quad (\because \text{ Cauchy-Schwarz}).$$

Using $\|\boldsymbol{a}^*\| = \tilde{O}(r^{3\text{ge}(\sigma_*)/4} m^{-1/2})$ in Part 1 of Theorem 1, now we obtain

$$|f_{\text{TF}}(\boldsymbol{X}_{1:N_{\text{pt}}}, \boldsymbol{y}_{1:N_{\text{pt}}}, \boldsymbol{x}; \boldsymbol{\Gamma}^*, \boldsymbol{v}^*, \boldsymbol{b}^*, \boldsymbol{a}^*) - f_{\text{TF}}(\boldsymbol{X}_{1:N_{\text{test}}}, \boldsymbol{y}_{1:N_{\text{test}}}, \boldsymbol{x}; \boldsymbol{\Gamma}^*, \boldsymbol{v}^*, \boldsymbol{b}^*, \boldsymbol{a}^*)| \le o_d(1)$$

with high probability. It remains to convert this bound into the expectation. Note that the $o_d(1)$ quantity at right-hand side does not depend on the data: we write this quantity as $q_d$. We can state that $\mathbb{P}[|f_{\text{TF}}(\boldsymbol{X}_{1:N_{\text{pt}}}, \boldsymbol{y}_{1:N_{\text{pt}}}, \boldsymbol{x}; \boldsymbol{\Gamma}^*, \boldsymbol{v}^*, \boldsymbol{b}^*, \boldsymbol{a}^*) - f_{\text{TF}}(\boldsymbol{X}_{1:N_{\text{test}}}, \boldsymbol{y}_{1:N_{\text{test}}}, \boldsymbol{x}; \boldsymbol{\Gamma}^*, \boldsymbol{v}^*, \boldsymbol{b}^*, \boldsymbol{a}^*)| \le q_d] \le 1 - d^{-4}$ from the definition of the high probability event. We write this event as $E$. Consequently,

$$\mathbb{E}[|f_{\text{TF}}(\boldsymbol{X}_{1:N_{\text{pt}}}, \boldsymbol{y}_{1:N_{\text{pt}}}, \boldsymbol{x}; \boldsymbol{\Gamma}^*, \boldsymbol{v}^*, \boldsymbol{b}^*, \boldsymbol{a}^*) - f_{\text{TF}}(\boldsymbol{X}_{1:N_{\text{test}}}, \boldsymbol{y}_{1:N_{\text{test}}}, \boldsymbol{x}; \boldsymbol{\Gamma}^*, \boldsymbol{v}^*, \boldsymbol{b}^*, \boldsymbol{a}^*)|]$$

$$\leq q_d + \mathbb{E}[|f_{\mathrm{TF}}(\boldsymbol{X}_{1:N_{\mathrm{pt}}}, \boldsymbol{y}_{1:N_{\mathrm{pt}}}, \boldsymbol{x}; \boldsymbol{\Gamma}^*, \boldsymbol{v}^*, \boldsymbol{b}^*, \boldsymbol{a}^*) - f_{\mathrm{TF}}(\boldsymbol{X}_{1:N_{\mathrm{test}}}, \boldsymbol{y}_{1:N_{\mathrm{test}}}, \boldsymbol{x}; \boldsymbol{\Gamma}^*, \boldsymbol{v}^*, \boldsymbol{b}^*, \boldsymbol{a}^*)|(1 - \mathbb{I}(E))]$$

$$\leq q_d + \mathbb{E}[|f_{\mathrm{TF}}(\boldsymbol{X}_{1:N_{\mathrm{pt}}}, \boldsymbol{y}_{1:N_{\mathrm{pt}}}, \boldsymbol{x}; \boldsymbol{\Gamma}^*, \boldsymbol{v}^*, \boldsymbol{b}^*, \boldsymbol{a}^*) - f_{\mathrm{TF}}(\boldsymbol{X}_{1:N_{\mathrm{test}}}, \boldsymbol{y}_{1:N_{\mathrm{test}}}, \boldsymbol{x}; \boldsymbol{\Gamma}^*, \boldsymbol{v}^*, \boldsymbol{b}^*, \boldsymbol{a}^*)|^2]^{1/2} \cdot (d^{-4})^{1/2}$$

$$\leq q_d + \mathbb{E}[2 f_{\mathrm{TF}}(\boldsymbol{X}_{1:N_{\mathrm{pt}}}, \boldsymbol{y}_{1:N_{\mathrm{pt}}}, \boldsymbol{x}; \boldsymbol{\Gamma}^*, \boldsymbol{v}^*, \boldsymbol{b}^*, \boldsymbol{a}^*)^2 + 2 f_{\mathrm{TF}}(\boldsymbol{X}_{1:N_{\mathrm{test}}}, \boldsymbol{y}_{1:N_{\mathrm{test}}}, \boldsymbol{x}; \boldsymbol{\Gamma}^*, \boldsymbol{v}^*, \boldsymbol{b}^*, \boldsymbol{a}^*)^2]^{1/2} \cdot (d^{-4})^{1/2}$$

$$\leq q_d + \tilde{O}(r^{3\mathrm{ge}(\sigma_*)/4} \cdot d^{-2})$$

From Lemma 27. Then, from $\mathrm{ge}(\sigma_*) \leq 2$ and $r \leq d$, we obtain $|\mathcal{R}_{N_{\mathrm{pt}}}^{\mathrm{ICL}}(\boldsymbol{\Gamma}^*, \boldsymbol{v}^*, \boldsymbol{b}^*, \boldsymbol{a}^*) - \mathcal{R}_{N_{\mathrm{test}}}^{\mathrm{ICL}}(\boldsymbol{\Gamma}^*, \boldsymbol{v}^*, \boldsymbol{b}^*, \boldsymbol{a}^*)| = o_d(1)$
and arrive at the conclusion from Lemma 26. $\qquad\square$

# F  Auxiliary Lemmas

## F.1  High Probability Events on Gaussian and Spherical Variables

**Lemma 28** ((Wainwright, 2019), Example 2.11). *Let $z \sim \sqrt{x_1^2 + \cdots + x_r^2}$ where $\boldsymbol{x} \sim \mathcal{N}(0, \boldsymbol{I}_r)$. Then, for any $0 < t < 1$,*

$$\mathbb{P}[|z^2 - r| \geq rt] \leq 2\exp(-rt^2/8)$$

*holds. Consequently, $z^2 \leq O(r\sqrt{\log d})$ holds with high probability, and if $r = d$, $d/2 \leq z^2 \leq 3d/2$ holds with high probability.*

**Lemma 29.** *Let $r \leq d$, $\boldsymbol{x} \sim \mathcal{N}(0, \boldsymbol{I}_r)$, $\boldsymbol{y} \sim \mathcal{N}(0, \boldsymbol{I}_r)$ independent of $\boldsymbol{x}$, and a $r \times r$ matrix $\boldsymbol{G}$ dependent of neither $\boldsymbol{x}$ nor $\boldsymbol{y}$. Then, $|\langle \boldsymbol{x}, \boldsymbol{G}\boldsymbol{y} \rangle| = O(\|\boldsymbol{G}\|_2 \sqrt{r\log d})$ holds with high probability.*

*Proof.* $\boldsymbol{x}$ can be decomposed as $\boldsymbol{x} = z\boldsymbol{\beta}$, where $\boldsymbol{\beta} \sim \mathrm{Unif}(\mathbb{S}^{r-1})$ and $z = \|\boldsymbol{x}\|$ independent of $\boldsymbol{\beta}$. Thus, $\langle \boldsymbol{x}, \boldsymbol{G}\boldsymbol{y} \rangle = \langle z\boldsymbol{G}^\top\boldsymbol{\beta}, \boldsymbol{y} \rangle$ holds. From Lemma 28 we have $\|z\boldsymbol{G}^\top\boldsymbol{\beta}\|^2 = O(\|\boldsymbol{G}\|_2^2 r\sqrt{\log d})$ with high probability, and if we fix $\boldsymbol{x}$ satisfying this, then $\langle z\boldsymbol{G}^\top\boldsymbol{\beta}, \boldsymbol{y} \rangle^2 = O(\|\boldsymbol{G}\|_2^2 r\sqrt{\log d} \cdot \sqrt{\log d})$ with high probability again from Lemma 28 (in the case $r = 1$). This yields the assertion. $\qquad\square$

**Corollary 30.** *Let $r \leq d$, $\boldsymbol{\beta} \sim \mathrm{Unif}(\mathbb{S}^{r-1})$, $\boldsymbol{y} \sim \mathcal{N}(0, \boldsymbol{I}_r)$ independent of $\boldsymbol{\beta}$, and a $r \times r$ matrix $\boldsymbol{G}$ dependent of neither $\boldsymbol{x}$ nor $\boldsymbol{y}$. Then, $|\langle \boldsymbol{\beta}, \boldsymbol{G}\boldsymbol{y} \rangle| = O(\|\boldsymbol{G}\|_2 \sqrt{\log d})$ holds with high probability.*

**Lemma 31.** *Let $\boldsymbol{x}_1, \ldots, \boldsymbol{x}_N \sim \mathcal{N}(0, \boldsymbol{I}_d)$ and let $z_1, \ldots, z_N$ be i.i.d. random variables satisfying $|z_i| \leq C$ ($z_i$ might depend on $\boldsymbol{x}_i$) with high probability. Then,*

$$\left\| N^{-1} \sum_{i=1}^N z_i \boldsymbol{x}_i - \mathbb{E}[z_1 \boldsymbol{x}_1] \right\| \leq \tilde{O}(C\sqrt{d/N})$$

*holds with high probability.*

*Proof.* Define $z_i' = \mathbb{I}(|z_i| \leq C)z_i$. Note that $z_i'\boldsymbol{x}_i$ is a sub-Gaussian vector, i.e., $\langle z_i'\boldsymbol{x}_i, \boldsymbol{u} \rangle$ is $C$-sub Gaussian for any $\boldsymbol{u} \in \mathrm{Unif}(\mathbb{S}^{d-1})$. Then, a standard concentration bound for sub-Gaussian vectors yields

$$\left\| N^{-1} \sum_{i=1}^N z_i' \boldsymbol{x}_i - \mathbb{E}[z_1' \boldsymbol{x}_1] \right\| \leq \tilde{O}(C\sqrt{d/N})$$

with high probability. As $N^{-1}\sum_{i=1}^N z_i\boldsymbol{x}_i = N^{-1}\sum_{i=1}^N z_i'\boldsymbol{x}_i$ with high probability, it remains to show that $\|\mathbb{E}[z_1'\boldsymbol{x}_1] - \mathbb{E}[z_1\boldsymbol{x}_1]\|$ is sufficiently small. This can be seen by

$$\begin{aligned}
\|\mathbb{E}[z_1'\boldsymbol{x}_1] - \mathbb{E}[z_1\boldsymbol{x}_1]\| &= \|\mathbb{E}[\mathbb{I}(|z_1| \geq C)\boldsymbol{x}_1]\| \\
&= \mathbb{E}[\mathbb{I}(|z_1| \geq C)^2]^{1/2}[\|\boldsymbol{x}_1\|^2]^{1/2} \\
&\leq O(d^{-C_*}).
\end{aligned}$$

where $C_*$ can be taken to be sufficiently large from the definition of the high-probability event. This completes the proof. $\qquad\square$

Applying Lemma 31 to each row yields the following:

**Corollary 32.** *Let* $x_1, \dots, x_N \sim \mathcal{N}(0, I_d)$ *and let* $v_1, \dots, v_N$ *be i.i.d. random vectors satisfying* $\|v_i\| \leq C$ ($v_i$ *might depend on* $x_i$) *with high probability. Then,*

$$\left\| \sum_{i=1}^N v_i x_i - \mathbb{E}[v_1 x_1] \right\|_F \leq \tilde{O}(Cd/\sqrt{N})$$

*holds with high probability.*

### F.2 Hermite Polynomials

**Lemma 33.** *For* $g(z) = \sum_{i \geq 0} \frac{c_i}{i!} \mathrm{He}_i(z)$,

$$\sum_{i \geq 0} \frac{c_i^2}{i!} = \mathbb{E}_{z \sim \mathcal{N}(0,1)}[g(z)^2]$$

*holds.*

**Lemma 34** ((Damian et al., 2023), Property 1). *For* $\alpha, \beta \in \mathbb{S}^{d-1}$,

$$\mathbb{E}_{x \sim \mathcal{N}(0, I_d)}[\mathrm{He}_i(\langle x, \alpha \rangle) \mathrm{He}_j(\langle x, \beta \rangle)] = i! \cdot \mathbb{I}(i = j) \langle \alpha, \beta \rangle^i$$

*holds.*

## G Experimental Details

For the experiment in Section 6, we used a 6-layer GPT-2 model (Radford et al., 2019) following the configuration used in (Garg et al., 2022; Oko et al., 2024b). Precisely, the model take the length-$(N+1)$ prompt $(\boldsymbol{X}_{1:N+1}, \boldsymbol{y}_{1:N+1}, \boldsymbol{x})$ as input. Then, we first make the embedding as

$$\boldsymbol{E} = \boldsymbol{W}^{\mathrm{in}} \left[ \boldsymbol{x}_1, \boldsymbol{y}_1^{\mathrm{pad}}, \dots, \boldsymbol{x}_{N+1}, \boldsymbol{y}_{N+1}^{\mathrm{pad}} \right] \in \mathbb{R}^{d \times (2N+2)} \tag{G.1}$$

where $\boldsymbol{y}_i^{\mathrm{pad}} = [y_i, 0, \dots, 0]^\top$ and $\boldsymbol{W}^{\mathrm{in}} \in \mathbb{R}^{D \times d}$ is a trainable read-in parameter. A 6-layers GPT-2 backbone with 4 attention heads, whose configuration is same as that in Garg et al. (2022) then convert this embedding into $\boldsymbol{E}' \in \mathbb{R}^{D \times (2N+2)}$ where $D = 256$, and we obtain

$$[z_1, \dots, z_{2N+2}] = (\boldsymbol{w}^{\mathrm{out}})^\top \boldsymbol{E}' \tag{G.2}$$

where $\boldsymbol{w}^{\mathrm{out}} \in \mathbb{R}^D$ is also trainable. Here, we use $z_{2i-1}$ as the prediction of $y_i$ based on $x_1, y_1, \dots, x_{i-1}, y_{i-1}$ and $x_i$. Causal attention mask ensure that there is no information leakage. We pretrain the model using the Adam (Kingma & Ba, 2015) optimizer with learning rate 0.0001 on the mean-squared loss calculated over all the positions. We utilize curriculum learning setting which is also adopted in (Garg et al., 2022) to reduce the pretraining cost. we start from $d = 2$ and increased the input dimensionality of by two until the target dimensionality.

The test loss is averaged over 256 independent runs, where each run contains 8,192 independent batches.

