# OpenReview forum: "Nonlinear transformers can perform inference-time feature learning"
_ICML.cc/2025/Conference — ICML 2025 poster_

### Official Review · Reviewer_asP8 · 2025-03-10

**Overall Recommendation:** 3

**Summary:**

This paper studies the in-context learning capacities of transformers when the prompt sequences are given by a (possibly low-dimensional) Gaussian single-index model. When the length of the prompt sequences exceed certain (information-theoretic) limit, transformer trained with modified gradient descent method is shown to achieve vanishing in-context prediction error.

**Claims And Evidence:**

N/A

**Essential References Not Discussed:**

N/A

**Experimental Designs Or Analyses:**

N/A

**Methods And Evaluation Criteria:**

N/A

**Other Comments Or Suggestions:**

It is a bit confusing that the dependency on $\log^{\mathrm{deg}(\sigma)}(d)$ is hidden in the complexity bounds.

**Other Strengths And Weaknesses:**

N/A

**Questions For Authors:**

As I understand it, the techniques developed in this paper (and prior works) are specific to the Gaussian single-index model. To what extent can they be extended beyond this setting, such as by relaxing the Gaussian assumption?

**Relation To Broader Scientific Literature:**

In-context learning capability of transformers (the backbone of LLMs) is an important topic of both theoretical and practical relevance.

**Theoretical Claims:**

I generally feel that the theoretical results in this paper, to some extent, illustrate the in-context learning capabilities of transformers. The mechanism of how the learned transformer performs feature learning in single-index model is well-explained.

However, I have concerns about the “gradient-based training” (Algorithm 1) analyzed in this work, particularly given the setting under consideration:

(1) The K and Q matrices (as well as the V matrix and the matrix from the MLP layer) are merged into a single matrix, with certain coordinates fixed at 0 or 1 (i.e., they are not trained).

(2) In the analysis of optimization, the weight matrices of the attention layer are trained using only a single step of gradient descent.

Given that the current analysis is already quite complex, it is unclear whether these simplifications are essential to the theoretical claims. In particular, simplification (1) ensures that the output takes the exact form of Equation (3.2), which computes a weighted average over the labels $y_i$ before applying the nonlinear MLP. Additionally, the use of a single gradient descent step suggests the analysis of optimization operates in the NTK regime, which is a clear simplification for theoretical tractability.

Overall, I believe the paper would benefit from clearer explanations and a more detailed justification of this gap.

---

> ### Author Rebuttal · Authors · 2025-04-01
>
> We thank the reviewer for the helpful feedback.  We address the technical comments and questions below.
>
> **On Algorithm 1**
>
> We make the following remarks on Algorithm 1.
> - The strategy of merging attention matrices and zeroing out some submatrices is extensively used in theoretical analysis of both linear and nonlinear transformers [Ahn et al. 2023; Mahankali et al. 2023; Kim and Suzuki 2024a; Oko et al. 2024b] – even with such simplifications, theories can reproduce fruitful insights into attention modules.  We also empirically demonstrated in [[link]](https://osf.io/p87yc?view_only=be342dc212bb49cb8f72309fdc37d376) (see also our response to reviewer 7xsp for details) that pretraining GPT2 model can achieve an test-time sample complexity that goes beyond the CSQ lower bound.  We believe this suggests that mechanisms similar to our theoretical construction may be present in more realistic settings.
> - Our layer-wise training procedure is inspired by recent works in the feature learning theory literature, which simplifies statistical and optimization analyses. In particular, a single gradient descent step (Stage I) has been used to model the early phase of gradient-based training on neural networks [[Ba et al. 2022]](https://openreview.net/forum?id=akddwRG6EGi) [[Damian et al. 2022]](https://proceedings.mlr.press/v178/damian22a/damian22a.pdf) and transformers [Oko et al. 2024b], during which the first-layer parameters align with the feature vector of the single-index target and escape saddle points in the nonconvex landscape.  The layer-wise training strategy – dividing the algorithm into two stages, one gradient descent step on the inner layer and ridge regression for the outer layer – is also common in these prior works, where the lower layer captures the relevant feature, while the top layer (+ reinitialized bias units) approximates the link function.
>
>
> **On the NTK Regime**
>
> We would like to clarify that our analysis is *not* in the NTK/lazy regime. Quite the contrary, the one-step feature learning paradigm has been proposed to establish statistical separations and superiority over the NTK, due to the adaptivity to low-dimensional structure – see [Ba et al. 2022][Damian et al. 2022] for discussion.
>
> **Beyond Gaussian Single-index Models**
>
> Thank you for the suggestion. Extending our analysis beyond Gaussian single-index models is definitely an interesting direction. We make the following remarks.
> - In terms of data distribution, recent works have established universality results for single-index learning beyond Gaussian data, see [[Zweig et al. 2023]](https://openreview.net/forum?id=JkmvrheMe7), by utilizing the orthonormal basis functions of the data distribution or CLT-type arguments.
> - In terms of target functions, a natural extension would be multi-index functions such as XOR/parity, or more interestingly, hierarchical functions that require a deeper model to express. We intend to investigate such extensions as future work.
>
> We would be happy to clarify any concerns or answer any questions that may come up during the discussion period.

---

### Official Review · Reviewer_hCez · 2025-03-13

**Overall Recommendation:** 4

**Summary:**

This paper studies the tasking of learning single-index models $y = \sigma(x \cdot \beta)$ using a two-layer single-head
softmax transformer. The authors prove that a pretrained transformer can solve this task in context (different
$\beta$ in different prompts). When $\beta$ is sampled from the unit sphere in an $r$-dimensional subspace of
$\mathbb{R}^n$, the authors provide the following bounds on the sample complexity ($\mathrm{IE}$ and $\mathrm{GE}$
stand for the information and generative exponents of the link function $\sigma$, respectively):
* For pretraining, the number of prompts is $\tilde{O}( r^2 d^{\mathrm{IE}+2} \vee r^{1.5 \mathrm{GE}} )$ and
  the length/number of samples in each prompt is $\tilde{O}( r^{1.5 \mathrm{GE}} )$.
* For inference, the length/number of samples in the prompt is $\tilde{O}( r^{1.5 \mathrm{GE}} )$, which matches
  the SQ lower bound up to a multiplicative constant in the exponent.

The pretraining algorithm is a rather standard (standard in this area instead of in practice): they first
train the attention layer using one large step of GD and then train the output layer using ridge regression. They show
that after the first stage, the attention layer can recover the relevant $r$-dimensional subspace, and after the second
stage, the model learns to reconstruct the link function using samples.

## update after rebuttal
I thank the authors for the response. I agree that while the correlation can be easily extracted from the attention in a linear way, leveraging the nonlinearity to improve the sample complexity is non-trivial. Hence, I'll keep my score (4).

**Claims And Evidence:**

This is a theoretical paper and the authors provide complete proofs for their claims.
They also include some synthetic experiments to support their claims.

**Essential References Not Discussed:**

No.

**Experimental Designs Or Analyses:**

This is a theory paper and the synthetic experiments they provide are serviceable.

**Methods And Evaluation Criteria:**

Yes. (In-context) sample complexity is a standard metric to measure an algorithm's ability to learn single-index models
(in context).

**Other Comments Or Suggestions:**

Typos:
* line 188 right: ... inference-time sample complexity of $\tilde{O}( r^{\Theta(\mathrm{ie}(\sigma_*))} )$ ....
  Should be $\tilde{O}( r^{\Theta(\mathrm{deg}(\sigma_*))} )$.
* line 229 right: $\Gamma \in \mathbb{R}^m$. Should be $\Gamma \in \mathbb{R}^{d \times d}$.

**Other Strengths And Weaknesses:**

Overall, this is a well-written paper and has strong results. See the previous sections for the strengths.

One potential issue of this paper is that it relies heavily on certain properties of single-index models. In particular,
in this setting, the correct way to measure the correlation between two samples (in a given prompt) is exactly the
standard Euclidean inner product, so the attention layer only needs to identify the relevant subspace. This is more of
a coincidence instead of something universal.

**Questions For Authors:**

See the "Other Strengths And Weaknesses" section.

**Relation To Broader Scientific Literature:**

The main contribution of this paper over [Oko et al., 2024b] is that it improves the in-context sample complexity from
$r^{\Theta(\mathrm{deg})}$ to $r^{\Theta(\mathrm{ge})}$. The former can be achieved via using kernel methods (in
the relevant subspace) while to get the GE bound, the model needs to recover the actual direction $\beta$.

In addition, the authors use softmax attentions instead of linear attentions (which is widely used in theory as it is
usually easier to analyze) and the existence of this nonlinearity leads to a bound beyond the CSQ lower bound
$r^{\Theta(\mathrm{ie})}$ (which is what one will get from using algorithms such as online SGD to learn
single-index models).

**Theoretical Claims:**

I checked the proof strategy but did not check the detailed calculation. The proof looks believable to me, as
its components are either standard (ridge regression on the output layer) or believable (but non-trivial) extensions of
existing argument (e.g., learning the subspace using one large GD step, lower the IE to GE using monomial transformation
that comes from expanding the nonlinearity).

---

> ### Author Rebuttal · Authors · 2025-04-01
>
> We thank the reviewer for the helpful feedback. We will correct the typos and improve the writing of the manuscript. We address the technical points below.
>
> **Properties of single-index target**
>
> We agree with the reviewer that our current learnability result relies on the Euclidean inner product in the attention layer. We make the following remarks.
> - While the correlation between samples can be easily extracted from the attention, our analysis makes use of the Softmax function to transform the training data in some nonlinear fashion --- such nonlinear transformation is crucial to obtain an improved in-context sample complexity. In contrast, prior works using linear attention layers can only make use of (linear) correlation through the attention block.
> - We speculate that deeper architectures would be able to construct more complicated correlations between data points and further improve the in-context sample complexity (e.g., see recent works on the gradient-based learning of three-layer neural networks [[Nichani et al. 2024]](https://openreview.net/forum?id=fShubymWrc&noteId=efqPkOAyOx) [[Dandi et al. 2025]](https://arxiv.org/abs/2502.13961)), which is an important future direction.
>
> We would be happy to clarify any concerns or answer any questions that may come up during the discussion period.

---

### Official Review · Reviewer_7xsp · 2025-03-14

**Overall Recommendation:** 3

**Summary:**

This paper studies the optimization and statistical guarantees for the in-context learning of the Gaussian single-index function class using a Transformer with softmax attention. The derived inference-time sample complexity is tighter than the existing works, which indicates that pre-trained transformers can implement full statistical query (SQ) algorithms. During the proof, a mechanism of ICL is also characterized.

--------------------------
## Update after rebuttal.

My concerns are addressed overall. I prefer to keep the current score and support an acceptance.

**Claims And Evidence:**

The analysis is very impressive and solid.

**Essential References Not Discussed:**

N/A

**Experimental Designs Or Analyses:**

The experimental design makes sense, although I feel the experiments are not enough. Please check the part of weaknesses.

**Methods And Evaluation Criteria:**

N/A. This paper is mainly theoretical.

**Other Comments Or Suggestions:**

N/A

**Other Strengths And Weaknesses:**

The paper is well-written, and the theoretical analysis is impressive. I would like to mention some weaknesses here.

1. Experiments are not enough. (a) Figure 6.1 only compares the kernel method and ICL, which correspond to the kernel bound and the SQ lower bound. It is better to also show the result of the CSQ bound, which shall correspond to one-pass SGD on two-layer neural networks from lines 316-318 Right. (b) It is better if you can conduct some quantitative experiments on the order of the sample complexity, even if you use a much simpler model.

2. Algorithm 1 seems quite complicated. Is it designed for the simplicity of theoretical analysis, or is there any practical training strategy that is close to Algorithm 1?

**Questions For Authors:**

1. Can you summarize the technical novelty compared with [Oko et al. 2024b], especially about the difference in your settings (if any) and why you can have a stronger result than theirs?

2. Can you discuss some practical insights from your results? For example, what can people learn from your results to improve the training and testing performance?

Oko et al. 2024b. Pretrained transformer efficiently learns low-dimensional target functions in-context.

**Relation To Broader Scientific Literature:**

N/A

**Theoretical Claims:**

I checked the proof sketch, which looks reasonable and rigorous.

---

> ### Author Rebuttal · Authors · 2025-04-01
>
> We thank the reviewer for the helpful feedback. We address the technical comments below.
>
> **Additional experiments**
>
> Thank you for the suggestions on experiments. We have conducted an additional experiment to probe the test time sample complexity of GPT-2 models for learning single-index target functions, and demonstrate that the sample complexity surpasses the CSQ lower bound.
> - We pretrained GPT-2 on a degree-3 single-index model  $y=He_3(\langle\beta,x\rangle)$, in the setting where $r=d$, and plotted the relationship between the ambient dimensionality $d$ and the minimal test prompt length $N_\mathrm{test}$ required to surpass given test error thresholds.
> - In [[link]](https://osf.io/p87yc?view_only=be342dc212bb49cb8f72309fdc37d376), we observe that the estimated inference-time sample complexity is approximately $d^{1.13}\sim d^{1.16}$, which is lower than the kernel lower bound of $d^3$ and CSQ lower bound of $d^{1.5}$ for $\mathrm{He}_3$. Note that the CSQ lower bound also suggests a statistical hardness for online SGD – see e.g. [[Damian et al. 2023]](https://openreview.net/forum?id=73XPopmbXH&noteId=eMjWYg6qko).
>
>
> **Clarification on Algorithm 1**
>
> We make the following remarks on Algorithm 1.
> - Indeed Algorithm 1 involves a few ingredients that simplify the theoretical analysis, similar to recent works in the feature learning theory literature. In particular, a single gradient descent step (Stage I) has been used to model the early phase of gradient-based training on neural networks [[Ba et al. 2022]](https://openreview.net/forum?id=akddwRG6EGi) [[Damian et al. 2022]](https://proceedings.mlr.press/v178/damian22a/damian22a.pdf) and transformers [Oko et al. 2024b], during which the first-layer parameters align with the feature vector of the single-index target and escape saddle points in the nonconvex landscape.  The layer-wise training strategy – dividing the algorithm into two stages, one gradient descent step on the inner layer and ridge regression for the outer layer – is also common in these prior works, where the lower layer captures the relevant feature, while the top layer (+ reinitialized bias units) approximates the link function.
> - While our theoretical result applies to a specific parameterization and gradient-based learning algorithm, we empirically demonstrated in the additional experiment above that standard end-to-end training of a GPT2 model can achieve an in-context sample complexity that goes beyond the CSQ lower bound. We believe this suggests that mechanisms similar to our theoretical construction may be present in more realistic settings.
>
>
>
> **Distinctions from Oko et al. (2024b)**
>
> We will include a dedicated comparison section in the paper; here we briefly summarize the key differences.
>
> - Improved inference-time sample complexity – In Oko et al. (2024b), the derived inference-time sample complexity is $O(r^{\Theta(deg(\sigma_*))})$, which corresponds to the sample complexity of kernel methods on an $r$-dimensional subspace. In our paper, we derived an improved sample complexity of $O(r^{\Theta(gen(\sigma_*))})$ by showing that nonlinear transformers are capable of inference-time feature learning (beyond the kernel regime).
> - Absence of low-dimensional assumption – While Oko et al. (2024b) assumes that the dimensionality $r$ of pretraining task distribution is much smaller than the ambient dimensionality $d$, our analysis does not rely on this assumption.
> - Theoretical analysis – To derive the improved sample complexity $O(r^{\Theta(gen(\sigma_*))})$, we demonstrate that the nonlinear transformation applied to the output label $\{y_i\}$ reduces the information exponent of the link function $\sigma_*$ to its generative exponent. In particular, it is essential to show that the degree-$gen(\sigma_*)$ term in the Hermite expansion of the nonlinearly transformed labels (via the softmax transformation) is non-vanishing. This requires a careful tracking of the corresponding Hermite coefficients, which constitutes one of the most technically challenging parts of the analysis. This aspect is not addressed in Oko et al. (2024b), where it is only shown that the linear attention module computes correlations on the raw label $\{y_i\}$ and a fixed kernel basis $\{\phi(x_i)\}$ (constructed by a fixed MLP block).
>
>
> **Practical insights**
>
> Our goal is to rigorously study the statistical efficiency of ICL in a well-defined target function class (single-index models). Therefore, our analysis does not directly yield algorithmic improvements in practical settings. This being said, we believe that the mechanism that we theoretically investigate (in-context feature learning) serves as a foundation of future studies on the capabilities and limitations of ICL compared to other algorithms that act on the test prompt.
>
> We would be happy to clarify any concerns or answer any questions that may come up during the discussion period.

---

> > ### Comment · Reviewer_7xsp · 2025-04-02
> >
> > Thank you for the response. My concerns are addressed overall. I prefer to keep the current score and support an acceptance.

---

### Official Review · Reviewer_V9j6 · 2025-03-14

**Overall Recommendation:** 4

**Summary:**

This work studies in-context learning (ICL) of single-index models $\( y = \sigma(\langle x, \beta \rangle) \)$ using nonlinear transformers, focusing on inference-time sample complexity. The authors propose a two-stage training approach: (1) a single gradient descent step on the attention matrix to capture feature structure, and (2) ridge regression on the MLP layer to fit the nonlinear transformation. Unlike *Oko et al. (2024b)*, which assumes a low-dimensional feature structure, this work removes this restriction while achieving better inference-time sample complexity (Table 4.1).

The main findings show that nonlinear transformers outperform non-adaptive methods (e.g., kernel methods) by extracting task-specific features at inference time, without parameter updates. Moreover, the softmax attention mechanism plays a crucial role in feature learning, helping transformers surpass the Correlational Statistical Query (CSQ) lower bound, making them more sample-efficient than CSQ-based methods. The study provides rigorous theoretical guarantees on both pretraining complexity and inference complexity , demonstrating that pretrained transformers can adaptively learn features with superior statistical efficiency compared to traditional approaches.

## Update after rebuttal
Thank you for the thoughtful and detailed rebuttal. The clarification of the theoretical contributions—particularly the improved inference-time sample complexity and the removal of low-dimensional assumptions—meaningfully strengthens the paper. The additional experiments further support the main claims. Based on this, I am increasing my score and recommend acceptance.

**Claims And Evidence:**

The paper provides theoretical justification for its claims, particularly in demonstrating improved inference-time sample complexity over kernel methods and surpassing the CSQ lower bound. The main results are well-supported by formal optimization guarantees and statistical complexity bounds. Additionally, Table 4.1 effectively contrasts the findings with prior work (*Oko et al., 2024b*), highlighting the relaxed dimensional assumptions and improved efficiency.  However, some claims could benefit from stronger empirical validation. The synthetic experiments with GPT-2 provide limited practical evidence, as they do not comprehensively test real-world generalization or scalability. Additionally, while the softmax attention mechanism’s role in feature extraction is theoretically well-motivated, a deeper ablation study on its necessity and effectiveness would strengthen the argument. Overall, the core theoretical claims appear well-supported, but further empirical evidence would enhance the credibility of the results.

**Essential References Not Discussed:**

To the best of my knowledge, the paper appropriately cites and discusses relevant prior work.

**Experimental Designs Or Analyses:**

The synthetic experiment using GPT-2 provides a reasonable comparison between in-context learning with transformers and kernel methods, aligning with the paper’s theoretical claims. The choice of a single-index model with a Hermite polynomial link function is appropriate for testing inference-time feature learning.

**Methods And Evaluation Criteria:**

Yes, both the studied setting (in-context learning of single-index models with nonlinear transformers) and evaluation criteria (inference-time sample complexity) make sense. The single-index model is a well-established framework in statistical learning and serves as a meaningful testbed for understanding how transformers learn features at inference time. The focus on inference-time sample complexity is particularly relevant, as it directly measures the efficiency of transformers in extracting task-specific information without parameter updates—aligning well with real-world applications of in-context learning.

**Other Comments Or Suggestions:**

-

**Other Strengths And Weaknesses:**

Strengths

The paper tackles a more challenging setting than most theoretical works in the literature, which often rely on simplified transformer models. By incorporating softmax attention and an MLP layer, the study resembles real-world transformer architectures, making its findings more applicable. This more realistic setup allows the authors to establish better sample complexity results than *Oko et al. (2024b)*, demonstrating the advantages of nonlinear transformers for inference-time feature learning.

Weaknesses

1. Presentation and Organization Need Improvement:
   - The paper’s structure is difficult to follow, making it harder to grasp the key contributions at first glance.
   - Theorem 1 could be stated more clearly to improve readability.
   - The distinctions from Oko et al. (2024b) are spread throughout the paper, making them hard to evaluate. A dedicated comparison section in the related work would clarify these contributions.

2. Limited Experimental Evaluation:
   - The paper only includes a single synthetic experiment, which is not sufficient to validate the theoretical claims in real-world settings.
   - Adding an experiment on real-world data would significantly strengthen the empirical support for the paper’s conclusions.

**Questions For Authors:**

- Distinctions from Oko et al. (2024b): While the paper improves sample complexity results compared to *Oko et al. (2024b)*, the distinctions are spread across different sections. Could you provide a more explicit and structured comparison in the related work section to clarify the key differences?

- Role of Softmax Attention: The paper argues that softmax attention plays a crucial role in inference-time feature learning. Have you considered ablation studies or comparisons with linear attention to isolate its impact more clearly?

- Empirical Evaluation: The experimental validation is limited to a single synthetic experiment. Would it be possible to evaluate the method on real-world datasets, and if so, what would be the key challenges?

- Scalability of Pretraining Requirements: The pretraining task/sample complexity scales with \( d^{\Theta(ie(\sigma))} \). In practical applications, would this requirement be computationally feasible, especially for large-scale transformers?

**Relation To Broader Scientific Literature:**

I agree—showing that nonlinear transformers achieve better sample complexity than kernel methods is a strong contribution to the theoretical literature on in-context learning (ICL). This work extends Oko et al. (2024b) by improving sample complexity bounds under more general assumptions, reinforcing that transformers can surpass fixed-basis methods through adaptive feature learning. More broadly, the paper connects to research on neural network adaptivity, highlighting how softmax attention enables efficient inference-time learning. These findings contribute to the study of statistical efficiency in deep learning, showing that transformers can generalize more efficiently than traditional approaches.

**Theoretical Claims:**

The theoretical claims, particularly regarding inference-time sample complexity and softmax attention’s role in feature extraction, are well-structured and follow established techniques. However, I did not verify every step in detail. Key results, such as the gradient descent analysis on the attention matrix and the Hermite expansion argument, appear reasonable but would require closer scrutiny to confirm correctness. No obvious errors stood out, but a deeper review is needed for full validation.

---

> ### Author Rebuttal · Authors · 2025-04-01
>
> We thank the reviewer for the helpful feedback. We address the technical comments and questions below.
>
> **Distinctions from Oko et al. (2024b)**
>
> Based on your suggestion, we will include a dedicated comparison section in the paper; here we briefly summarize the key differences.
>
> - **Improved inference-time sample complexity** – In Oko et al. (2024b), the derived inference-time sample complexity is $O(r^{\Theta(deg(\sigma_*))})$, which corresponds to the sample complexity of kernel methods on an $r$-dimensional subspace. In our paper, we derived an improved sample complexity of $O(r^{\Theta(gen(\sigma_*))})$ by showing that nonlinear transformers are capable of inference-time feature learning (beyond the kernel regime).
> - **Absence of low-dimensional assumption** – While Oko et al. (2024b) assumes that the dimensionality $r$ of pretraining task distribution is much smaller than the ambient dimensionality $d$, our analysis does not rely on this assumption.
> - **Theoretical analysis** – To derive the improved sample complexity $O(r^{\Theta(gen(\sigma_*))})$, we demonstrate that the nonlinear transformation applied to the output label $\{y_i\}$ reduces the information exponent of the link function $\sigma_*$ to its generative exponent. In particular, it is essential to show that the degree-$gen(\sigma_*)$ term in the Hermite expansion of the nonlinearly transformed labels (via the softmax transformation) is non-vanishing. This requires a careful tracking of the corresponding Hermite coefficients, which constitutes one of the most technically challenging parts of the analysis. This aspect is not addressed in Oko et al. (2024b), where it is only shown that the linear attention module computes correlations on the raw label $\{y_i\}$ and a fixed kernel basis $\{\phi(x_i)\}$ (constructed by a fixed MLP block).
>
>
> **Additional experimental evaluation**
>
> Thank you for the suggestions on experiments. We make the following remarks.
> - Our goal is to rigorously study the statistical efficiency of ICL in a well-defined target function class (single-index models). Therefore, similar to prior works that studied ICL for linear regression [Garg et al., 2022; Von Oswald et al., 2023; Ahn et al., 2023; Mahankali et al., 2023a; Zhang et al., 2023], we conducted experiments (on GPT2 models) in a controlled synthetic setting to probe the sample complexity scaling. Pretraining on real-world datasets where the information/generative exponent of the target function is unknown is tangential to our main contribution.
> - To further validate the derived test-time sample complexity rate, we have added additional experiments to validate that the sample size scaling is indeed beyond CSQ (please refer to [[link]](https://osf.io/p87yc?view_only=be342dc212bb49cb8f72309fdc37d376), see also our response to reviewer 7xsp for details). Following your suggestion, we are currently running ablation studies to probe the necessity of the Softmax attention.
>
> We would be happy to clarify any concerns or answer any questions that may come up during the discussion period.

---

### Decision · Program_Chairs · 2025-05-01

**Decision:**

Accept (poster)

**Comment:**

Single-index models are functions of the form $x \mapsto \sigma(w_* \cdot x)$, where $\sigma$ is typically a nonlinear function characterized by favorable properties in its Hermite polynomial expansion. This paper examines how transformers can learn such functions from in-context examples provided as pairs $(x_i, \sigma(w_* \cdot x_i))$, and provides an analysis of the associated sample complexity. A key finding is that transformers can surpass kernel methods. While single-index models are not representable within a reproducing kernel Hilbert space, transformers exhibit an adaptive capability to learn relevant features, thereby reducing the sample complexity for learning single index models.

This paper differentiates itself from previous studies on in-context learning through its emphasis on the training dynamics of nonlinear attention mechanisms. However, the main theorem establishes a statistical performance bounds without explaining the mechanism of attention layers. In contrast, other studies in the literature have explicitly connected attention mechanisms to optimization procedures, thus explaining phenomena such as out-of-distribution generalization and adaptation to test distributions differing from the training distribution.

The results have been positively received by reviewers, who acknowledge that the paper is well-written and addresses an interesting theoretical topic. Therefore, I recommend acceptance.